# Common neural choice signals can emerge artefactually amid multiple distinct value signals

Romy Frömer ®[1,2,3,4] ✉, Matthew R. Nassar ®[2,5], Benedikt V. Ehinger ®[6] & Amitai Shenhav ®[1,2,7,8]

Previous work has identified characteristic neural signatures of value-based decision-making, including neural dynamics that closely resemble the ramping evidence accumulation process believed to underpin choice. Here we test whether these signatures of the choice process can be temporally dissociated from additional, choice-'independent' value signals. Indeed, EEG activity during value-based choice revealed distinct spatiotemporal clusters, with a stimulus-locked cluster reflecting affective reactions to choice sets and a response-locked cluster reflecting choice difficulty. Surprisingly, 'neither' of these clusters met the criteria for an evidence accumulation signal. Instead, we found that stimulus-locked activity can 'mimic' an evidence accumulation process when aligned to the response. Re-analysing four previous studies, including three perceptual decision-making studies, we show that response-locked signatures of evidence accumulation disappear when stimulus-locked and response-locked activity are modelled jointly. Collectively, our findings show that neural signatures of value can reflect choice-independent processes and look deceptively like evidence accumulation.

Over the past few decades, research has made major advances toward understanding how people make value-based choices between competing options (for example, items on a restaurant menu or in a store catalogue). This research has identified consistent neural correlates of the values of the options under consideration[1,2] and characterized the process that gives rise to decisions among them, both neurally and computationally[3–7]. However, drawing clear links between neural and computational investigations of value-based choice has been complicated by the fact that neural correlates of value can reflect processes outside of the ongoing decision (for a review see ref. 8). For instance, engaging with a choice set can trigger evaluations of one's options that are relatively automatic (for example, Pavlovian)

and potentially independent of the decision process itself[2,9–16]. Distinguishing such choice-independent neural value signals from those that play a mechanistic role in the choice process requires disentangling the two types of signal within a measure of neural activity that provides the temporal resolution to uncover their unfolding dynamics. Here we use electroencephalography (EEG) to explicitly tease apart value-based neural dynamics attributable to decision making from those that are not, and reveal, surprisingly, that 'only' the latter, choice-independent value signals were to be found.

Prevailing computational models show that decision making can be described as a process of noisy evidence accumulating to a decision threshold, providing an account of choice behaviour (choices and

[1]Cognitive, Linguistic, and Psychological Sciences, Brown University, Providence, RI, USA. [2]Carney Institute for Brain Sciences, Brown University, Providence, RI, USA. [3]School of Psychology, University of Birmingham, Birmingham, UK. [4]Centre for Human Brain Health, University of Birmingham, Birmingham, UK. [5]Department of Neuroscience, Brown University, Providence, RI, USA. [6]Stuttgart Center for Simulation Science, University of Stuttgart, Stuttgart, Germany. [7]Department of Psychology, University of California Berkeley, Berkeley, CA, USA. [8]Helen Wills Neuroscience Institute, University of California Berkeley, Berkeley, CA, USA. ✉e-mail: r.froemer@bham.ac.uk

response times (RTs)) across a variety of different choice settings[17,18]. In the context of value-based decision-making, putative correlates of this evidence accumulation process have been identified throughout the brain[4–6,19], often reflecting variability in the strength of evidence in favour of a particular option or attribute, and a subset of studies has used temporally resolved estimates of neural activity to capture the dynamics of this evidence accumulation process. From this work, a putative EEG signature of evidence accumulation has emerged in the centroparietal positivity (CPP), both for perceptual[20–22] and value-based[4] choice. Researchers have shown that the CPP demonstrates three characteristic elements of evidence accumulation (cf. Fig. 1a): (1) following stimulus presentation, activity is greater and peaks earlier when decision-related evidence is stronger (consistent with a more rapid rise of evidence accumulation when choices are easier), (2) activity peaks around the time of the response (consistent with a common response threshold) and (3) in the period leading up to the response, due to the slower accumulation and thus shallower slope, activity is greater when evidence is weaker and/or responses are slower. The latter effect is sometimes part of a cross-over pattern and complemented by an opposite effect at the time of response, reflecting perhaps a decrease in decision threshold for longer RTs through an urgency signal or a modulation of the overlapping readiness potential[20,23,24], specifically in paradigms where there is a clear onset and offset of the physical evidence[25,26]. The CPP is thus a potential index of value-based processing that is integral to decision making per se.

However, recent studies have shown that neural correlates of choice value can reflect appraisals of the choice set as a whole, that take place irrespective of whether the participant is comparing their options[9,13]. For instance, using functional magnetic resonance imaging (fMRI), dissociable components of the brain's valuation network[1] were found to track how much participants liked a set of choice options overall versus elements of the choice process itself, for example, whether they were engaged in choice versus appraisal and how demanding the choice was[11–13]. These studies suggest that value-related activity may emerge soon after the stimuli are presented that is tied to choice-independent, appraisal-like processes. They further predict that signatures of this appraisal process should be distinguishable from the evidence accumulation signatures described above, both in terms of the specific correlates of value that each of these tracks and, critically, in terms of their temporal dynamics (Fig. 1a): whereas appraisal-related processes should index the overall value of a choice set, and occur transiently and locked to the presentation of the choice options, choice-related processes should index comparisons between one's options (for example, the relative value of the chosen vs unchosen option). The latter may reflect evidence accumulation, in which case such activity should ramp up between stimulus presentation and response selection (cf. refs. 3,4), or other choice-related processes (for example, monitoring one's confidence). Past work has been unable to test these predictions because it lacked the temporal resolution needed to demonstrate these distinct temporal profiles and to formally tease apart signals that meet the criteria of evidence accumulation from those that do not. As a result, it is unknown whether these value-related signals are indeed distinct or merely two components of a unitary choice process (Fig. 1a; cf. refs. 5,27).

To fill this critical gap, we had participants make value-based decisions while undergoing EEG, and explicitly disentangled putative correlates of choice-independent appraisal processes (for example, overall value and set liking) from correlates of the process of choice comparison (for example, relative value and choice confidence). This allowed us to test two alternative hypotheses (Fig. 1a). One hypothesis is that value-related EEG activity would only emerge in the form of an evidence accumulation process, in which case we would expect any value-related variables (including overall value) to demonstrate characteristic patterns of stimulus-locked and response-locked

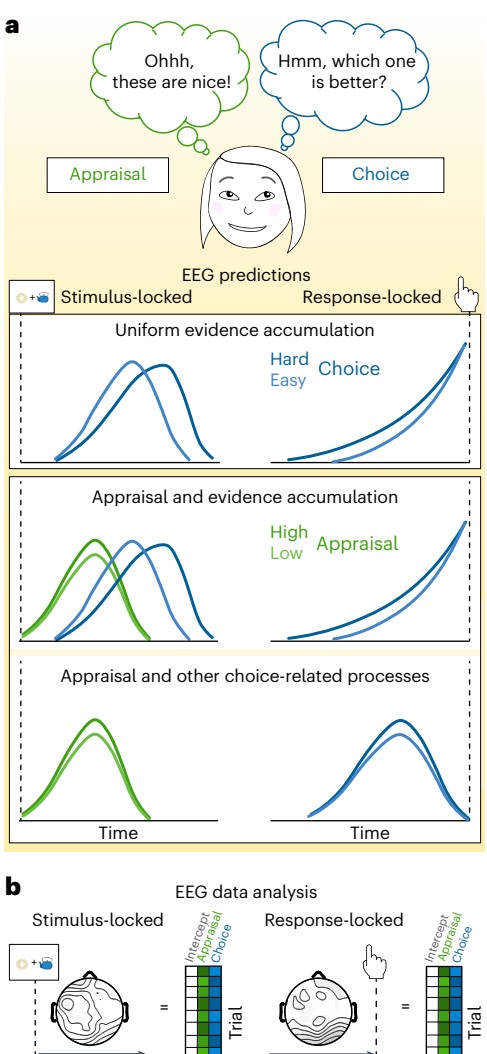

**Fig. 1 | Dissociating appraisal- and choice-related processes. a**, A set of options can elicit distinct evaluations, such as appraisal of the options and choice among them. Different frameworks make different predictions for whether and how these should affect neural activity locked to the response versus the stimulus. Top: one account predicts that appraisal and choice reflect different temporal stages of a unitary evidence accumulation process, such that relevant variables (for example, value similarity, blue) would be reflected first in stimulus-locked activity and culminate at the time of the response. Middle/bottom: alternative accounts predict that appraisal reflects an independent process that emerges during stimulus presentation. Under these accounts, neural activity correlated with choice-related variables may emerge as a parallel process of evidence accumulation (that is, both stimulus-locked and response-locked, middle) or in some other form as a non-accumulation-related signal (shown response-locked only as a stylized example, the shape and directionality of the signals may differ; bottom). **b**, To dissect the temporal dynamics of appraisal- and choice-related neural activity, we regress single-trial EEG activity onto appraisal-related and choice-related variables (see Fig. 2c) separately for stimulus- and response-locked activity. $b_0$, $b_1$ and $b_2$ are weights for intercept, appraisal PC and choice PC, respectively.

activity previously observed, for instance, in the CPP (Fig. 1a, top). The alternative hypothesis, motivated by our recent fMRI findings, is that we would observe appraisal-related patterns of activity that are selectively locked to stimulus presentation (reflecting their potentially more reflexive nature), independently of choice-related value signals. These choice-related value signals may take the form of CPP-like

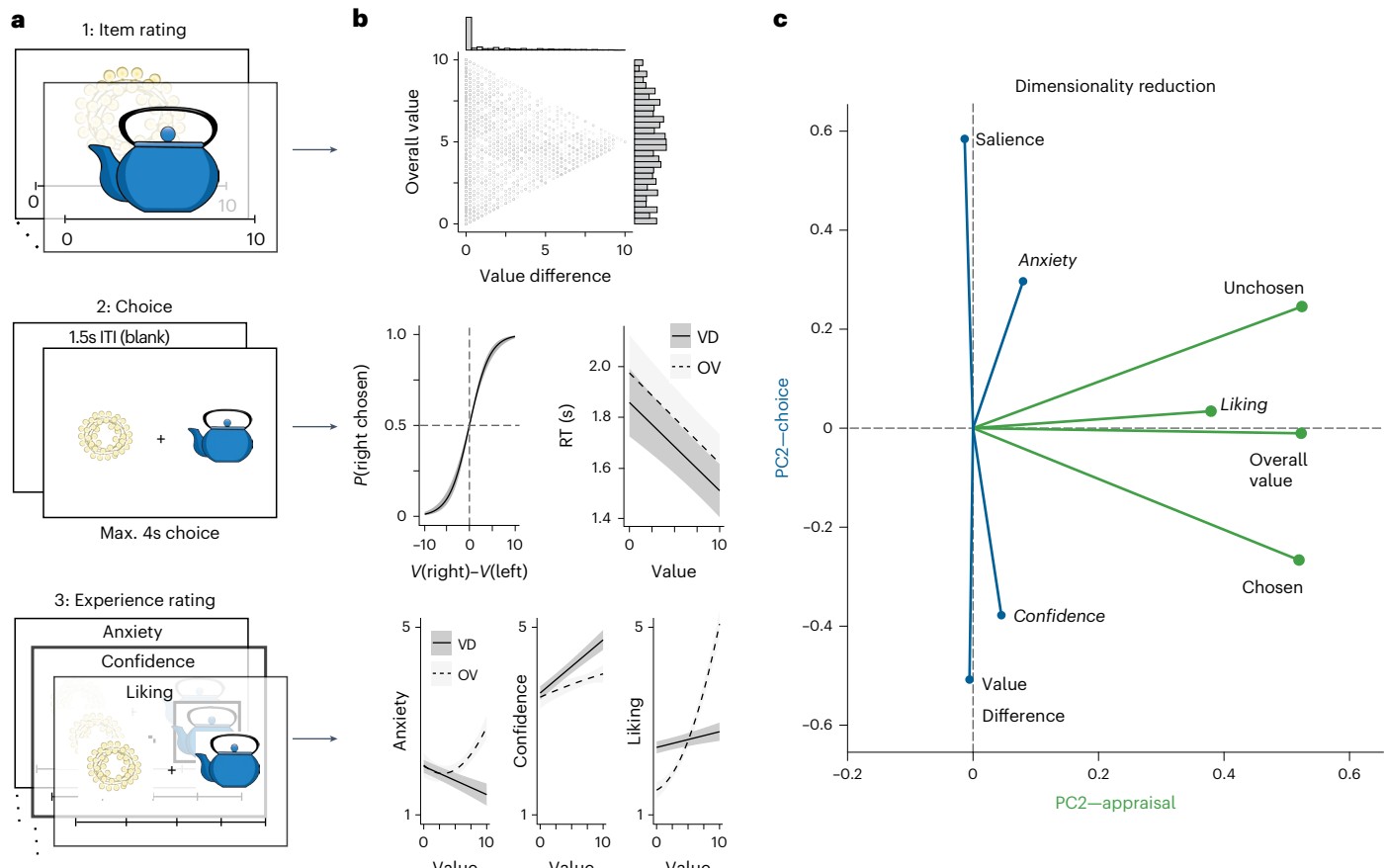

**Fig. 2 | Integrating multiple measures of appraisal and choice. a**, Participants performed the experiment in three phases, rating consumer goods individually (Phase 1) before choosing between pairs of those items (Phase 2, ITI denotes intertrial interval) and finally rating their subjective experiences of those choices (Phase 3: set liking (appraisal), confidence and anxiety). **b**, Responses across these phases provided different measures of appraisal and choice. Top: option sets for Phase 2 were generated on the basis of participants' initial item ratings to vary in their overall (average) value and the absolute difference between the values of the two options. Middle: choices varied with the relative value of the chosen vs unchosen option, and RTs varied with both overall value and value difference. Shown are linear mixed effects model predictions. Error bars indicate 95% CIs. Bottom: overall value (OV; dashed) and value difference (VD; solid)

differentially influenced experiences of choice anxiety, confidence and set liking. Shown are linear mixed effects model predictions. Error bars indicate 95% CIs. *P*(right chosen) denotes the probability of choosing the item on the right. *V*(right) denotes the value of the item on the right, and *V*(left) denotes the value of the item on the left. **c**, We used principal component analysis to reduce the dimensionality of our measures, identifying two principal components in our variable set, clustering naturally into variables associated with appraisal (PC1) versus choice (PC2). Component loadings for each measure are represented by their distance from the origin. Anxiety is self-reported anxiety/stress while making the decision. Liking is self-reported attractiveness of the item set. Confidence is the self-reported confidence in the ultimate choice.

evidence accumulation signals (Fig. 1a, middle), or some other form (Fig. 1a, bottom).

We were able to rule out the first hypothesis, instead finding appraisal-related EEG activity that was both stimulus-locked and independent of choice comparison-related activity. Putatively choice-related EEG activity, by contrast, occurred in a distinct temporal window (response-locked) and with a different spatial profile (fronto-posterior) than the spatiotemporal cluster we identified for appraisal (stimulus-locked and parietal). Remarkably, these putative choice value signals 'also' did not meet key criteria for an evidence accumulation signal. Instead, and even more striking, we found that such apparent evidence accumulation signals can emerge from 'choice-independent' stimulus-locked activity, as an artefact of standard approaches to investigating evidence accumulation processes, due to bleed-over between stimulus-locked and response-locked activity (particularly for rapid choices). Confirming this, when we apply a novel analysis approach that deconvolves stimulus and response-related activity to four previous decision-making studies, we eliminate response-locked signatures of evidence accumulation previously observed in those data. As a result, our findings collectively,

and unexpectedly, support a third hypothesis (Fig. 1a, bottom): that value signals separately correlated with appraisal-related and choice-related processes emerge during value-based decision-making, but neither of these reflect evidence accumulation.

## Results

We recorded EEG while participants made incentive-compatible choices between pairs of options (consumer goods). Choice sets varied in the overall and relative value of the two options, as determined by ratings of individual items given earlier in the session (Fig. 2a,b). Participants' choice behaviour was consistent with that observed in previous studies and predicted by prevailing models of evidence accumulation[3,4,17]: participants chose faster (linear mixed models (LMM) fixed effect: $b = -348.7$, $t = -6.00$, $P < 0.001$, two-sided, 95% confidence interval (CI) = [−462.72 to −234.72]) and in a manner more accurate/consistent with their initial item ratings (generalized linear mixed model (GLMM) fixed effect: $b = 4.54$, $z = 12.40$, $P < 0.001$, two-sided, 95% CI = [3.83 to 5.26]) as value difference increased, and also chose faster as overall value increased (LMM fixed effect: $b = 357.14$, $t = 6.70$, two-sided, $P < 0.001$, 95% CI = [−461.62 to −252.65]; Fig. 2b and Supplementary

Table 1). After making all of their choices, participants provided subjective ratings of the choice sets (how much they liked the sets as a whole) and of the choices themselves (how much choice anxiety they had experienced while making the choice, and how confident they were in their final decision).

## Distinct spatiotemporal clusters for appraisal vs choice

We predicted that we would find a temporal dissociation between neural activity associated with appraisal versus choice, whereby appraisal-related activity would be temporally coupled with the onset of the stimuli, whereas choice-related activity would be temporally coupled with the response. To test this prediction, we analysed the effects of appraisal- and choice-related variables on the same EEG data locked to the onset of the stimuli versus locked to the response. Given that a number of different variables captured our two constructs of interest (for instance, appraisal was captured by the overall (average) value of the choice set and subjective ratings of set liking, and choice was captured by the difference between the option values and subjective ratings of confidence (cf. Fig. 2b)), we used principal component analysis (PCA) to reduce the dimensionality of these single-trial measures and improve the robustness of our estimates of each construct.

This PCA identified two reliable principal components (Fig. 2c and Supplementary Table 2), one associated with how positively the option set had been assessed overall (for example, positively loading on overall value and on ratings of choice set liking) and the other associated with how difficult the choice comparison was (for example, negatively loading on value difference and on ratings of choice confidence). We termed these the Appraisal PC and Choice PC, respectively.

We regressed stimulus and response-locked single-trial EEG activity for each participant at each sensor and each time point onto these appraisal- and choice-related PCs (cf. Fig. 1b), and identified significant stimulus- and response-related clusters associated with each PC using cluster-based permutation tests on the resulting $t$-statistics at the group level. We found that the PCs mapped onto distinct spatiotemporal patterns (Fig. 3 and Supplementary Fig. 1). In line with our predictions, we found that our Appraisal PC explained EEG activity locked to (and following) stimulus onset (Fig. 3a; $P = 0.040$, two-sided cluster permutation test; see Methods), but not locked to the response (neither preceding nor following). The largest stimulus-locked cluster had a parietal distribution, peaking around 710 ms at CP2. Further in line with our hypothesis, we observed significant Choice PC-related activity locked to (and preceding) the response (Fig. 3b; $P = 0.002$ for a positive cluster and $P < 0.001$ for the negative cluster based on two-sided cluster permutation tests), but not locked to and following the stimulus. The response-locked Choice PC activity included a frontocentral positive cluster, peaking around −566 ms at FC4, and a posterior negative cluster, peaking around −818 ms at P5. Similar clusters emerged when performing separate analyses on variables that constituted each of the PCs (Supplementary Table 3).

## Neither cluster is consistent with evidence accumulation

Our analyses suggest that two value-related EEG patterns emerge during value-based choice. The first of these was stimulus-locked, tracked appraisal-related measures (that is, assessments of how much the participant liked the set overall), and had a timing and spatial distribution similar to that of the late positive potential (LPP), an event related potential (ERP) commonly found to index the affective salience of stimuli[28–30], suggesting that this stimulus-locked cluster may index processes unrelated to the choice itself. By contrast, the response-locked value clusters we observed tracked measures of choice comparison (for example, how much more valuable one option was than the other and how certain the participant was in their choice) and had a spatiotemporal profile consistent with frontoparietal EEG patterns that have been previously implicated in value-based decision-making[3,31]. The posterior cluster overlapped topographically with the CPP[4,20–22].

We therefore reasoned that this response-locked cluster was a good candidate for providing an index of the evidence accumulation process leading up to the choice, and performed follow-up analyses to test whether activity in this or the more anterior cluster met the criteria for such a process.

Typically, evidence accumulation signals are also evident in stimulus-locked activity because responses fully overlap with the stimulus time-window, leading to the characteristic greater and earlier peaks for faster evidence accumulation. Since in our study response times are longer, the absence of this pattern is expected. Surprisingly, however, neither of our response-locked clusters met the two response-locked criteria for signatures of evidence accumulation. First, rather than ramping towards a temporally common peak (marking the response threshold) immediately before a response, we found that activity peaked more than 500 ms before the response. Second, rather than seeing greater activity leading up to a response on harder choice trials, reflecting the slower rate of evidence accumulation expected for those trials (compare light and dark lines in Fig. 3b), we instead found the opposite. Both frontal and posterior clusters showed greater amplitudes for easier as opposed to more difficult trials.

The fact that our data failed to meet either of these criteria was particularly notable for our posterior cluster, given its apparent overlap with the centroparietal positivity, the event-related potential most strongly associated with evidence accumulation. To better understand this discrepancy from previous work, we performed follow-up analyses focused directly on the CPP proper. Specifically, we tested for a key marker of evidence accumulation traditionally observed in the CPP: that slower trials (which require more evidence accumulation and exhibit a shallower slope) should show larger amplitudes leading up to the response than faster trials[3,4]. In our study, CPP amplitudes in the pre-response time window (700 to −200 ms as in previous work[4]) instead showed the opposite pattern: significantly larger for shorter relative to longer RTs (LMM fixed effect: $b = -0.47$, $t = -3.10$, $P = 0.002$, 95% CI = [−0.77 to −0.17]). Similar findings emerge when using value difference as a proxy for choice difficulty (as in the analyses above): CPP amplitude was larger for easier than harder choice trials (LMM fixed effect: $b = 0.62$, $t = 1.77$, $P = 0.077$, 95% CI = [−0.07 to 1.31]) rather than the reverse.

One possible explanation for this apparent contradiction has to do with differences in the timing of choices in our study relative to previous studies. Our participants were given up to 4 s to make their choice, in contrast to shorter response windows in earlier work (for example, 1.25 s; ref.[4]). The evidence accumulation signal may therefore have been more spread out in time within our data, leading to the expected greater activity for slower/more difficult trials to occur earlier. To investigate this possibility, we examined the average ERP curves on trials above and below the median RT (Fig. 4a).

Moving far enough back in time, to ~1 s before response onset, we do see that the relative magnitudes of slow and fast trials reverse such that slow trials elicit greater activity than fast trials, as predicted by an evidence accumulation account. However, at odds with this account, we also see that slower trials elicit much 'earlier' peaks than faster trials. To understand why these peaks were systematically shifting in time, we plotted the single-trial amplitudes underlying the median RT averages and sorted them by RT (Fig. 4a, top). This revealed a marked positive amplitude response in all trials (red line) ~350 ms following stimulus onset (black line), and the rise and peak of the ERP curves approximately followed the respective temporal distributions of this response (compare Fig. 4a top and bottom).

We therefore considered that the discrepancy between our findings and those previously observed (cf. Fig. 4b, top) may have been caused by overlap with this choice-unrelated response (stimulus ERP jitter, Fig. 4b, middle) which may have masked the expected evidence accumulation signal. We therefore performed a separate analysis,

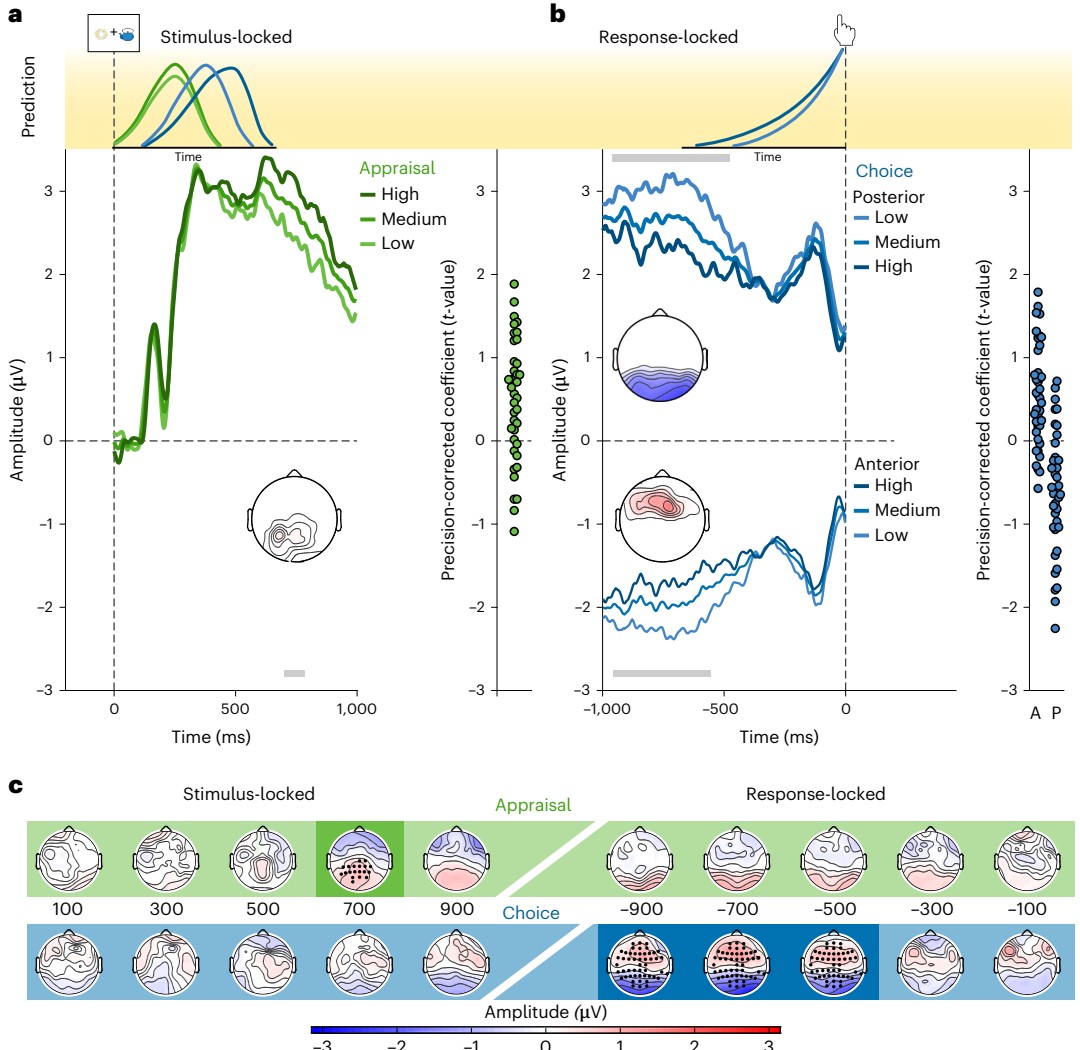

**Fig. 3 | Appraisal and choice exhibit dissociable spatiotemporal profiles.**
**a,b**, Curves show predicted ERPs for each level of a given PC from the regression model (visualized in discrete terciles), averaged within the electrodes in the respective cluster, within 1 s following stimulus onset (**a**) and preceding the response (**b**). Note that the median RT was -1.7 s, so there is little overlap between stimulus- and response-locked data. Grey bars indicate cluster time points that significantly exceed permutation cluster masses (two-tailed test) for either the positive or negative clusters. Topographies display *t*-values within these clusters aggregated across cluster time points. To visualize the variability in the data underlying these clusters, individual participants' *t*-values are displayed

on the right of each panel, aggregated within cluster times and electrodes.
**a**, Stimulus-locked centroparietal positive activity increases with higher Appraisal PC scores (more positive appraisal). **b**, Response-locked posterior positivities and frontocentral negativities are reduced for higher Choice PC scores (more difficult trials). Individual participants' *t*-values are shown separately for the anterior (A) and posterior (P) cluster on the right. **c**, Coefficient topographies for the Appraisal PC (top) and Choice PC (bottom), averaged across sliding 200 ms windows aligned to stimulus (left) and response (right). Darker regions indicate time windows encompassing significant clusters (cluster permutation corrected *P* < 0.05).

analogous to standard event-related analyses for fMRI, which explicitly modelled stimulus-locked and response-locked activity, allowing them to be formally deconvolved from one another[32–34]. Similar to our previous analyses, this approach again identified a positive stimulus-locked appraisal cluster with a centroparietal distribution (peak ~810 ms at electrode Pz, *P* = 0.004, two-sided cluster permutation test) and response-locked choice clusters over frontal (positive, peak around −722 ms at electrode AFz, *P* = 0.010, two-sided cluster permutation test) and parietal (negative, peak around −616 ms at electrode P8, *P* = 0.010, two-sided cluster permutation test) sites, respectively (Supplementary Figs. 1 and 2). This approach also showed no stimulus-locked choice effects, or response-locked appraisal effects. Thus, despite successfully disentangling the stimulus- and response-locked activity, it did not change our overall pattern of results; even after controlling for component overlap, our response-locked pattern remained inconsistent with evidence accumulation.

**Component overlap can look like evidence accumulation**
These findings led us to question whether rather than 'masking' evidence accumulation signals in our findings, stimulus-locked activity may have spuriously 'caused' signatures of evidence accumulation in previous work. As we indicated earlier, most studies investigating evidence accumulation signals in EEG involved much faster decisions of ~750 ms on average in ref. 4 (cf. Fig. 5a), ~800 ms on average in ref. 20 and ~1 s on average in ref. 21 (except for a perturbation condition with additional stimulus dynamics). One possibility is therefore that the characteristic response-locked evidence accumulation pattern in previous studies was driven by overlap between 'stimulus-related' activity (for example, related to the salience of the stimuli) and response-locked activity[34]. This 'component overlap' account can explain basic features of CPP data (Fig. 5a, bottom) and makes a distinct prediction: rather than activity remaining locked to the response (as predicted by an evidence accumulation account), a component overlap account predicts

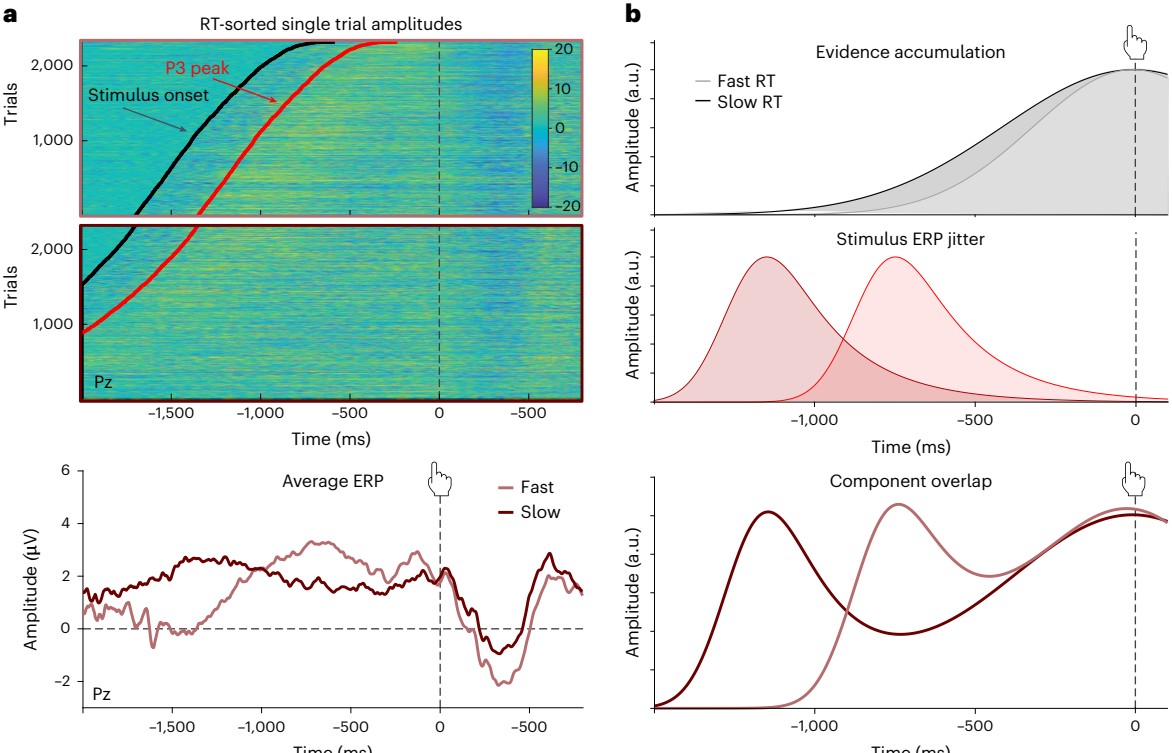

**Fig. 4 | Stimulus-locked EEG activity can produce spurious patterns of response-locked activity due to variability in response times. a**, Top: response-locked single-trial ERPs at Pz sorted by RT for trials faster (top box) and slower (bottom box) than the median RT. Black lines mark stimulus onsets and red lines mark the average onset of the stimulus-locked P3. Amplitudes are colour-coded with the colour scale ranging from −20 to 20 µV (colour bar). Bottom panel: averaging across these trials produces differential patterns of response-locked ERPs for faster trials relative to slower trials. **b**, The 'unexpected'

patterns observed in **a** can result from component overlap. Top: evidence accumulation signals expected for fast and slow RTs, respectively. Middle: when activity is locked to the response, this introduces jitter in stimulus-locked activity, with stimulus-related activity appearing earlier and earlier as RTs increase (that is, the greater the delay between stimulus and response). Bottom: the convolution of the two patterns above can produce a pattern that is dominated by component overlap and obscures signatures of evidence accumulation. Such a pattern is similar to that shown in **a**.

that the peak of the CPP should move back in time as RTs increase (Fig. 5b, bottom). Consequently, for short RTs, variability in the extent of overlap between stimulus-related and response-related activity with RT would produce an artefactual ramping signal in average ERPs that appears steeper for faster and shallower for slower responses. To test this possibility, we produced the same ERP plots as above for the subset of trials that had RTs shorter than 1.25 s, as in previous studies (Fig. 5b, top). Compared with the entire dataset, the peaks for this subset of short RTs moved closer to the response and, crucially, a pattern reminiscent of the CPP emerged, with slower RT trials in this range displaying a larger parietal positivity of up to 500 ms before the response. Our collective pattern of results across both short and long RTs therefore exactly matches the predictions of the component overlap account (Fig. 5a,b, bottom).

To test whether these findings replicate in an independent sample, we examined the CPP in a separate value-based decision-making dataset (Study 2; $N = 39$) and found the same pattern: at odds with typical findings, CPP amplitudes in the pre-response time window (700 to −200 ms; ref. 4) were again significantly larger for shorter relative to longer RTs (LMM fixed effect: $b = -1.01$, $t = -6.58$, $P < 0.001$, two-sided, 95% CI = [−1.30 to −0.71]), and for easier than for harder choice trials (LMM fixed effect: $b = 0.87$, $t = 2.32$, $P = 0.027$, two-sided, 95% CI = [0.13–1.60]). When plotting fast and slow RT trials in a fast subset of the data (RT < 1.25 s) and across all trials as in Fig. 5, we reproduce the moving peaks that follow the distribution of stimulus onsets (see Supplementary Fig. 3).

However, it is still difficult to generalize from these results because in both studies, our average RTs were longer than those in previous

studies, even when only focusing on our subset of short-RT trials (those below 1.25 s). As a result, rather than peaking exactly at the time of response (as is characteristic of past CPP results), EEG activity during that subset of trials peaks slightly before the response. To provide a more direct test of our hypothesis that the evidence accumulation effects in the CPP could originate from a component overlap artefact, we re-analysed EEG findings from four previous datasets in which response times were more tightly constrained: one collected during value-based decision-making (Study 3; ref. 4) and the other three collected during perceptual decision-making[25,35,36]. In the value-based decision-making study, participants ($N = 21$) chose among pairs of snack items while their EEG was recorded and had to respond within 1.25 s from stimulus onset. The authors found that behaviour was well-captured with a drift diffusion model (DDM) and reported the typical response-locked CPP evidence accumulation signal (cf. Fig. 4a, top). In the first perceptual decision-making study (Study 4; ref. 36), EEG was recorded while participants ($N = 40$) decided whether a deviant object in a circular array of objects was on the left or right side of the display (Fig. 6). Objects in the array were chosen to be visually similar and presented either intact or blurred, which serves as an index of evidence strength and modulated performance accordingly (lower accuracy and slower RTs for blurred compared with intact stimuli). Stimuli were presented for 200 ms and participants had up to 2 s from stimulus onset to respond. In the second perceptual decision-making study (Study 5; ref. 25), EEG was recorded while participants ($N = 17$) determined which of two overlaid gratings had a higher contrast. Unlike the other studies, stimuli were not suddenly presented but faded in at equal contrast and switched to the target contrasts after a random

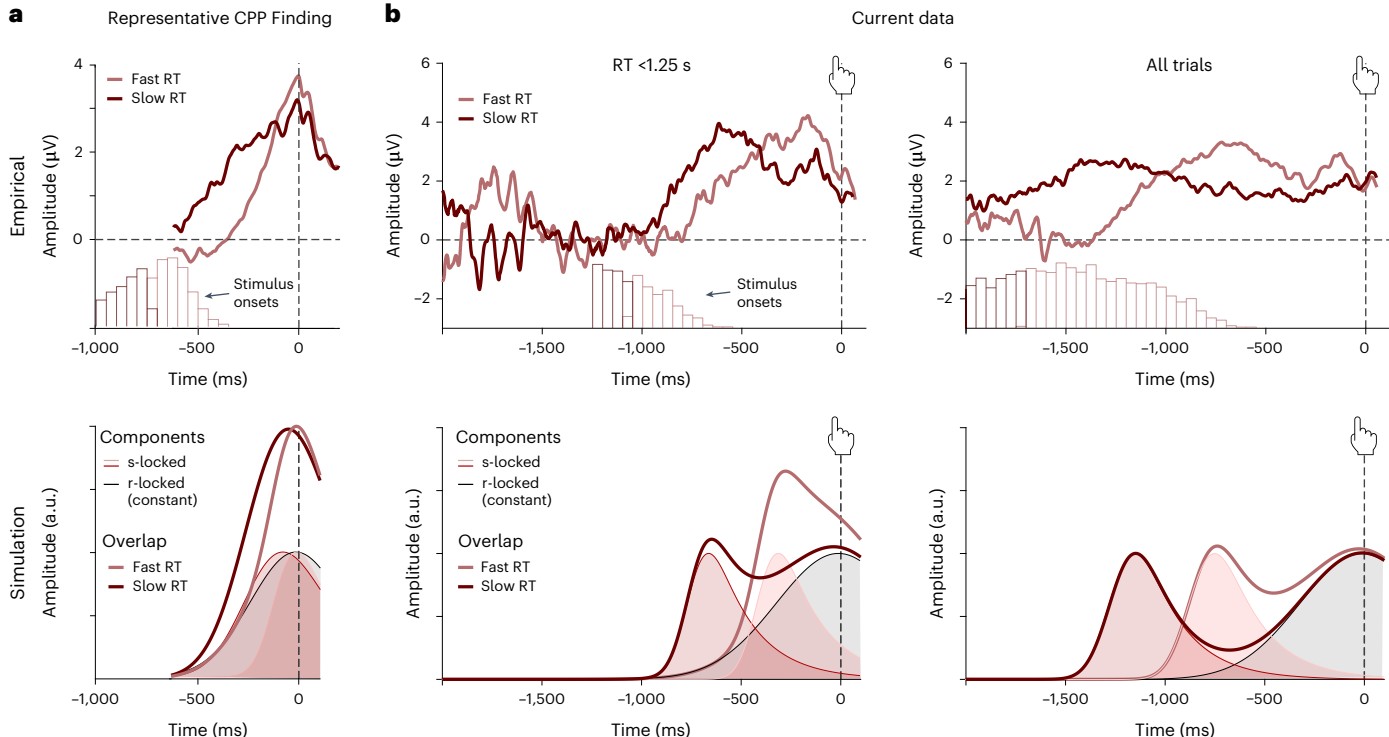

**Fig. 5 | Evidence accumulation signals emerge as an artefact of component overlap. a**, A representative CPP finding (data from ref. 4) shows faster ramping for ERP curves from fast compared with slow trials. Histograms show distributions of stimulus onsets relative to the response for each average ERP curve. Simulation: component overlap can generate an evidence accumulation-like pattern under plausible assumptions about RT distributions. Note that these simulations assume the same response-related component for all trials (black line), omitting any evidence accumulation. For fast response times (<900 ms on average), overlapping stimulus- and response-related components are predicted to resemble a single ramp-like component. The peak time and shape of the underlying component will depend on the mean and width of the RT distribution of trials in each ERP average. **b**, Data from our decision-making study are consistent with component overlap predictions. Shown are average ERPs for median split fast and slow RTs below 1.25 s (left) RT across all trials (right), respectively. Histograms show distributions of stimulus onsets relative to the response. Stimulus-evoked peaks move further away from the response as response times increase (top). Peak times and widths of the observed ERP curves vary with the mean and width of the RT distribution of the underlying trials. This pattern of results is consistent with a component overlap account (bottom).

time interval simultaneously with a tone signalling the change. The evidence was then presented for 2,400 ms within which participants had to respond. In the third perceptual decision-making dataset (Study 6; ref. 35), EEG was recorded while participants ($N = 17$) judged the overall blueness versus redness of compound stimuli that varied in the blue, red and purple hue of their components. Stimuli were presented for 160 ms and participants had 1,200 ms to decide.

When we separately re-analyse stimulus-locked and response-locked activity using mass-univariate analyses analogous to standard ERP analyses, we find the characteristic CPP indices of evidence accumulation over centroparietal sites in all four studies. Response-locked analyses revealed that activity peaked at the time of the response and rose with a steeper slope for fast relative to slow trials in value-based decision-making (Fig. 6a, top), and for intact relative to blurred stimuli (Fig. 6b, top) and fast relative to slow trials (Fig. 6c,d, top) in perceptual decision-making, resulting in more positive CPP amplitudes for trials that putatively required more evidence accumulation (cf. Fig. 1a). The centroparietal positivity in all studies therefore meets response-locked criteria for signatures of evidence accumulation. However, because these analyses do not explicitly account for the overlap between stimulus-related and response-related components, they cannot distinguish whether the response-locked patterns reflect evidence accumulation or stimulus-related activity. To formally disentangle these, we again applied the deconvolution approach introduced earlier[32], including stimulus and response events in a single model of neural responses. After deconvolution, we no longer find significant response-locked signatures of evidence accumulation in any of the

datasets (Fig. 6a–d, bottom), suggesting that in our previous analysis, this characteristic pattern of evidence accumulation was predominantly an artefact of component overlap (see Supplementary Fig. 4 for corresponding stimulus-locked activity).

So far, we assumed that signatures of evidence accumulation should be closely tied to the response. This is the typical prediction in the literature and plausible if non-decision time (comprising stimulus and motor processes at both ends of the evidence accumulation process) is constant, and the time required for motor processes is short. However, both stimulus and motor processes can vary from trial to trial, leading to an evidence accumulation process that is not closely tied to either stimulus or response. The only model that explicitly and independently accounts for these variance components is the neurally informed DDM[26]. To test whether unfold's deconvolution approach would be robust to detecting evidence accumulation signals, either response- or stimulus-locked, when accounting for these different forms of variability, we simulated the neurally informed DDM (simulation code adapted from ref. 20) for strong and weak signals with varying ratios of stimulus and response variability. The simulations confirm that signatures of evidence accumulation need not be response locked. The simulations also show that, as response variability increases, unfold will increasingly assign activity to the stimulus event rather than the response event (Supplementary Fig. 5). Importantly, these simulations show that 'any' signatures of evidence accumulation are recoverable, either at the stimulus, the response or both. The simulations also provide us with qualitative alternative predictions for signatures of evidence accumulation under different ratios of stimulus and motor

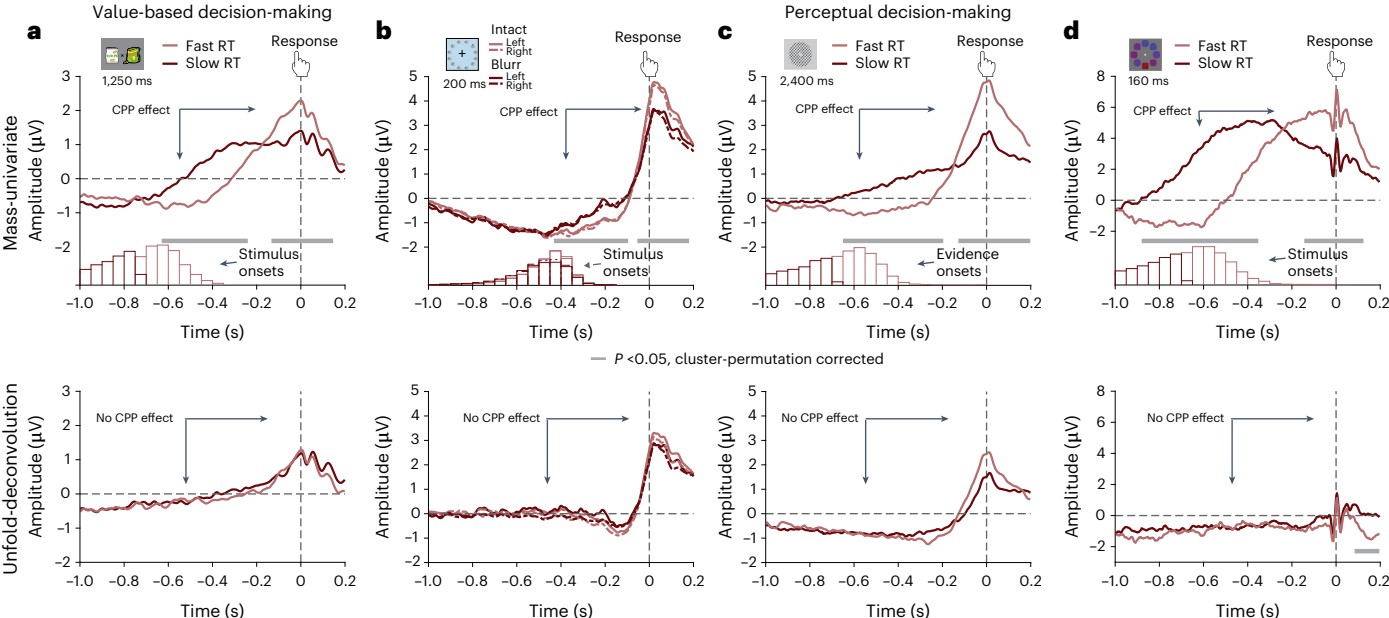

**Fig. 6 | Response-locked evidence accumulation patterns vanish when correcting for component overlap. a**, Regression ERPs (rERPs) from a mass-univariate re-analysis (top) of an independent value-based decision-making dataset (ref. 4, cf. Fig. 5a) exhibit a CPP with characteristic signatures of evidence accumulation. Decisions based on weaker evidence (slow compared with fast) are associated with a slower response-locked ramping of CPP amplitude, resulting in larger CPP amplitudes before the response. When re-analysing these data with a deconvolution approach that models both stimulus-related and response-related activity (bottom), we find that response-locked rERPs no longer show the characteristic evidence accumulation pattern. **b–d**, rERPs from a mass-univariate analysis (top) of three independent perceptual decision-making datasets exhibit a CPP with characteristic signatures of evidence accumulation. Decisions based on weaker evidence (blurred stimuli/slower responses) are associated with slower response-locked ramping of CPP amplitude, resulting in larger CPP amplitudes before the response. When re-analysing these data with a deconvolution approach that models both stimulus-related and response-related activity (bottom), we find that response-locked rERPs no longer show the characteristic evidence accumulation pattern. Data are shown with average reference. **b**, Data from ref. 36. **c**, Data from ref. 25. **d**, Data from ref. 35. **a–d**, Grey bars indicate temporal clusters significant at $P < 0.05$, cluster permutation corrected for multiple comparisons.

variability. Specifically, under the model of ref. 26, we would expect that as motor variability increases, evidence accumulation signals would be evident in a greater 'stimulus'-locked amplitude for easy/fast compared with hard/slow trials.

Of the four datasets we analysed, the only one in which we observed a pattern in principle consistent with such stimulus-locked evidence accumulation signatures, but also with non-integration accounts, was the visual search dataset[36] (see Supplementary Fig. 4b). To test whether signatures of evidence accumulation may have been moved towards the stimulus in the remaining datasets and thus missed in our previous analyses, we re-analysed these data again, this time allowing stimulus-locked data to vary with response time in addition to allowing response-locked data to vary. The resulting patterns in the remaining datasets showed neither significant response-locked signatures of evidence accumulation, nor the predicted stimulus-locked amplitude modulations, and were thus inconsistent with any of the simulated patterns (Supplementary Fig. 6). For Study 5, the results are somewhat ambiguous due to a significant baseline difference, which if taken into account may produce a stimulus-locked peak difference in the expected direction. However, these findings are complemented by results showing that signatures of evidence accumulation can no longer be found using mass-univariate analysis following RT-agnostic overlap correction of the data (Supplementary Fig. 7).

## Discussion

Previous work has identified reliable neural correlates of choice value and interpreted them as elements of a uniform choice process in which option values are compared through an accumulation-to-bound process. These interpretations have been reinforced by evidence of such neural correlates ramping up towards the response, as would be expected of activity associated with evidence accumulation[4]. However,

recent work suggests that certain neural correlates of choice value are unrelated to goal-directed processes such as evidence accumulation and instead reflect the appraisal of one's options[9,11,13]. Here we tested whether we could use EEG to temporally dissociate such choice-independent value signals from choice-related value signals. We anticipated that choice-independent value signals would follow shortly after stimulus onset, whereas choice-related activity should be coupled to and lead up to the response. We found this expected temporal dissociation. Remarkably, however, we found that the identified choice-related activity was inconsistent with evidence accumulation and that instead, putative signatures of evidence accumulation can emerge artefactually in standard response-locked analyses from overlapping stimulus-related activity. Across three value-based decision-making studies and three perceptual decision-making studies, we show that signatures of evidence accumulation are absent when stimulus-locked and response-locked activity are sufficiently separated in time and disappear when overlapping activity is formally deconvolved.

It is important to note that our observation that correlates of appraisal (for example, overall value) occur earlier in time than correlates of choice (for example, value difference) does not in and of itself suggest that these signals arose from independent processes. Indeed, this same temporal pattern (overall value signals preceding value difference signals) is predicted to emerge from certain forms of unitary evidence accumulation processes, such as those of refs. 5,27,37,38. However, models such as these 'also' predict that all these value signals should emerge locked to the response (Supplementary Fig. 8). At odds with such an account, we only found stimulus-locked correlates of appraisal (unlike response-locked choice correlates). Our findings are thus better explained by 'separate' mechanisms related to appraisal and choice.

The distinctiveness of these two sets of value signals is further supported by the fact that they were linked to distinct topographies. As predicted, we found appraisal-related activity temporally locked to stimulus onset, as reflected in a parietal positivity consistent with an LPP ERP component[28–30]. The distribution and timing of this component parallel previous ERP findings on single-item valuation[39,40] and therefore may be interpreted as reflecting an initial valuation stage before the onset of an independent choice comparison process[41–43]. Notably, the LPP is sensitive to affective information even when that information is not task relevant[29,44] and its putative sources[45] overlap with the pregenual anterior cingulate cortex and posterior cingulate cortex regions in which we previously found choice-independent set appraisals[13]. Collectively, these findings suggest that rather than an initial choice-related valuation step, these appraisal-related signals reflect an automatic valuation signal[46], or enhanced attention to such motivationally relevant events[47,48]. Accordingly, we found that the variable that best predicted activity in our appraisal cluster was a participant's affective appraisal of the set (that is, set liking; Supplementary Table 2). Thus, appraisal-related activity may reflect initial (and perhaps reflexive) affective reactions to the stimuli (cf. refs. [11,13]) and possibly serve to inform control decisions[45,49] and/or future choices[50].

In contrast, choice-related activity was temporally locked to the response and was characterized by a prominent frontocentral negativity and concomitant posterior positivity, consistent with previous findings demonstrating increased time–frequency coupling between frontoparietal regions and stronger frontocentral beta power during value-based compared with perceptual decision-making[3,31]. However, across two independent studies, follow-up analyses showed that this pattern of activity was inconsistent with it reflecting the 'evidence accumulation' process leading up to the choice[3,4,20,21] in that amplitudes were larger for easier (or faster) rather than for more difficult (or slower) trials. Thus, rather than merely constituting a negative result (that is, lacking support for evidence accumulation reflected in the CPP), our results consistently 'positively' contradict such predictions in favour of a different explanation. Our findings also rule out alternative versions of this evidence accumulation account whereby the decision threshold (or urgency signal) varies between decision types with known differences in difficulty[35] or over the course of the decision[26]. These varying-threshold accounts would still predict that activity would be locked to one's response and are thus ruled out by the backward-shifting peaks we observed.

If these choice value correlates do not in fact reflect elements of the evidence accumulation process, what might they reflect? A prominent alternative account of such correlates would propose that signals associated with choice difficulty (for example, value difference) that we observe in our choice clusters might instead reflect monitoring (for example, conflict or confidence), which could inform higher-order decisions about further information sampling and potential information gain[51–63]. Recent work in value-based decision-making is converging on the idea that value-based choice as studied here requires higher-order decisions on gaze/attention allocation in the service of information sampling that fundamentally rely on representations of both value and uncertainty[54–56,58,64–67]. This functional interpretation is consistent with proposed loci of CPP activity in dorsal anterior cingulate cortex[4] and decrements in choice consistency when frontocentral coupling is disrupted during value-based choice[31]. While intriguing, this interpretation requires additional work to test specific predictions of a monitoring or active information search account.

Whatever the nature of these signals, our results call for caution when interpreting response-locked neural patterns as evidence accumulation. Across six datasets, we found that evidence accumulation signatures in the response-locked CPP may artificially arise from response time-dependent overlap with stimulus-related processing. This was true across both value-based and perceptual decision-making tasks. Considering the value-based decision-making studies only, one may have hypothesized that there is no CPP signature of value-based evidence accumulation, because the CPP is specific to perceptual evidence accumulation that precedes and is a necessary precursor to value-based evidence accumulation (cf. ref. [8]), perhaps implemented through subcortical circuits. Yet, the collective findings of our re-analysis of three perceptual decision-making datasets speak against this hypothesis.

This is particularly notable since in their original analyses, ref. [25] had used a different algorithm, RIDE[68], to subtract stimulus-locked activity associated with the auditory evidence onset cue and found the typical CPP patterns intact. This method separates early stimulus-locked components that are tightly temporally coupled from later activity that is not closely temporally coupled to the response, a central component. However, merely subtracting the stimulus-locked component determined this way neither offers information as to whether this central component is more aligned with the stimulus or the response, nor does it account for this component in response-locked analyses of the remaining data. If this central component occurred somewhat aligned to the stimulus, albeit more loosely than the early stimulus-locked activity, as a transient ERP, analysing it locked to the response would still result in the same component overlap effect identified in this paper, and under certain conditions give the appearance of a ramp. Our findings suggest that the central component preserved in ref. [25] is indeed locked to the stimulus as a canonical P3 ERP. This component is commonly observed for most behaviourally relevant visual stimuli, including feedback, regardless of response times, and is thought to potentially reflect a form of monitoring[69].

A crucial signature of evidence accumulation is that the corresponding signal peaks close to the time of the response, with that peak occurring earlier for faster compared with slower decisions. This is frequently observed for the CPP in perceptual decision-making when the onset of the relevant stimulus is purposefully obscured, thus when the 'subjective' onset of the stimulus can vary relative to the 'objective' onset[20,26,70,71]. However, other decision-making studies that have identified the CPP as a signature of evidence accumulation only show response-locked activity[4,35]; in some cases where stimulus-locked activity was examined, including our present results, the expected latency effect was not found[45]. Our work highlights the importance of testing multiple predictions.

Doing so, several studies have shown that evidence accumulation signals simulated from diffusion model fits to behaviour reproduce the CPP in detail, including its stimulus- and response-aligned waveforms, the relationship between its pre-response amplitude and choice RT, and accuracy and modulations of its pre-response amplitude by experimental manipulations of time pressure and previous knowledge (for example, refs. [21,22,26]). Here we highlight several cases in which some of the relevant characteristics could emerge artefactually from component overlap by virtue of evidence accumulation and component overlap sharing a common link to response time. Thus, in addition to testing multiple predictions of evidence accumulation accounts, future research can avoid misinterpretations of neural activity by inspecting ERP image plots for temporal patterns in single-trial data and deconvolving stimulus and response-locked signals[32] to show that the putative evidence accumulation pattern cannot be produced by component overlap alone.

Perhaps more importantly, our results reveal that the very nature of the evidence accumulation signal expected in the study at hand may vary drastically as a function of decision parameters, such as stimulus or motor variability, the shape of the decision bound/urgency parameters and so on. Rather than testing out-of-the-box predictions, we therefore recommend that future research into signatures of evidence accumulation apply the analysis approaches outlined above to the empirical data as well as simulations from a best-fitting model informed by independent measures of stimulus and motor variability, as well as bound/urgency parameters[26].

While we have provided evidence for the pervasive risk that these artefacts pose for inferring evidence accumulation from response-locked signals, we cannot claim to have definitively ruled out that the CPP can in certain cases carry the signature of an evidence accumulation signal. For instance, one finding that is not easily accounted for by component overlap is the observation of ref. 21 that perturbations in the stimulus lead to expected patterns in the CPP (see also ref. 72). In a continuous detection task, where participants needed to indicate when a stimulus was disappearing, O'Connell and colleagues[21] showed that briefly increasing stimulus contrast led to an attenuation of the average stimulus-locked CPP that ramped up again as stimulus contrast continued to reduce. Such a pattern is difficult to explain with component overlap alone but may still be accounted for by other non-integration accounts. Novel approaches are therefore needed to test the evidence accumulation hypothesis against alternative non-integration models[73,74]. Along these lines, some studies have employed repeated stimulus presentations to better characterize how signatures of evidence accumulation evolve with each piece of evidence[75,76].

Our findings build on recent work in non-human animals, which has demonstrated that signatures of evidence accumulation can be necessary but not sufficient to conclude that a given neural population underpins the evidence accumulation process that drives choice. Using deactivation approaches, such studies have called into question the role of candidate regions of decision making in parietal and prefrontal cortex that showed patterns expected for evidence accumulation using standard approaches[59,77–79] (but see also ref. 80). Similar to these findings, our work shows that evidence of accumulation is not sufficient to argue for an evidence accumulation account, and that to better understand the array of signals that appear over the course of a decision, we need to incorporate insights from affective science, metacognition and cognitive control[8].

## Methods

### Participants

For the main study, 48 participants were recruited from Brown University and the general community. Of these, 9 had to be excluded due to technical problems during data acquisition. The final sample consisted of 39 participants (27 female), with a mean age of 20.84 years (s.d. = 3.90). Participants gave informed consent and received US$10 per hour for their participation (US$30 for the entire experiment). In addition to the compensation, participants could win one of their choices at the end of the experiment. The study was approved by Brown University's Institutional Review Board (IRB) (Approval number: 1606001529).

The sample in study 2 comprised 39 participants recruited from Brown University and the general community. Participants (26 female, mean age 23.92, s.d. = 5.14) gave informed consent and received US$10 or US$15 per hour for their participation (the hourly rate as per IRB approval was increased after the study commenced and participants' payments were adjusted accordingly moving forward). This study was approved by Brown University's IRB (Approval number: 1606001529) and not incentivized.

Please see the original publications for detailed participant information in Study 3 ($N$ = 21; ref. 4), Study 4 ($N$ = 40; ref. 36), Study 5 ($N$ = 17; ref. 25) and Study 6 ($N$ = 17 after exclusion of 3 participants with non-matching behavioural and EEG data[35]).

### Task and procedure

The main experiment consisted of 3 parts: value rating, choice and subjective experience rating (Fig. 1a). The experimental procedure is an adapted version of that used in previous studies[11,13] to meet the requirements of EEG, specifically in the choice part.

In the first part, participants were presented with consumer goods, one at a time, and asked to rate how much they would like to have each

of them on a continuous scale from 0 to 10 with zero being 'not at all' and 10 being 'a great deal'. Labels presented below each item supported their identification. Participants were encouraged to use the entire scale. On the basis of individual ratings, choice sets were created automatically, varying value difference and set value such that in half of the choices, variance in value difference was maximized, while in the other half, value difference was minimal and variance in set value was maximized[12].

In the second part, participants had to choose between two items presented left and right of a fixation cross by pressing the 'A' or 'L' key on a keyboard with their left or right index finger, respectively. At the beginning of the choice part, participants were placed at 90 cm distance to the screen with the keyboard in their lap and their fingers placed on the response keys. Images were presented with a size of 2 ° visual angle (115 pixels) each, centred at 1.3 ° visual angle (77 pixels) from a centrally presented fixation cross. Thus, the entire choice set extended to maximally 2.3 ° visual angle in each hemifield. This small stimulus size was chosen as to reduce eye movements by presenting the major portion of the stimuli foveally within a radius of ~2 ° visual angle[81]. At the time of the response or after a maximum duration of 4 s, the stimuli vanished from the screen and a fixation cross was presented for a constant 1.5 s intertrial interval. Before the beginning of the choice part, participants were informed that one of the choices would be randomly selected for a final gamble at the end of the experiment that would give them the opportunity to win the item they chose on that trial ($N$ = 20 who won and received an item).

In the third part, participants were presented with all choices again to sequentially rate (1) their anxiety while making each particular choice, (2) their confidence in each choice and (3) how much they liked each choice set, respectively. For all subjective evaluations, the scales ranged from 1 to 5, mapped onto the corresponding number keys on the keyboard.

In the beginning and at the end of the experimental session, demographic and debrief data were collected, respectively, using Qualtrics. All subsequent parts were programmed in Psychophysics Toolbox[82,83] for MATLAB (v.2016b, MathWorks) and presented at 60 Hz on a 23 inch screen with a 1,920 × 1,080 resolution. Before the main experiment, participants filled in computerized personality questionnaires (Behavioural Inhibition/Activation Scales (BIS/BAS), Neuroticism subscale of the NEO Five Factor Inventory, Intolerance for Uncertainty Scale, and Need for Cognition). These data were not analysed for the present study.

Study 2 differed in the following respects (cf. ref. 84): before the value ratings, participants viewed all items twice, once with labels and once without. During the second viewing, they were asked to indicate whether they could recognize each item without seeing the label. During value ratings, participants for each item additionally evaluated their confidence in the value rating on a scale from 1 (not at all confident) to 5 (very confident). During the choice phase, rather than all at once, items were presented by alternating individually at the centre of the screen until a decision was made, or 5 s elapsed while the duration of each item presentation was varied. One item was always presented longer on average than the other. Specific presentation durations on each turn were drawn from different distributions for long (mean = 500 ms, s.d. = 100 ms) and short presentations (mean = 200, s.d. = 50); these distributions were informed by previous work[85]. Response buttons corresponding to each item were coded via blue and red coloured frames around the options and manipulated independently of the order and duration of item presentation.

In Study 3 (ref. 4), participants chose among pairs of previously rated snack items and had to respond within 1.25 s from stimulus onset. The difficulty of trials was manipulated by varying the value difference between options across 4 levels (1 through 4).

In Study 4 (ref. 36), participants decided whether a deviant object in a circular array of objects was on the left or right side of the display.

Objects in the array were chosen to be visually similar and presented either intact or blurred, which serves as an index of evidence strength and modulated performance accordingly (lower accuracy and slower RTs for blurred compared with intact stimuli). Stimuli were presented for 200 ms and participants had up to 2 s from stimulus onset to respond.

In Study 5 (ref. 25), participants performed a contrast discrimination task in which they decided which of two overlaid grating patterns had a higher contrast. Participants were cued to either emphasize accuracy or speed and were rewarded when their responses met the current condition and punished when they did not. A trial began with a regimen cue, followed by neutral stimuli with equal contrast, which changed to the target stimulus alongside a tone to signal evidence onset and stayed on screen for 2,400 ms. Participants received feedback following their response.

In Study 6 (ref. 35), participants judged whether stimuli, each consisting of a circular array of 8 red, blue and purple circles, were more blue or more red on average. Across trials, the coloured circles varied in their colour strength (for example, how clearly the stimulus was blue or red rather than purple) and in their variance (for example, how variable the hues of the circles were). Stimuli were presented for 160 ms and participants had 1,200 ms to respond. Participants were cued to one of four conditions (high mean, low variance; low mean, low variance; high mean, high variance; low mean, high variance). They also provided prospective and retrospective confidence judgments.

## Psychophysiological recording and processing

EEG data were recorded from 64 active electrodes (ActiCap, Brain Products) referenced against Cz with a sampling rate of 500 Hz using Brain Vision Recorder (Brain Products). Eye movements were recorded from electrodes placed at the outer canti (LO1, LO2) and below both eyes (IO1, IO2). Impedances were kept below 5 kΩ. EEG analyses were performed using customized MATLAB (v.2022b; MathWorks) scripts and EEGLab (v.13_6_5b; ref. 86) functions (cf. ref. 36 for an earlier version of the pipeline). Offline data were re-referenced to average reference and corrected for ocular artefacts using brain electric source analyses (BESA[87]) based on individual eye movements recorded after the experiment. The continuous EEG was low-pass filtered at 40 Hz (eeglab FIR-filter, default filter coefficients). For mass-univariate analyses (see below), choice data were segmented into epochs of 4.2 s locked to stimulus onset and 2.8 s relative to the response, with 2 s pre and 800 ms post response. Epochs were baseline corrected to the 200 ms pre-stimulus interval for both segmentations. Trials containing artefacts (exceeding amplitude thresholds of ±150 µV or a gradient of 50 µV) were excluded from further analyses. Unfold analyses were performed on unsegmented, preprocessed data as described below.

EEG data acquisition and processing for study 2 was identical except that passive Ag/Cl electrodes were used, and that segments were restricted to 2 s post stimulus and pre response, with 200 ms pre stimulus and post response, respectively.

Please see the original publications for detailed information on EEG data acquisition in Study 3 (ref. 4), Study 4 (ref. 36), Study 5 (ref. 25) and Study 6 (ref. 35). We obtained preprocessed data for Studies 3 and 4. Raw data for Study 5 were concatenated across blocks, re-referenced and low-pass filtered at 40 Hz. A subset of blocks (18 total from 298 blocks across all participants) could not be matched with the behavioural data and were excluded from further analyses. We obtained data for Study 6 with ocular artefact reduction, downsampled to 250 Hz and removed horizontal and vertical EEG channels as well as M2. Data were then matched with behavioural data and low-pass filtered at 40 Hz.

## Analyses

Behavioural data were analysed using linear mixed effects models as implemented in the lme4 package (v.1.1–31)[88] for R (4.2.2 (2022-10-31)[89]) in RStudio (v.2022.12.0 + 353). P values were computed using the sjplot package (v.2.8.12)[90]. We modelled main effects for value variables (both fixed and random effects) in line with previous work[5,9]. Random effects components were removed if they explained no variance[91]. Predictors in all analyses were mean centred, values were scaled to maximum equals 1 for ease of reporting. Choices were analysed using generalized linear mixed effects models using a binomial link function, with the dependent variable being the probability of choosing the right item. In these cases, reported fixed effects are conditional on the random effects because marginal fixed effects are difficult to estimate using generalized linear mixed models.

Appraisal and Choice principal components were derived from principal component analysis of all participants data on all trials on which a choice was made in time (4,637/4,680 trials), with the following variables normalized to 0–1 ranges: chosen value (value of the item that was ultimately chosen), unchosen value (value of the item that was ultimately not chosen), value difference (maximum value minus minimum value), overall value (average of both values), set salience (absolute mean-centred overall value), anxiety, liking and confidence. Thus, the dimensions of the input matrix were 4,637 × 8. An initial exploratory PCA with permutation testing identified 2 principal components with eigenvalues greater than the 95th percentile of the distribution from shuffled data. We thus derived 2 principal components using MATLAB's pca function, rotated the factors (normalized varimax) and derived trial-wise scores by multiplying the trial indicators with the factor loadings. We thus reduced the variables above to one score for Appraisal and one for Choice for each trial.

EEG data were analysed using a mass-univariate approach employing custom MATLAB scripts adapted from refs. 92,93: for each participant, voltages at each electrode and time point (downsampled to 250 Hz) as dependent variables were regressed (using MATLAB's regress function which implements least squares regression) against trial parameters and an intercept term as independent variables to obtain regression weights for each predictor, similar to difference wave ERPs for each condition in traditional approaches (cf. ref. 33). These regression weights were weighted by transforming them into t-values (dividing them by their standard error), effectively biasing unreliable estimates towards zero, and then submitted to group-level cluster-based permutation tests, employing a cluster-forming threshold of $P = 0.005$. Clusters with masses (summed absolute t-values) larger than 2.5% of the maximum cluster masses obtained from 1,000 random permutation samples were considered significant. Observed cluster mass was compared to a permutation distribution to get a percentile rank, which was inverted (such that lower numbers corresponded to more unlikely events), divided by 100 (to convert the percentile to a decimal) and multiplied by 2 (to obtain a P value for a two-sided test). We separately analysed stimulus-locked and response-locked EEG data in the 1,000 ms time interval following the stimulus and preceding the response, respectively. These time intervals were chosen to include sufficient trials at all time points. Data points outside the current trial range (following the response in stimulus-locked data and preceding the stimulus onset in response-locked data) were set to 'nan' to avoid spill-over from other trials or intertrial intervals. In the main analyses, the PC loadings for Appraisal and Choice PCs were included as independent variables with the intercept term. In three control analyses with the sets of variables underlying the PCs, we entered as independent variables either overall value and value difference, chosen and unchosen value, or liking, confidence and anxiety alongside the intercept term.

For the deconvolution analyses, we conducted first-level analyses on preprocessed data using the unfold toolbox[32]. This MATLAB/Julia toolbox implements multiple regression with combined linear deconvolution for multivariate time-series similar to FIR-GLM analyses in fMRI. That is, by providing event-timings and per-event regression formulas, it allows us to disentangle temporally overlapping ERP responses (for detailed introduction to the method, see ref. 32 or ref. 34). Stimulus onsets and responses were modelled

simultaneously with the same regressors as in the main analyses. Deconvolution was implemented using FIR/stick basis functions, with time expanded to ±2 s around the respective events. Artefacts (amplitudes exceeding ±250 μV) were detected and removed using the built-in threshold functions. No baseline corrections were applied. The obtained betas were submitted to the same cluster-based permutation analyses for second-level analyses as described above.

Study 4 perceptual decision-making data[36] were analysed using the same procedures as just described, except that we used ±1 s time windows due to the faster pace of the task and additionally computed mass-univariate betas without overlap correction for comparison. The value-based decision-making data in Study 3 (ref. 4), as well as perceptual decision-making data in Studies 5 (ref. 25) and 6 (ref. 35) were re-analysed analogously using the Julia implementation of the unfold toolbox[94]. Regressors for ref. 36 were visual field and stimulus quality, and both stimulus and response-locked activity were modelled with both regressors. Regressors for the three other studies were median split RT for response-locked activity, whereas we initially only modelled intercepts for stimulus-locked activity. In a set of control analyses, we re-analysed the data of the latter three studies to test whether stimulus- and response-locked activity varied in ways predicted by unfold analyses of simulated data under different stimulus and response variability regimes.

**Simulations.** We simulated the neurally informed DDM on the basis of code shared by Simon Kelly and Redmond O'Connell. In this model, stimulus processing as well as motor processes vary, so that the timing of the evidence accumulation process in between both is jittered. Both jitters were modelled using the simplifying assumption of uniform distributions with duration 'stim_jitter' and 'motor_jitter', respectively. Further, we report the ratio stim_jitter/motor_jitter, indicating whether the evidence accumulation process is more closely associated with the stimulus or the motor. We simulated 3 scenarios: (1) 2/1 ratio, whereby stim_jitter was set to 0.2 s and motor_jitter to 0.1 s, (2) 1/1 ratio, where both were set to 0.2 s and (3) 1/2 ratio, where stim_jitter was set to 0.1 s and motor_jitter to 0.2 s.

### Reporting summary
Further information on research design is available in the Nature Portfolio Reporting Summary linked to this article.

## Data availability
Data for Studies 1, 2 (only data used here), 3 and 4 are available on different platforms with links provided in GitHub at https://github.com/froemero/Common_Neural_Choice_Signals_emerge_artifactually (ref. 95).

## Code availability
All code can be accessed at https://github.com/froemero/Common_Neural_Choice_Signals_emerge_artifactually (ref. 95).

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

## Acknowledgements

This work was funded by a Center of Biomedical Research Excellence grant P20GM103645 from the National Institute of General Medical Sciences (A.S.), grant R01MH124849 from the National Institute of Mental Health (A.S.), a grant from the National Science Foundation (Collaborative Research in Computational Neuroscience Award #2309022) to A.S. and R.F., and by the Deutsche Forschungsgemeinschaft (DFG, German Research Foundation) under Germany's Excellence Strategy – EXC 2075 – 390740016 (B.V.E.). The funders had no role in study design, data collection and analysis, decision to publish or preparation of the manuscript. W thank W. Mahaphanit, C. Braun and H. Xu for assistance in data collection; K. Unger, A. Mueller and D. McCarthy for support with the EEG setup; A. Vaidya for generously sharing code and advice; A. Pisauro et al., N. Steinemann et al. and A. Boldt et al. for sharing their data, additional information for their re-analysis, and thoughtful questions; R. Polania and M. J. Frank for helpful discussion; and S. Kelly and R. O'Connell for extensive, fruitful discussions and for sharing their simulation code.

## Author contributions

R.F. and A.S. conceived the study. R.F. collected the data. R.F. analysed the data, except for unfold analyses and simulations in Julia, which were performed by B.V.E. B.V.E. and M.R.N. provided analysis guidance and example analysis code. R.F. wrote the first draft, and M.R.N., B.V.E. and A.S. edited the paper.

## Competing interests

The authors declare no competing interests.

## Additional information

**Correspondence and requests for materials** should be addressed to Romy Frömer.

# Reporting Summary

## Statistics

For all statistical analyses, confirm that the following items are present in the figure legend, table legend, main text, or Methods section.

| n/a | Confirmed | |
|---|---|---|
| ☐ | ☒ | The exact sample size ($n$) for each experimental group/condition, given as a discrete number and unit of measurement |
| ☐ | ☒ | A statement on whether measurements were taken from distinct samples or whether the same sample was measured repeatedly |
| ☐ | ☒ | The statistical test(s) used AND whether they are one- or two-sided *Only common tests should be described solely by name; describe more complex techniques in the Methods section.* |
| ☐ | ☒ | A description of all covariates tested |
| ☐ | ☒ | A description of any assumptions or corrections, such as tests of normality and adjustment for multiple comparisons |
| ☐ | ☒ | A full description of the statistical parameters including central tendency (e.g. means) or other basic estimates (e.g. regression coefficient) AND variation (e.g. standard deviation) or associated estimates of uncertainty (e.g. confidence intervals) |
| ☐ | ☒ | For null hypothesis testing, the test statistic (e.g. $F$, $t$, $r$) with confidence intervals, effect sizes, degrees of freedom and $P$ value noted *Give P values as exact values whenever suitable.* |
| ☒ | ☐ | For Bayesian analysis, information on the choice of priors and Markov chain Monte Carlo settings |
| ☒ | ☐ | For hierarchical and complex designs, identification of the appropriate level for tests and full reporting of outcomes |
| ☒ | ☐ | Estimates of effect sizes (e.g. Cohen's $d$, Pearson's $r$), indicating how they were calculated |

*Our web collection on statistics for biologists contains articles on many of the points above.*

## Software and code

Policy information about availability of computer code

| | |
|---|---|
| Data collection | Data were collected using Psychopphysics toolbox (Version 3) for Matlab and Matlab (2016b). |
| Data analysis | Data were analyzed using Matlab (Version 2022b), EEGLAB (Version 13_6_5b), R(4.2.2 (2022-10-31)), Rstudio (Version 2022.12.0+353 (2022.12.0+353)), unfold for matlab (Version 3) and Unfold for Julia (Version 0.4.1), lme4 package (Version 1.1-31), sjPlot package (Version 2.8.12), analysis code is available under https://github.com/froemero/Common_Neural_Choice_Signals_emerge_artifactually |

For manuscripts utilizing custom algorithms or software that are central to the research but not yet described in published literature, software must be made available to editors and reviewers. We strongly encourage code deposition in a community repository (e.g. GitHub). See the Nature Portfolio guidelines for submitting code & software for further information.

## Data

Policy information about availability of data

All manuscripts must include a data availability statement. This statement should provide the following information, where applicable:
- Accession codes, unique identifiers, or web links for publicly available datasets
- A description of any restrictions on data availability
- For clinical datasets or third party data, please ensure that the statement adheres to our policy

Data for Studies 1, 2 (only data used here), 3 and 4 are available through on different platforms with links provided through github under: https://github.com/froemero/Common_Neural_Choice_Signals_emerge_artifactually.

## Human research participants

Policy information about studies involving human research participants and Sex and Gender in Research.

| | |
|---|---|
| Reporting on sex and gender | We collected gender information and report it as follows: Study 1:The final sample consisted of 39 participants, (27 female). Study 2: The sample in study 2 comprised 39 participants recruited from Brown University and the general community. Participants (26 female, mean age 23.92, SD = 5.14), gave informed consent and received $10 or $15 per hour for their participation. We do not report gender for the reanalyzed datasets, but this information can be found in the original publications. Gender is recorded in the original source data, but not considered as a relevant factor for the present study. |
| Population characteristics | Study 1: The final sample consisted of 39 participants, (27 female) with a mean age of 20.84 years (SD = 3.90). Study 2: The sample in study 2 comprised 39 participants recruited from Brown University and the general community. Participants (26 female, mean age 23.92, SD = 5.14), gave informed consent and received $10 or $15 per hour for their participation. Health status information was not assessed. |
| Recruitment | Participants for study 1 and 2 were recruited from Brown University and the general community. Participants signed up in response to advertisements and through SONA (an experiment sign-up software). It is likely that participants signing up for EEG experiments are generally more active and interested in science. Since we are recruiting from a Psychology participant pool, we also have more female than male participants. Since we are not investigating individual differences and only study within subject effects, it is unlikely that any self-selection characteristics have impacted the results. |
| Ethics oversight | Study 1 & 2 were approved by Brown University's IRB. |

Note that full information on the approval of the study protocol must also be provided in the manuscript.

# Field-specific reporting

Please select the one below that is the best fit for your research. If you are not sure, read the appropriate sections before making your selection.

☐ Life sciences  ☒ Behavioural & social sciences  ☐ Ecological, evolutionary & environmental sciences

For a reference copy of the document with all sections, see nature.com/documents/nr-reporting-summary-flat.pdf

# Behavioural & social sciences study design

All studies must disclose on these points even when the disclosure is negative.

| | |
|---|---|
| Study description | The studies are quantitative experimental within-subject designs. |
| Research sample | See population characteristics above. Participants were recruited from Brown and the general community. The sample is not representative of the general community since it oversamples Brown undergrads and females in particular. Study 1: The final sample consisted of 39 participants, (27 female) with a mean age of 20.84 years (SD = 3.90). Study 2: The sample in study 2 comprised 39 participants recruited from Brown University and the general community. Participants (26 female, mean age 23.92, SD = 5.14), gave informed consent and received $10 or $15 per hour for their participation. Health status information was not assessed. |
| Sampling strategy | Participants were sampled pseudo randomly (they signed up). The sample sizes were based on previous studies in this line of work, i.e., 30 Shenhav & Karmarkar (2019). We oversampled relative to this study to assure sufficient observations for linear mixed effects models. |
| Data collection | Data were collected using a computerized experiment and EEG recording equipment. No people were present besides the participant and the experimenter(s). The participant performed the tasks on their own following intermittent instructions. The experimenters were not blind to the study hypotheses, but since choice sets were determines individually, they were blind to individual choice conditions. |
| Timing | Study 1 data collection: June 2017 through October 2017 Study 2 data collection: April 2022 through June 2022 |
| Data exclusions | For the main study 48 (Study 1) participants were recruited from Brown University and the general community. Of these 9 had to be excluded due to technical problems during data acquisition. |
| Non-participation | No participants dropped out. |
| Randomization | N/A this is not a between subject design |

# Reporting for specific materials, systems and methods

We require information from authors about some types of materials, experimental systems and methods used in many studies. Here, indicate whether each material, system or method listed is relevant to your study. If you are not sure if a list item applies to your research, read the appropriate section before selecting a response.

| Materials & experimental systems | | Methods | |
|---|---|---|---|
| **n/a** | **Involved in the study** | **n/a** | **Involved in the study** |
| ☒ ☐ | Antibodies | ☒ ☐ | ChIP-seq |
| ☒ ☐ | Eukaryotic cell lines | ☒ ☐ | Flow cytometry |
| ☒ ☐ | Palaeontology and archaeology | ☒ ☐ | MRI-based neuroimaging |
| ☒ ☐ | Animals and other organisms | | |
| ☒ ☐ | Clinical data | | |
| ☒ ☐ | Dual use research of concern | | |

