## [Peer Review File · Nature Human Behaviour]

Peer Review Information

Journal: Nature Human Behaviour

Manuscript Title: Common neural choice signals can emerge artifactually amidst multiple distinct value signals

Corresponding author name(s): Romy Frömer

Reviewer Comments & Decisions:

Decision Letter, initial version:

22nd May 2023

Dear Dr Froemer,

Thank you once again for your manuscript, entitled "Common neural choice signals emerge artifactually amidst multiple distinct value signals", and for your patience during the peer review process.

Your Article has now been evaluated by 3 referees. You will see from their comments copied below that, although they find your work of considerable potential interest, they have raised quite substantial concerns. In light of these comments, we cannot accept the manuscript for publication, but would be interested in considering a revised version if you are willing and able to fully address reviewer and editorial concerns.

We hope you will find the referees' comments useful as you decide how to proceed. If you wish to submit a substantially revised manuscript, please bear in mind that we will be reluctant to approach the referees again in the absence of major revisions. We are committed to providing a fair and constructive peer-review process. Do not hesitate to contact us if there are specific requests from the reviewers that you believe are technically impossible or unlikely to yield a meaningful outcome.

In particular, we feel that Reviewer #1's comments raise significant concerns, and that it will be necessary for you to carry out additional analysis of a perceptual task dataset with a longer mean RT as suggested by this Reviewer.

If you wish to submit a suitably revised manuscript, we would hope to receive it within 4 months. I would be grateful if you could contact us as soon as possible if you foresee difficulties with meeting this target resubmission date.

- Include a “Response to the editors and reviewers” document detailing, point-by-point, how you addressed each editor and referee comment. If no action was taken to address a point, you must provide a compelling argument. When formatting this document, please respond to each reviewer comment individually, including the full text of the reviewer comment verbatim followed by your response to the individual point. This response will be used by the editors to evaluate your revision and sent back to the reviewers along with the revised manuscript.
- Highlight all changes made to your manuscript or provide us with a version that tracks changes.

[REDACTED]

Thank you for the opportunity to review your work. Please do not hesitate to contact me if you have any questions or would like to discuss the required revisions further.

Sincerely,

[REDACTED]

REVIEWER COMMENTS:

Reviewer #1:

Remarks to the Author:

In this paper, Fromer and colleagues seek EEG signatures of value-based decisions. In the main task participants chose among a series of pairs of items which they had previously rated on a value scale. The EEG data were reduced using PCA which indicated two key components, one reflecting how positively the item set on a trial had been rated and another reflecting how difficult the choice was. The latter had a topography consistent with the centro-parietal positivity (CPP) a component that has been previously reported to exhibit evidence accumulation characteristics on a wide range of perceptual choice tasks. However, in this case, the CPP did not exhibit the characteristics that had been predicted by the authors – it peaked up to 500ms before response execution and had a larger pre-response build-up for easy compared to hard comparison trials. Further analyses of an independent value-based choice task and a visual search task in which stimulus- and response-locked activity were deconvolved again did not show evidence accumulation characteristics expected by the authors. The authors conclude from their findings that the build-to-peak properties previously reported for the CPP may be an artifact of overlapping stimulus and response-locked component. This is an interesting paper on an important issue. EEG signatures of value-based choices have not been as well characterised as those for perceptual choice. Furthermore, if the authors are correct in their claim that the CPP is artifactual then this would have very significant implications for the field as this signal has and is being used to draw inferences about neural decision processes in a lot of studies these days.

However, for the reasons outlined below, my current assessment is that the authors strong claims are not justified by the presented analysis.

1. The CPP has not been previously linked with purely value-based choices to my knowledge and so the fact that it does not predict them is interesting but not surprising – it is entirely plausible that the brain could rely on one type of signal for accumulating sensory evidence for perceptual choice and another for comparing internal value signals. Therefore it is the analysis of the visual search data that is really central to the authors' strong claim that the evidence accumulation characteristics previously reported for the CPP may be artifactual as the overwhelming majority of CPP studies have involved perceptual tasks. However I have several concerns regarding the target selection data. First, the choices on this task were made relatively quickly (circa 450ms based on checking the original paper – these details should be provided in this manuscript too). As a result, it is entirely possible that the CPP on this type of task is equally aligned to both stimulus and response if the delays associated with initial sensory coding roughly correspond to those associated with motor execution. If this is the case then the deconvolution algorithm may well be forced to divide the CPP across the two components. It would have been helpful to see the stimulus-locked components for this analysis but only the response-locked one is shown. Even here, Figure 6B (bottom) shows a signal modulation between -300 to 0 that is consistent with and only slightly smaller than that shown in the standard ERP analysis (Fig6B top). Were stats conducted to directly compare these effects across the standard and deconvolved signals? Second, the authors have overlooked a highly relevant paper: Steinemann et al (2018) applied a residual iteration decomposition (RIDE) algorithm to their CPP data (from a task involving much slower decisions) which also served to separate stimulus- and response-locked activity and they observed all of the evidence accumulation characteristics that have been reported for the CPP in the response-locked component. This paper seems to directly contradict the current findings and would need to be carefully considered and discussed. Given what a strong and impactful claim the authors are making here I think these concerns would warrant careful consideration by the authors. The authors could even consider repeating their analyses on a perceptual choice dataset involving slower decisions, many of which are likely to be freely available online.
2. In the case of the value-based choice data, have the authors considered the possibility that the CPP reflects the initial perceptual choices (object discrimination/detection) that are required in order for the value-comparisons to take place? If this is so then this would dramatically change the predictions that one would make regarding CPP activity patterns when aligned to value-based choice. For example, this would naturally explain why the CPP appears to peak long before response execution on this task when RTs were slower. If the CPP is specifically involved in evidence accumulation for perceptual and not value-based choices then this would be a really important and interesting observation.
3. In the Discussion the authors state "However, decision-making studies that have identified the CPP as a signature of evidence accumulation often only show response-locked activity; in cases where stimulus-locked activity was examined, including our present results, the expected latency effect was not found (Sun et al., 2017)." Are the authors referring specifically to studies that have used deconvolution approaches? If so, please see Steinemann et al (2018). If not, then I am confused as many CPP papers would present stimulus as well as response-locked waveforms with the expected patterns observable in both. Note, this is a minor comment in many respects but I list it as a major one only because the authors are making the claim that a pretty large body of research has erroneously ascribed evidence accumulation characteristics to the CPP.

Minor:

There is some ambiguity in the predictions that are made for the CPP in the Introduction and Results. The authors stated that the CPP has been seen to 'ramp towards a common peak' and yet in the Discussion they cite papers which have shown that the CPP's amplitude actually decreases alongside accuracy with increasing RT. This should all be clarified in the introduction as the authors lay out their hypotheses. In the original CPP studies a build-to-threshold pattern was observed but this was using continuous monitoring tasks in which the participant could not predict when the targets would appear. In more recent studies in which participants are presented with discrete trials and clear knowledge of the onset and offset of the physical evidence, the CPP's amplitude diminished with

increasing RT (Steinemann et al 2018 Nat Comms; Kelly et al 2021, Nat Hum Beh). Consequently, the CPP is also seen to have larger response-locked amplitudes on easier compared to difficult trials (as was observed here) The authors also refer to the prediction that the response-locked CPP should show 'increased activity' on more difficult trials. I think the authors are referring to the fact that its slope should be shallower on more difficult trials and this results in larger amplitudes at time-points further back in time from the response. This again could be clarified to avoid potential confusion.

Reviewer #2:

Remarks to the Author:

In the article titled 'Common neural choice signals emerge artifactually amidst multiple distinct value signals', Frömer and colleagues report the results from studies on evoked EEG activity during value-based decision making. Based on previous literature, the authors hypothesized that value-related EEG activity would either resemble an evidence accumulation process, or that appraisal and choice-related evidence-accumulation could be disentangled, so that two distinct processes would be observable in the EEG. Using various methods, the authors observed that appraisal- and choice-related EEG activity could be disentangled, but also that neither of these processes reflect an evidence-accumulation process. They conclude that 1) the interpretation of the CPP as a component reflecting evidence-accumulation need to be questioned and 2) that the role of evidence-accumulation itself in decision making needs to be researched further.

The methods used for the data analysis presented in the manuscript are sophisticated. A PCA was conducted to integrate subjective measures and choice behavior into predictors of appraisal and choice on a single-trial basis. Then, these measures were used to calculate a regression with EEG data in sensor*time-space to identify appraisal and choice related EG components using a cluster-based permutation approach. Being an EEG researcher myself, I would have had an easier time following along while reading if these steps were presented one after the other, instead of just presenting the ERP predictions (Figure 3). If the relevant clusters were identified on regression-coefficient level (by testing against zero I assume?), I would have liked the cluster-permutation results with the corresponding coefficients being presented in sensor*time-space as well.

In addition to this regression-based approach, the authors replicated the results using the unfold toolbox on the same data. In a next step the authors analyze the data for different RT bins and show that an overlap of stimulus- and response-related ERP components mimic an evidence accumulation signal for shorter RTs. The authors then replicate these findings using a separate, but similar dataset (similar regarding the RTs). Finally, they analyzed two more datasets from studies with shorter RTs. Here, they observed CPP activity suggesting evidence-accumulation. When conducting a deconvolution analysis using the unfold toolbox, however, this result pattern disappeared.

Overall, I think that the investigation of value-based decision making by means of the EEG is very promising, as the temporal resolution allows for exactly what the authors demonstrate: Disentangling automatic, bottom-up driven control processes, from choice behavior itself. The question of which role an information accumulation process plays in choice behavior is very interesting. Despite it is not being possible to answer this question with the research presented in this manuscript, the main message, that CPP activity might look like evidence accumulation, but maybe/probably is not, is still relevant. It is positive that the authors replicated their findings using different methods and datasets and I also appreciate that the author do not try to give the impression that all observations (and the resulting follow-up analyses) are in line with initial expectations. On the other side, this is also related to my main issue with the manuscript: To me, reading this felt more like being on the journey of data analysis together with the authors, rather than being guided through existing results. Sometimes, I had a hard time following the flow of information. I think the manuscript could be more

concise conveying the main message.

Some minor points:

Page 11, results: For the stimulus-locked, appraisal PC related cluster a p-value of .04 is reported. Can this be considered significant in a two-sided test if the p-values are computed as the percentile of permutation clusters larger than the observed clusters (c.f. page 31).

Page 25, discussion: "Across four datasets, we found..." Only two datasets/experiments are mentioned in the method section. The other two are Pisauro et al. 2017 and Frömer, Maier, & Abdel Rahman 2018? Please clarify.

Page 27, method section: Why did different participants receive different amounts of money for participating in study 2? (\$10 or \$15)

Page 29, method section: In the details for study 2 it is described that the items were presented alternating for a maximum duration of five seconds. How long were the items presented, i.e., how fast were they cycling?

Page 31, method section: The regression approach needs further clarification. What kind of regression was calculated (which model, regularization)? Were both identified PCs used in a single model or were separate models calculated for appraisal and choice-related processes?

Page 31, method section: "...cluster masses obtained from 1000 random permutation samples..." Did you just consider the largest cluster-mass of each permutation or the mass of every cluster? This affects the obtained H0-distribution.

Page 32, method section: The introduction of the PC loadings comes as a surprise. I think the PCA based dimensionality reduction of appraisal and choice related data should be described in the method section itself as well.

Page 32, method section (and maybe also page 16): It should be described, at least roughly, how the unfold toolbox works. I never used it and felt a bit lost.

Reviewer #3:

Remarks to the Author:

Review of Common neural choice signals emerge artifactually amidst multiple distinct value signals

The authors present interesting results suggesting that event-related signatures of choice and overall value in EEG can be mistaken for markers of evidence accumulation when not accounting for the different ERP components, as well as stimulus timing relative to response timing. I believe it is important work, as the field of "finding ERPs that look like evidence accumulators" requires more experimental and theoretical effort. I also like the approach by the authors to better understand the result of averaging / convolving ERPs together with different stimulus timings, as well as the approach by the authors to find the cognitive components in behavioural data using PCA.

However I believe there needs to be more detail about the authors' Methods in order to better convince knowledgeable readers as well as encourage reproducibility of their work. It is also not clear from the data and results that the authors have correctly refuted the prevailing theory of CPPs reflecting evidence accumulation.

Specific concerns:

The PCA regression is a clever and clear way to analyze the data. However some of the basic model and mathematical details are missing from the Results and Methods that would be necessary to replicate this work. It could also be more clear for technically minded readers. For instance:

-How are "Unchosen" and "Chosen" variables coded in the data, as reflected by Figure 2c.

-What was the dimension of the matrix that was the input of the PCA?
 -What exactly are the independent and dependent variables in the regressions? Experimental trials were the observations? The second paragraph on page 31 could be interpreted in multiple ways.
 -How are these regressions collapsed across participants to produce single cluster topographic maps? We would expect participants to have different topographies.

Study 2 is referenced in the Methods but does not clearly have any point of reference before this section in the Results.

Details about the deconvolution used (references to Ehinger & Dimigen, 2019; Smith & Kutas, 2015a, 2015b) in the Methods would be useful to readers. This seems like an accidental omission from the Methods section.

The response-locked components found look more like Readiness Potentials, thought be driven by the motor system. These EEG components are thought to encode choice and decision time in certain scenarios (see Gluth et al. 2013 and Lui et al. 2021). This does not necessarily refute the authors points, and establishing the difference between CPPs and RPs is often difficult. However it is literature the authors should consider, especially when critical of the previous CPP literature.

Page 17 second paragraph: "... most studies investigating evidence accumulation signals in EEG involved much faster decisions..." : The authors need to convince readers of this claim. What are some example studies? O'Connell et al. (2012) found longer response times even longer than the authors' current work. For instance, see Figure 3 of the paper by O'Connell et al. (2012).

References

Gluth, S., Rieskamp, J., & Büchel, C. (2013). Classic EEG motor potentials track the emergence of value-based decisions. *Neuroimage*, 79, 394-403.

Lui, K. K., Nunez, M. D., Cassidy, J. M., Vandekerckhove, J., Cramer, S. C., & Srinivasan, R. (2021). Timing of readiness potentials reflect a decision-making process in the human brain. *Computational Brain & Behavior*, 4, 264-283.

O'Connell, R. G., Dockree, P. M., & Kelly, S. P. (2012). A supramodal accumulation-to-bound signal that determines perceptual decisions in humans. *Nature neuroscience*, 15(12), 1729-1735.

Author Rebuttal to Initial comments

Reviewer #1:

Remarks to the Author:

In this paper, Fromer and colleagues seek EEG signatures of value-based decisions. In the main task participants chose among a series of pairs of items which they had previously rated on a value scale. The EEG data were reduced using PCA which indicated two key components, one reflecting how positively the item set on a trial had been rated and another reflecting how difficult the choice was. The latter had a topography consistent with

the centro-parietal positivity (CPP) a component that has been previously reported to exhibit evidence accumulation characteristics on a wide range of perceptual choice tasks. However, in this case, the CPP did not exhibit the characteristics that had been predicted by the authors – it peaked up to 500ms before response execution and had a larger pre-response build-up for easy compared to hard comparison trials. Further analyses of an independent value-based choice task and a visual search task in which stimulus- and response-locked activity were deconvolved again did not show evidence accumulation characteristics expected by the authors. The authors conclude from their findings that the build-to-peak properties previously reported for the CPP may be an artifact of overlapping stimulus and response-locked component

This is an interesting paper on an important issue. EEG signatures of value-based choices have not been as well characterised as those for perceptual choice. Furthermore, if the authors are correct in their claim that the CPP is artifactual then this would have very significant implications for the field as this signal has and is being used to draw inferences about neural decision processes in a lot of studies these days.

Thank you for this positive evaluation of our manuscript's topic and its potential impact.

However, for the reasons outlined below, my current assessment is that the authors strong claims are not justified by the presented analysis.

1. The CPP has not been previously linked with purely value-based choices to my knowledge and so the fact that it does not predict them is interesting but not surprising – it is entirely plausible that the brain could rely on one type of signal for accumulating sensory evidence for perceptual choice and another for comparing internal value signals. Therefore it is the analysis of the visual search data that is really central to the authors' strong claim that the evidence accumulation characteristics previously reported for the CPP may be artifactual as the overwhelming majority of CPP studies have involved perceptual tasks. However I have several concerns regarding the target selection data. First, the choices on this task were made relatively quickly (circa 450ms based on checking the original paper – these details should be provided in this manuscript too). As a result, it is entirely possible that the CPP on this type of task is equally aligned to both stimulus and response if the delays associated with initial sensory coding roughly correspond to those associated with motor execution. If this is the case then the deconvolution algorithm may well be forced to divide the CPP across the two components.

The CPP has in fact been linked to value-based choices. We are reanalyzing data from a highly-cited publication that argues for this link (Pisauro et al. 2017, >140 citations). We therefore think that evidence contradicting such a signal carries significant value on its own.

Nevertheless, we agree that it is important to know how specific the confound we have identified is to evidence accumulation signals in research on value-based versus perceptual decision-making. As you will see below, we have now supplemented our previous findings with analyses of two additional perceptual tasks, and find further evidence that our value-based decision-making findings (across 3 previous datasets) replicate in the domain of perceptual decision-making (now across a total of 3 datasets).

It would have been helpful to see the stimulus-locked components for this analysis but only the response-locked one is shown. Even here, Figure 6B (bottom) shows a signal modulation between -300 to 0 that is consistent with and only slightly smaller than that shown in the standard ERP analysis (Fig6B top). Were stats conducted to directly compare these effects across the standard and deconvolved signals?

Thank you for pointing out the lack of clarity in our reporting. We have indeed previously included statistics for the analysis of the perceptual decision-making dataset in the Supplementary material. However, to be consistent, we are now reporting cluster-based permutation test results - at Pz only - for *all* reanalyzed datasets in the figure (revised Fig. 6, see below) and show that there are no significant response-locked clusters when performing the deconvolution analysis. We are now also including stimulus-locked findings for all of these analyses in the Supplement for those interested (see below).

Second, the authors have overlooked a highly relevant paper: Steinemann et al (2018) applied a residual iteration decomposition (RIDE) algorithm to their CPP data (from a task involving much slower decisions) which also served to separate stimulus- and response-locked activity and they observed all of the evidence accumulation characteristics that have been reported for the CPP in the response-locked component. This paper seems to directly contradict the current findings and would need to be carefully considered and discussed. Given what a strong and impactful claim the authors are making here I think these concerns would warrant careful consideration by the authors. The authors could even consider repeating their analyses on a perceptual choice dataset involving slower decisions, many of which are likely to be freely available online.

We thank the reviewer for pointing out this oversight on our end and making us aware of this paper! After carefully reading the paper, our understanding is that RIDE was applied to remove the stimulus-evoked potential associated with an auditory evidence onset cue. This procedure did indeed not change the typical CPP pattern in that dataset, but there are some important considerations when interpreting the conclusions drawn from these analyses:

1. *Objective evidence onset, even if cued is not the same as subjective evidence onset.* While the auditory evidence onset cue did alert participants to the onset of the visual evidence, that does not necessarily imply that the onset of the *subjective percept* of the visual evidence onset is temporally locked to the tone onset.
2. *RIDE assumes that the latency of the stimulus component is constant - the perception*

of evidence onset most likely is not. While the latency of activity evoked by the tone satisfies the RIDE assumption, the subjective onset of the evidence and hence its associated activity is likely delayed and significantly more variable for low compared to high contrast. Any such variability would result in the associated activity to be separated into the central RIDE component rather than the stimulus one.

3. *Not stimulus evoked does not imply not stimulus-locked.* There is an important distinction between stimulus-evoked and stimulus-locked, in that the latter can be considered a matter of degree whereas the former cannot. The P3 component has not generally been considered to be stimulus-evoked, but it can still be more closely aligned with the stimulus or more closely aligned with the response. In the datasets at hand, we find that the component is more closely aligned with the stimulus.

To address this issue, we obtained Steinemann et al.'s data and reanalyzed it in the same way we did the Pisuro data (Figure 6C, reproduced below). When analyzing their response-locked data without accounting for a stimulus-locked component, we replicate their finding of a CPP (top panel). After deconvolution (i.e., accounting for the stimulus-locked component), we no longer find any significant clusters indicative of a CPP. To confirm these findings, we re-analyzed a third perceptual decision-making dataset obtained from Boldt et al (2019, SciRep), which also has longer RTs than Steinemann et al.'s visual search task, but a clear stimulus onset. We again find that the pattern associated with evidence accumulation vanishes.

Note that to give the CPP the best possible shot of appearing, with these two new re-analyses we took the additional conservative step of not allowing stimulus-locked events to vary with RT. That is, by merely accounting for the *occurrence* of the stimulus event, we eliminate the response-locked effects.

We now include all of these results in the manuscript:

Figure 6. Response-locked evidence accumulation patterns vanish when correcting for component overlap. **A.** Regression ERPs (rERPs) from a mass-univariate re-analysis (top) of an independent value-based decision-making dataset (Pisauro, Fouragnan, Retzler, & Philiastides, 2017, cf Fig. 5A) exhibit a CPP with characteristic signatures of evidence accumulation. Decisions based on weaker evidence (slow compared to fast) are associated with a slower response-locked ramping of CPP amplitude, resulting in larger CPP amplitudes prior to the response. When re-analyzing these data with a deconvolution approach that models both stimulus-related and response-related activity (bottom), we find that response-locked rERPs no longer show the characteristic evidence accumulation pattern. **B - C.** Regression ERPs (rERPs) from a mass-univariate analysis (top) of three independent perceptual decision-making datasets exhibit a CPP with characteristic signatures of evidence accumulation. Decisions based on weaker evidence (blurred stimuli/slower responses) are associated with slower response-locked ramping of CPP amplitude, resulting in larger CPP amplitudes prior to the response. When re-analyzing these data with a deconvolution approach that models both stimulus-related and response-related activity (bottom), we find that response-locked rERPs no longer show the characteristic evidence accumulation pattern. Data are shown with average reference. **B.** Data from Frömer, Maier, & Abdel Rahman (2018). **C.** Data from Steinemann, O'Connell, & Kelly (2018). **D.** Data from Boldt, Schiffer, Waszak, & Yeung (2019). **A.-D.** Grey bars indicate temporal clusters significant at $p < .05$, cluster permutation-corrected for multiple comparisons.

Supplementary Figure 3. Stimulus-locked activity in MASS-univariate analyses compared to unfold deconvolution. Stimulus-locked activity across all tasks exhibits typical ERP responses, including a marked P3 component, peaking between 400 and 600 ms, regardless of response-times. This canonical ERP sequence is only marginally affected by deconvolution of response-locked activity, and is also observed in our Study 1 where there is no substantial overlap with responses to begin with, and further typically observed in responses to feedback, which is not overlapping with any immediate responses at all (e.g., Frömer et al., 2021). **A.** Data from Pisauro, Fouragnan, Retzler, & Philiastides (2017). **B.** Data from Frömer, Maier, & Abdel Rahman (2018). **C.** Data from Steinemann, O'Connell, & Kelly (2018). **D.** Data from Boldt, Schiffer, Waszak, & Yeung (2019).

p 19: “To provide a more direct test of our hypothesis that the evidence accumulation effects in the CPP could originate from a component overlap artifact, we re-analyzed EEG findings from **four** previous datasets in which response times were more tightly constrained, one collected during value-based decision-making (Study 3; PISAURO et al., 2017) and **the other three collected during perceptual decision-making (Boldt, Schiffer, Waszak, & Yeung, 2019; Frömer, Maier, & Abdel Rahman, 2018; Steinemann, O'Connell, & Kelly, 2018)**. In the value-based decision-making study, participants (N = 21) chose among pairs of snack items while their EEG was recorded and had to respond within 1.25s from stimulus onset.

The authors found that behavior was well-captured with a DDM and reported the typical response-locked CPP evidence accumulation signal (cf. Fig. 4A top). In the **first** perceptual decision-making study (Study 4; Frömer et al., 2018), EEG was recorded while participants (N = 40) decided whether a deviant object in a circular array of objects was on the left or right side of the display (Fig 6). Objects in the array were chosen to be visually similar and presented either intact or blurred, which serves as an index of evidence strength and modulated performance accordingly (lower accuracy and slower RTs for blurred compared to intact stimuli). Stimuli were presented for 200 ms, and participants had up to 2 s from stimulus onset to respond. **In the second perceptual decision-making study (Study 5; Steinemann et al., 2018) EEG was recorded while participants (N = 17) determined which of two overlaid gratings had a higher contrast. Unlike the other studies, stimuli were not suddenly presented, but faded in at equal contrast and switched to the target contrasts after a random time interval simultaneously with a tone signaling the change. The evidence was then presented for 2400 ms within which participants had to respond. In the third perceptual decision-making dataset (Study 6; Boldt et al., 2019), EEG was recorded while participants (N = 17) judged the overall blueness versus redness of compound stimuli that varied in the blue, red and purple hue of their components.**

Stimuli were presented for 160ms and participants had 1200ms to decide.

When we separately reanalyze stimulus-locked and response-locked activity using mass-univariate analyses analogous to standard ERP analyses, we find the characteristic CPP indices of evidence accumulation over centroparietal sites in all four studies. Response-locked analyses revealed that activity peaked at the time of the response and rose with a steeper slope for fast relative to slow trials in value-based decision-making (Fig. 6A, top) and intact relative to blurred stimuli (Fig. 6B, top) **and fast relative to slow trials (Fig. 6C-D, top)** in perceptual decision-making, resulting in more positive CPP amplitudes for trials that putatively required more evidence accumulation, respectively (cf. Fig. 1A). The centroparietal positivity in both studies therefore meets response-locked criteria for signatures of evidence accumulation. However, because these analyses do not explicitly account for the overlap between stimulus-related and response-related components, they cannot distinguish whether the response-locked patterns reflect evidence accumulation or stimulus-related activity. To formally disentangle these, we again applied the deconvolution approach introduced earlier (Ehinger & Dimigen, 2019), including stimulus and response events in a

single model of neural responses. After deconvolution, we no longer find response-locked signatures of evidence accumulation in **any of the datasets** (Fig. 6A-D, bottom), suggesting that this characteristic pattern of evidence accumulation only appeared in our previous analysis as an artifact of component overlap.”

We discuss the Steinemann et al study and the results as follows:

p.26: “This is particularly notable since in their original analyses Steinemann et al. (2018) had used a different algorithm (RIDE; Ouyang, Herzmann, Zhou, & Sommer, 2011) to subtract stimulus-locked activity associated with the auditory evidence onset cue and found the typical CPP patterns intact. This method separates early stimulus-locked components that are tightly temporally coupled from later activity that is not closely temporally coupled to the response, a central component. However, merely subtracting the stimulus-locked component determined this way does not offer information as to whether this central component is more aligned with the stimulus or the response, nor will it account for this component in response-locked analyses of the remaining data. If this central component occurred somewhat aligned to the stimulus, albeit more loosely than the early stimulus-locked activity, as a transient ERP, analyzing it locked to the response would still result in the same component overlap effect identified in this paper, and under certain conditions give the appearance of a ramp. Our findings suggest that the central component preserved in Steinemann et al. (2018) is indeed locked to the stimulus as a canonical P3 ERP. This component is commonly observed for most behaviorally relevant visual stimuli, including feedback, regardless of response times, and is thought to potentially reflect a form of monitoring (Ullsperger, Fischer, Nigbur, & Endrass, 2014).“

2. In the case of the value-based choice data, have the authors considered the possibility that the CPP reflects the initial perceptual choices (object discrimination/detection) that are required in order for the value-comparisons to take place? If this is so then this would dramatically change the predictions that one would make regarding CPP activity patterns when aligned to value-based choice. For example, this would naturally explain why the CPP appears to peak long before response execution on this task when RTs were slower. If the CPP is specifically involved in evidence accumulation for perceptual and not value-based choices then this would be a really important and interesting observation.

This is a really great suggestion, and in fact something we briefly speculated about in a previous review article (Frömer & Shenhav, 2022): “If familiarity with a small total set of options allows the choice to be reduced to a perceptual one or a simple recognition task, researchers may in some of these cases have identified correlates of choice value that really reflect signatures of perceptual decision-making (Pisauro et al., 2017)...” . We agree that this would be really important to distinguish, and was further motivation for the additional analyses reported in the revised manuscript. However, we do not as of yet see clear evidence that this is the case - even in perceptual decisions, the CPP did not hold in the new

analysis. We make this reasoning explicit now:

p. 25/26 “This was true across both value-based and perceptual decision-making tasks. Considering the value-based decision-making studies only, one may have hypothesized that there is no CPP signature of value-based evidence accumulation, because the CPP is specific to perceptual evidence accumulation that precedes and is a necessary precursor to value-based evidence accumulation (cf. Frömer & Shenhav, 2022), perhaps implemented through subcortical circuits. Yet, the collective findings of our reanalysis of three perceptual decision-making datasets speak against this hypothesis.”

3. In the Discussion the authors state “However, decision-making studies that have identified the CPP as a signature of evidence accumulation often only show response-locked activity; in cases where stimulus-locked activity was examined, including our present results, the expected latency effect was not found (Sun et al., 2017).” Are the authors referring specifically to studies that have used deconvolution approaches? If so, please see Steinemann et al (2018). If not, then I am confused as many CPP papers would present stimulus as well as response-locked waveforms with the expected patterns observable in both. Note, this is a minor comment in many respects but I list it as a major one only because the authors are making the claim that a pretty large body of research has erroneously ascribed evidence accumulation characteristics to the CPP.

The reviewer is correct. This was written in the context of a longer line of reasoning, where the stimulus-locked effect was originally shown and typically seen in tasks where the decision-onset was obscured. In other tasks, where this was not the case, the stimulus-locked effect was typically not reported or if so, it was not always found. However, we do acknowledge that we missed the Steineman article and that this nuance was lost in the final version of the manuscript.

We have revised this section accordingly as follows:

“However, **other** decision-making studies that have identified the CPP as a signature of evidence accumulation only show response-locked activity (Boldt et al., 2019; PISAURO et al., 2017); **and in some cases** where stimulus-locked activity was examined, including our present results, the expected latency effect was not found (Sun et al., 2017).”

Minor:

There is some ambiguity in the predictions that are made for the CPP in the Introduction and Results. The authors stated that the CPP has been seen to ‘ramp towards a common peak’ and yet in the Discussion they cite papers which have shown that the CPP’s amplitude actually decreases alongside accuracy with increasing RT. This should all be clarified in the introduction as the authors lay out their hypotheses. In the original CPP studies a build-to-threshold pattern was observed but this was using continuous monitoring tasks in which the participant could not predict when the targets would appear. In more recent studies in

which participants are presented with discrete trials and clear knowledge of the onset and offset of the physical evidence, the CPP's amplitude diminished with increasing RT (Steinemann et al 2018 Nat Comms; Kelly et al 2021, Nat Hum Beh). Consequently, the CPP is also seen to have larger response-locked amplitudes on easier compared to difficult trials (as was observed here) The authors also refer to the prediction that the response-locked CPP should show 'increased activity' on more difficult trials. I think the authors are referring to the fact that its slope should be shallower on more difficult trials and this results in larger amplitudes at time-points further back in time from the response. This again could be clarified to avoid potential confusion.

Thank you for your helpful pointers. We have now clarified both in the introduction. Please note that the time window in which the amplitude is expected to differ between easy and difficult trials is very different from where we found it in our study. What would be expected is a crossover, pretty much exactly like is seen in Pisauro's data and our visual search data (see Fig. 6 above). The higher amplitude for easier trials, consistent with a higher bound at the time of the response should occur at the time of the response, not 1s to 500ms prior to the response. We now clarify all the above in the introduction as follows:

p. 5 "Researchers have shown that the CPP demonstrates three characteristic elements of evidence accumulation (cf. Fig 1A): (1) following stimulus presentation, activity is greater and peaks earlier when decision-related evidence is stronger (consistent with a more rapid rise of evidence accumulation when choices are easier), (2) activity peaks around the time of the response (consistent with a common response threshold), and (3) in the period leading up to the response, **due to the slower accumulation and thus shallower slope**, activity is greater when evidence is weaker and/or responses are slower. **The latter effect is sometimes part of a cross-over pattern and complemented by an opposite effect at the time of response, reflecting perhaps a decrease in decision-threshold for longer RTs through an urgency signal or a modulation of the overlapping readiness potential (Gluth, Rieskamp, & Buchel, 2013; Kelly & O'Connell, 2013; Lui et al., 2021), specifically in paradigms where there is a clear on- and offset of the physical evidence (Kelly, Corbett, & O'Connell, 2021; Steinemann, O'Connell, & Kelly, 2018).** The CPP is thus a potential index of value-based processing that is integral to decision-making per se."

Reviewer #2:

Remarks to the Author:

In the article titled 'Common neural choice signals emerge artifactually amidst multiple distinct value signals', Frömer and colleagues report the results from studies on evoked EEG activity during value-based decision making. Based on previous literature, the authors hypothesized that value-related EEG activity would either resemble an evidence accumulation process, or that appraisal and choice-related evidence-accumulation could be

disentangled, so that two distinct processes would be observable in the EEG. Using various methods, the authors observed that appraisal- and choice-related EEG activity could be disentangled, but also that neither of these processes reflect an evidence-accumulation process. They conclude that 1) the interpretation of the CPP as a component reflecting evidence-accumulation need to be questioned and 2) that the role of evidence-accumulation itself in decision making needs to be researched further.

The methods used for the data analysis presented in the manuscript are sophisticated. A PCA was conducted to integrate subjective measures and choice behavior into predictors of appraisal and choice on a single-trial basis. Then, these measures were used to calculate a regression with EEG data in sensor*time-space to identify appraisal and choice related EG components using a cluster-based permutation approach. Being an EEG researcher myself, I would have had an easier time following along while reading if these steps were presented one after the other, instead of just presenting the ERP predictions (Figure 3). If the relevant clusters were identified on regression-coefficient level (by testing against zero I assume?), I would have liked the cluster-permutation results with the corresponding coefficients being presented in sensor*time-space as well.

We thank the reviewer for this positive assessment of our approach!

We have now clarified the approach in the manuscript:

“We regressed stimulus and response-locked single-trial EEG activity for each participant at each sensor and each time point onto these appraisal- and choice-related PCs (cf. Fig. 1 B), and identified significant stimulus- and response-related clusters associated with each PC using cluster-based permutation tests on the resulting t-statistics at the group level. We found that the PCs mapped onto distinct spatiotemporal patterns (Fig. 3).”

Figure 3 was actually already showing the sensor*time-space coefficients, albeit in 100ms averages and on topographies because we felt that this was easier to read for people not using this analysis approach but standard ERPs or other types of brain signals. We have now added a more standard supplementary figure for this type of analysis in the Supplement.

Supplementary Figure 1. Time-Sensor plot of cluster permutation test results. A-B. Columns show Stimulus- (left) and Response-locked (right) results, respectively. Rows show results for Appraisal PC (top) and Choice PC (bottom) respectively. Plotted in each subplot are t-values at each time-point and sensor masked by cluster significance. Positive clusters show positive values (yellow) and negative clusters negative values (blue). **A.** Mass-univariate analysis results. **B.** Unfold-Deconvolution results.

In addition to this regression-based approach, the authors replicated the results using the unfold toolbox on the same data. In a next step the authors analyze the data for different RT

bins and show that an overlap of stimulus- and response-related ERP components mimic an evidence accumulation signal for shorter RTs. The authors then replicate these findings using a separate, but similar dataset (similar regarding the RTs). Finally, they analyzed two more datasets from studies with shorter RTs. Here, they observed CPP activity suggesting evidence-accumulation. When conducting a deconvolution analysis using the unfold toolbox, however, this result pattern disappeared.

Overall, I think that the investigation of value-based decision making by means of the EEG is very promising, as the temporal resolution allows for exactly what the authors demonstrate: Disentangling automatic, bottom-up driven control processes, from choice behavior itself. The question of which role an information accumulation process plays in choice behavior is very interesting. Despite it is not being possible to answer this question with the research presented in this manuscript, the main message, that CPP activity might look like evidence accumulation, but maybe/probably is not, is still relevant. It is positive that the authors replicated their findings using different methods and datasets and I also appreciate that the author do not try to give the impression that all observations (and the resulting follow-up analyses) are in line with initial expectations. On the other side, this is also related to my main issue with the manuscript: To me, reading this felt more like being on the journey of data analysis together with the authors, rather than being guided through existing results. Sometimes, I had a hard time following the flow of information. I think the manuscript could be more concise conveying the main message.

Thank you for the very positive assessment of our paper. We have now tried to reduce detours where possible while staying true to what we had originally predicted and what we found incidentally.

Some minor points:

Page 11, results: For the stimulus-locked, appraisal PC related cluster a p-value of .04 is reported. Can this be considered significant in a two-sided test if the p-values are computed as the percentile of of permutation clusters larger than the observed clusters (c.f. page 31).

Thank you, these are in fact two-sided p-values. We computed them by multiplying the observed percentile by 2 and we only obtain these for clusters that surpassed the 2.5% cluster mass threshold to begin with. The description in the method incorrectly omits the multiplication part. We are now clarifying how these p-values were computed in the method and refer the reader in the first instance we report these.

p. 35: "Clusters with masses (summed absolute t-values) larger than 2.5 % of cluster masses obtained from 1000 random permutation samples were considered significant. **Observed cluster mass was compared to a permutation distribution to get a percentile rank, which was inverted (such that lower numbers corresponded to more unlikely events), divided by 100 (to convert the percentile to a decimal), and multiplied by 2 to obtain a p-value for a two-sided test.** P-values were computed as the percentile of permutation clusters larger than the observed clusters **and multiplied by 2 to obtain two-sided p-values.**"

p. 11: "In line with our predictions, we found that our Appraisal PC explained EEG activity locked to (and following) stimulus onset (Fig. 2A, $p = .040$, two-sided cluster permutation

test, cf. **Method**), but not locked to the response (neither preceding nor following).”

Page 25, discussion: “Across four datasets, we found...” Only two datasets/experiments are mentioned in the method section. The other two are PISAURO et al. 2017 and Frömer, Maier, & Abdel Rahman 2018? Please clarify.

Thank you for pointing this out. We will have now added information on the two previously published datasets in the Method as well.

p. 28 Participants: “Please see the original publications for detailed participant information in Study 3 (N = 21; PISAURO et al., 2017), Study 4 (N = 40; Frömer et al., 2018), Study 5 (N = 17; Steinemann, O’Connell, & Kelly, 2018), and Study 6 (N= 17 after exclusion of 3 participants with non-matching behavioral and EEG data; Boldt et al., 2019).”

p. 31 Task and Procedure: “In Study 3 (PISAURO et al., 2017), participants chose among pairs of previously rated snack items and had to respond within 1.25s from stimulus onset. The difficulty of trials was manipulated by varying the value difference between options across 4 levels (1 through 4).

In Study 4 (Frömer et al., 2018), participants decided whether a deviant object in a circular array of objects was on the left or right side of the display. Objects in the array were chosen to be visually similar and presented either intact or blurred, which serves as an index of evidence strength and modulated performance accordingly (lower accuracy and slower RTs for blurred compared to intact stimuli). Stimuli were presented for 200 ms, and participants had up to 2 s from stimulus onset to respond.

In Study 5 (Steinemann et al., 2018), participants performed a contrast discrimination task in which they decided which of two overlaid grating patterns had a higher contrast. Participants were cued to either emphasize accuracy or speed and were rewarded when their responses met the current condition and punished when they did not. A trial began with a regimen cue, followed by neutral stimuli with equal contrast, which changed to the target stimulus alongside a tone to signal evidence onset and stayed on screen for 2400 ms. Participants received feedback following their response.

In Study 6 (Boldt et al., 2019), participants judged whether stimuli, each consistent of a circular array of 8 red, blue and purple circles, were more blue or more red on average. Across trials, the colored circles varied in their color strength (e.g., how clearly the stimulus was blue or red rather than purple) and in their variance (e.g., how variable the hues of the circles was). Stimuli were presented for 160 ms and participants had 1200 ms to respond. Participants were cued to one of four conditions (high mean – low variance, low mean – low variance, high mean – high variance, low mean – high variance). They also provided prospective and retrospective confidence judgments.”

p. 33 Psychophysiological Recording and Preprocessing: “Please see the original publications for detailed information on EEG data acquisition in Study 3 (PISAURO et al., 2017), Study 4 (Frömer et al., 2018), Study 5 (Steinemann et al., 2018), and Study 6 (Boldt et al., 2019). We obtained preprocessed data for Studies 3 and 4. Raw data for Study 5 were concatenated across blocks, re-referenced and low-pass filtered at 40Hz. A subset of blocks (18 total from 298 blocks across all participants) could not be matched with the behavioral data and were excluded from further analyses. We obtained data for Study 6 with ocular artifact reduction and downsampled to 250Hz and removed horizontal and vertical EEG channels as well as M2. Data were then matched with behavioral data and low-pass filtered at 40Hz.”

p. 35/36 Analyses: “**Study 4 perceptual decision-making data (Frömer et al., 2018)** were analyzed using the same procedures as just described, except that we used +/-1 second time-windows due to the faster pace of the task, and additionally computed mass-univariate betas without overlap correction for comparison. The value-based decision-making data **in Study 3 (Pisauro et al., 2017), as well as perceptual decision-making data in Studies 5 (Steinemann et al., 2018) and 6 (Boldt et al., 2019)** were re-analyzed analogously using the Julia implementation of the unfold toolbox (Ehinger et al., 2023).”

Page 27, method section: Why did different participants receive different amounts of money for participating in study 2? (\$10 or \$15)

We apologize for not clarifying this. The difference has to do with a change to our IRB protocol over the course of data collection for this study, increasing the hourly participant compensation. We now clarify this as follows: “Participants (26 female, mean age 23.92, SD = 5.14), gave informed consent and received \$10 or \$15 per hour for their participation (**the hourly rate as per IRB was increased after the study commenced and participants’ payments were adjusted accordingly moving forward**).”

Page 29, method section: In the details for study 2 it is described that the items were presented alternating for a maximum duration of five seconds. How long were the items presented, i.e., how fast were they cycling?

We have now added more information on the design.

“During the choice phase, rather than all at once, items were presented alternating individually at the center of the screen, until a decision was made, or 5 seconds elapsed while the duration of each item presentation was varied. **One item was always presented longer on average than the other. Specific presentation durations on each turn were drawn from different distributions for long (M = 500 ms, SD = 100 ms) and short presentations (M = 200, SD = 50); these distributions were informed by previous work (Frömer & Shenhav, 2019).** Response buttons corresponding to each item were coded via blue and red colored frames around the options and manipulated independently of the order **and duration** of item presentation.”

Page 31, method section: The regression approach needs further clarification. What kind of regression was calculated (which model, regularization)? Were both identified PCs used in a single model or were separate models calculated for appraisal and choice-related processes?

We now expand this information in the Method: “EEG data were analyzed using a mass-univariate approach employing custom made Matlab scripts adapted from Collins and Frank (2016, 2018): For each subject, voltages at each electrode and time point (downsampled to 250 Hz) **as dependent variables** were regressed (**using MATLAB’s regress function which implements least squares regression**) against trial parameters and an intercept term as **independent variables** to obtain regression weights for each predictor (similar to difference wave ERPs for each condition in traditional approaches, cf.: Smith & Kutas, 2015a).”

Note that we already reported previously what went into each regression (now clarifying that these are independent variables, highlighted in bold): “In the main analyses, the PC loadings for Appraisal and Choice PCs were included **as independent variables** with the intercept term. In three control analyses with the sets of variables underlying the PCs we entered **as independent** variables either overall value and value difference, Chosen and Unchosen Value, or Liking, Confidence and Anxiety alongside the intercept term.”

Page 31, method section: “...cluster masses obtained from 1000 random permutation samples...” Did you just consider the largest cluster-mass of each permutation or the mass of every cluster? This affects the obtained H0-distribution.

We now clarify that we got the largest cluster mass for each simulation as follows: “Clusters with cluster masses (summed absolute t-values) larger than 0.25 % of **the maximum** cluster masses obtained from 1000 random permutation samples were considered significant.”

Page 32, method section: The introduction of the PC loadings comes as a surprise. I think the PCA based dimensionality reduction of appraisal and choice related data should be described in the method section itself as well.

We now describe the PCA in the methods section as follows: “Appraisal and Choice principal components were derived from principal component analysis of all participants data on all trials on which a choice was made in time (4637/4680 trials) with the following variables normalized to 0-1 ranges: chosen value (value of the item that was ultimately chosen), unchosen value (value of the item that was ultimately chosen), value difference (max value minus min value), overall value (average of both values), set salience (absolute mean centered overall value), anxiety, liking, and confidence. Thus, the dimensions of the input matrix were 4637 x 8. An initial exploratory pca with permutation testing identified 2 principal components with eigenvalues greater than the 95th percentile than the distribution from shuffled data. We thus derived 2 principal components using matlab’s pca function, rotated the factors (normalized varimax), and derived trial-wise scores by multiplying the trial indicators with the factor loadings. We thus reduced the variables above to one score for Appraisal and one for Choice for each trial.”

Page 32, method section (and maybe also page 16): It should be described, at least roughly, how the unfold toolbox works. I never used it and felt a bit lost.

We now include a brief description of the unfold toolbox:

“For the deconvolution analyses, we conducted first level analyses on preprocessed data using the unfold toolbox (Ehinger & Dimigen, 2019). **This MatLab/Julia toolbox implements multiple regression with combined linear deconvolution for multivariate time-series similar to FIR-GLM analyses in fMRI. That is, by providing event-timings and per-event regression formulas, it allows us to disentangle temporally overlapping ERP responses (for detailed introduction to the method, see Ehinger & Dimigen 2019 or Smith & Kutas 2015b).**”

Reviewer #3:

Remarks to the Author:

Review of Common neural choice signals emerge artifactually amidst multiple distinct value signals

The authors present interesting results suggesting that event-related signatures of choice and overall value in EEG can be mistaken for markers of evidence accumulation when not accounting for the different ERP components, as well as stimulus timing relative to response timing. I believe it is important work, as the field of “finding ERPs that look like evidence accumulators” requires more experimental and theoretical effort. I also like the approach by the authors to better understand the result of averaging / convolving ERPs together with different stimulus timings, as well as the approach by the authors to find the cognitive components in behavioural data using PCA.

We thank the reviewer for their positive assessment of our work.

However I believe there needs to be more detail about the authors’ Methods in order to better convince knowledgeable readers as well as encourage reproducibility of their work. It is also not clear from the data and results that the authors have correctly refuted the prevailing theory of CPPs reflecting evidence accumulation.

Specific concerns:

The PCA regression is a clever and clear way to analyze the data. However some of the basic model and mathematical details are missing from the Results and Methods that would be necessary to replicate this work. It could also be more clear for technically minded readers. For instance:

- How are “Unchosen” and “Chosen” variables coded in the data, as reflected by Figure 2c.
- What was the dimension of the matrix that was the input of the PCA?

Thank you, we have now expanded the method section to include these details as follows: p. 33/34: “Appraisal and Choice principal components were derived from principal component analysis of all participants data on all trials on which a choice was made in time (4637/4680 trials) with the following variables normalized to 0-1 ranges: chosen value (value of the item that was ultimately chosen), unchosen value (value of the item that was ultimately chosen), value difference (max value minus min value), overall value (average of both values), set salience (absolute mean centered overall value), anxiety, liking, and confidence. Thus, the dimensions of the input matrix were 4637 x 8. An initial exploratory pca with permutation testing identified 2 principal components with eigenvalues greater than the 95th percentile than the distribution from shuffled data. We thus derived 2 principal components using matlab’s pca function, rotated the factors (normalized varimax), and derived trial-wise scores by multiplying the trial indicators with the factor loadings. We thus reduced the variables above to one score for Appraisal and one for Choice for each trial.”

- What exactly are the independent and dependent variables in the regressions? Experimental trials were the observations? The second paragraph on page 31 could be interpreted in multiple ways.

p. 34: “EEG data were analyzed using a mass-univariate approach employing custom made Matlab scripts adapted from Collins and Frank (2016, 2018): For each subject, voltages at each electrode and time point (downsampled to 250 Hz) as **dependent variables** were regressed (**using MATLAB’s regress function which implements least squares regression**) against trial parameters and an intercept term as **independent variables** to obtain regression weights for each predictor (similar to difference wave ERPs for each condition in traditional approaches, cf.: Smith & Kutas, 2015a).”

-How are these regressions collapsed across participants to produce single cluster topographic maps? We would expect participants to have different topographies. “These regression weights were weighted by transforming them into t-values (dividing them by their standard error), effectively biasing unreliable estimates towards zero, and then submitted to group-level cluster-based permutation tests, employing a cluster forming threshold of $p = 0.005$.”

Study 2 is referenced in the Methods but does not clearly have any point of reference before this section in the Results.

Thank you for pointing out this omission, we now reference the study as study 2 when we introduce the results:

p. 19: “To test whether these findings replicate in an independent sample, we examined the CPP in a separate value-based decision-making dataset (**Study 2**; $N = 39$) and found the same pattern: At odds with typical findings, CPP amplitudes in the pre-response time window (-700 to -200 ms as in Pisauro et al., 2017) were again significantly larger for shorter relative to longer RTs (LMM fixed effect: $b = -1.01$ $t = -6.58$, $p < .001$), and for easier than harder choice trials (LMM fixed effect: $b = 0.87$, $t = 2.32$, $p = .027$). “

Details about the deconvolution used (references to Ehinger & Dimigen, 2019; Smith & Kutas, 2015a, 2015b) in the Methods would be useful to readers. This seems like an accidental omission from the Methods section.

Thank you, we now include these additional details in the method (bold):

“For the deconvolution analyses, we conducted first level analyses on preprocessed data using the unfold toolbox (Ehinger & Dimigen, 2019). **This MatLab/Julia toolbox implements multiple regression with combined linear deconvolution for multivariate time-series similar to FIR-GLM analyses in fMRI. That is, by providing event-timings and per-event regression formulas, it allows us to disentangle temporally overlapping ERP responses (for detailed introduction to the method, see Ehinger & Dimigen 2019 or Smith & Kutas 2015b).** Stimulus onsets and responses were modeled simultaneously with the same regressors as in the main analyses. Deconvolution was implemented using FIR/stick basis functions, time expanded +/- 2 seconds around the respective events. Artifacts (amplitudes exceeding +/- 250 μ V) were detected and removed using the built-in threshold functions. No baseline corrections were applied. The obtained betas were submitted to the same cluster-based permutation analyses for second level analyses as described above.”

The response-locked components found look more like Readiness Potentials, thought to be driven by the motor system. These EEG components are thought to encode choice and decision time in certain scenarios (see Gluth et al. 2013 and Lui et al. 2021). This does not necessarily refute the authors points, and establishing the difference between CPPs and RPs is often difficult. However it is literature the authors should consider, especially when critical of the previous CPP literature.

Thank you, we now provide a nod to that literature on p. 5:

“The latter effect is sometimes part of a cross-over pattern and complemented by an opposite effect at the time of response, reflecting perhaps a decrease in decision-threshold for longer RTs through an urgency signal or a modulation of the overlapping readiness potential (Gluth, Rieskamp, & Büchel, 2013; Kelly & O’Connell, 2013; Lui et al., 2021), specifically in paradigms where there is a clear on- and offset of the physical evidence (Kelly, Corbett, & O’Connell, 2021; Steinemann, O’Connell, & Kelly, 2018).”

Page 17 second paragraph: “... most studies investigating evidence accumulation signals in EEG involved much faster decisions...” : The authors need to convince readers of this claim. What are some example studies? O’Connell et al. (2012) found longer response times even longer than the authors’ current work. For instance, see Figure 3 of the paper by O’Connell et al. (2012).

Thank you for pointing out our lack of citations to support this claim, we have added some accordingly:

“As we indicated earlier, most studies investigating evidence accumulation signals in EEG involved much faster decisions (e.g., **~800ms on average in Kelly & O’Connell, 2013; ~1s on average in O’Connell et al., 2012 except for a perturbation condition with additional stimulus dynamics; ~750ms on average in Pisauro et al., 2017; cf. Fig 5A).**”

Please note that except for the condition with the perturbation, where there’s a change in the stimulus, modal response times in O’Connell et al are still below 1.6 seconds in the *slowest* condition and therefore below our median response time of 1.7 seconds.

References

Gluth, S., Rieskamp, J., & Büchel, C. (2013). Classic EEG motor potentials track the emergence of value-based decisions. *Neuroimage*, 79, 394-403.

Lui, K. K., Nunez, M. D., Cassidy, J. M., Vandekerckhove, J., Cramer, S. C., & Srinivasan, R. (2021). Timing of readiness potentials reflect a decision-making process in the human brain. *Computational Brain & Behavior*, 4, 264-283.

O’Connell, R. G., Dockree, P. M., & Kelly, S. P. (2012). A supramodal accumulation-to-bound signal that determines perceptual decisions in humans. *Nature neuroscience*, 15(12), 1729-1735.

Decision Letter, first revision:

29th September 2023

Dear Dr Froemer,

Thank you once again for your revised manuscript, entitled "Common neural choice signals emerge artifactually amidst multiple distinct value signals," and for your patience during the re-review process.

Your manuscript has now been evaluated by the same 3 reviewers who evaluated your original manuscript. All reviewer feedback is included at the end of this letter. You will see that while Reviewers #2 and #3 are now happy to recommend publication, Reviewer #1 continues to express some serious technical concerns. We remain interested in the possibility of publishing your study in Nature Human Behaviour, but we would like to consider your response to these outstanding concerns in the form of another revised manuscript before we make a decision on publication.

We believe that in order to address Reviewer #1's comments, you should carry out the simulations recommended by Reviewer #1 to provide a sufficient amount of new evidence testing the validity of your approach, as well as any other additional work necessary.

Finally, your revised manuscript must comply fully with our editorial policies and formatting requirements. Failure to do so will result in your manuscript being returned to you, which will delay its consideration.

In sum, we invite you to revise your manuscript taking into account all reviewer and editor comments. We are committed to providing a fair and constructive peer-review process. Do not hesitate to contact us if there are specific requests from the reviewers that you believe are technically impossible or unlikely to yield a meaningful outcome.

We hope to receive your revised manuscript within 4-8 weeks. I would be grateful if you could contact us as soon as possible if you foresee difficulties with meeting this target resubmission date.

- Include a "Response to the editors and reviewers" document detailing, point-by-point, how you addressed each editor and referee comment. If no action was taken to address a point, you must provide a compelling argument. This response will be used by the editors and reviewers to evaluate your revision.
- Highlight all changes made to your manuscript or provide us with a version that tracks changes.

[REDACTED]

We look forward to seeing the revised manuscript and thank you for the opportunity to review your work. Please do not hesitate to contact me if you have any questions or would like to discuss these revisions further.

Sincerely,

[REDACTED]

REVIEWER COMMENTS:

Reviewer #1:

Remarks to the Author:

The authors have provided a detailed response to my initial comments including new data analyses which I appreciate would have come at a considerable time and effort cost. These clarifications have brought to light the fact that I had not properly understood how their deconvolution method was being applied or the strong and, I believe invalid, assumptions that the authors were making about how evidence accumulation signals should align to stimulus and response. Having now examined the Unfold method in more detail and with the authors' clarification of their rationale, it is now my firm judgment that the authors are making a fundamental error in their interpretation of these results.

The authors response makes it clear that their approach rests on the strong assumption that an evidence accumulation signal should be exclusively or predominantly time-locked to response execution. In fact, in the case of the tasks that the authors have analyzed here, one would expect a real evidence accumulation signal to have a strong stimulus-aligned component given that its onset latency would be determined by stimulus onset. In close agreement with observations in area LIP, evidence-dependent CPP build-up has been observed to commence circa 200ms after evidence onset across a variety of tasks. An evidence accumulation signal would also of course have a response-aligned component as its peak would be expected to covary with RT. Immediately then we should see the problem with applying a deconvolution algorithm that is designed to assign activity to either the stimulus- or response-aligned component or the residuals. No evidence accumulation signal would be expected to be fully retained in the response-aligned deconvolution component except in the extreme circumstance where jitter in signal onset vastly exceeds jitter in the delay between choice commitment (peak of evidence accumulation) and response. Given the strong exogenous cues marking stimulus onset in each task (stimulus onset in the visual search and Boldt tasks, auditory cue in Steinemann et al 2018) it is highly likely that there is at least as much jitter in the interval between commitment and button press. Indeed, there is strong empirical evidence of substantial motor time variability in perceptual choice tasks, including the work of Mathieu Servant and colleagues which shows that pre-movement EMG burst durations are subject to systematic modulation by time pressure and that button presses can be countermanded even after an EMG burst has been initiated. If the pre- and post-accumulation latency jitters are equal then CPP variability would be equally shared between the stimulus and response locked components and I think Unfold would likely apportion this activity to its residuals i.e. the activity is excluded. If post-commitment latency jitter is greater than onset jitter then Unfold would assign most of the activity to its stimulus-locked component. In other words, the authors' results are entirely consistent with what you would expect to see of a true evidence accumulation signal under realistic conditions.

This misunderstanding is evident in the authors' response to my queries. For example, they state that a key advantage of the Unfold method over RIDE is that it allows for substantial variability in the onset of stimulus-locked components. The authors argue that 'the subjective onset of the evidence is likely delayed and significantly more variable for low compared to high contrast.' This statement neatly highlights the key problem here because this is also exactly what you would expect of an evidence accumulation signal (later/slower build-up for weaker evidence). The authors further state "The P3 component has not generally been considered to be stimulus-evoked, but it can still be more closely aligned with the stimulus or more closely aligned with the response. In the datasets at hand, we find that the component is more closely aligned with the stimulus." Indeed, as I have tried to outline above, evidence accumulation signals can also be more closely aligned with the stimulus or more closely aligned with the response depending on the circumstances. In sum, showing that the CPP is more aligned to stimulus does not in any way contradict claims that it traces an evidence accumulation signal.

If the authors disagree with the above assessment then I think a simple solution that would not take much time would be for them to generate a simulated evidence accumulation signal and to run it through Unfold deconvolution. These simulation results would need to be presented in the paper and their accordance/disagreement with the authors claims discussed.

The authors' response has also brought to light some other issues in the description of methods that were missing in the original submission. "Note that to give the CPP the best possible shot of

appearing, with these two new reanalyses we took the additional conservative step of not allowing stimulus-locked events to vary with RT". There was no mention in the original submission of allowing stimulus-locked events to vary with RT. Any deconvolution approach that allows stimulus-locked events to vary with RT would appear tailor-made to eliminate evidence accumulation signals. I am glad that this step was not included in the re-analysis of the Steinemann data, it should be removed for the other analyses too if that has not already been done already. I am concerned that this important element of the analysis was not reported in the original submission as far as I could tell. In fact, the regression equations used for the deconvolution method are not reported for any of the tasks analysed and these should now be added as again they may involve the application of certain assumptions that should be made clear to the reader and justified. I note too that the methods for analysis of the Boldt and Steinemann data appear to be entirely missing from the revised submission. The authors should report which trials/conditions were included in their analyses, how the data were correct for artifacts, filters applied etc so that their analyses can be easily replicated. I am grateful to the authors for now adding stimulus-aligned deconvolution waveforms, however it would be important to show separate waveforms for fast/slow RT and high/low difficulty so that the presence of evidence accumulation dynamics can be assessed.

Finally, if the authors do opt to publish their findings as they stand then I think there are some previous findings pertaining to the CPP that they should consider as they seem hard to reconcile with signal overlap. First, is Figure 3 of the O'Connell et al (2012, Nat Neurosci) paper where a brief pulse in the evidence time-course led to a corresponding sustained perturbation of the CPP. The data shown in that plot extend to 1300ms and show only data points before response execution. Second, several studies have shown that evidence accumulation signals simulated from diffusion model fits to behaviour reproduce the CPP in detail including its stimulus and response-aligned waveforms, the relationship between its pre-response amplitude and choice RT+accuracy and modulations of its pre-response amplitude by experimental manipulations of time pressure and prior knowledge (e.g. Twomey et al 2015, Eur J Neurosci; Pisauro et al 2017, Nat Comms; Kelly, Corbett & O'Connell, 2021, Nat Hum Beh). The authors should discuss how their interpretation of these deconvolution results tally with this previous work validating the CPP as an evidence accumulation signal.

Reviewer #2:

Remarks to the Author:

The authors addressed all of my previously raised concerns and remarks thoroughly.

Reviewer #3:

Remarks to the Author:

The authors have well-addressed the reviewers' comments. The new analysis of data from two additional perceptual decision making tasks to address reviewer concerns is particularly noteworthy and convincing. I have only 2 minor corrections and one suggested addition, to improve manuscript clarity and not necessary for final acceptance – in my opinion.

Minor

Bottom of page 16 and top of page 17: Should the deconvolved topographies using the algorithm of Ehinger & Dimigen (2019) be placed in the supplementals so that readers can compare to the topographies of Figure 3?

The references to Fig. 2A and Fig. 2B on page 11 should be Fig. 3A and Fig. 3B respectively.

Methods on page 34: In the new paragraph the first "pca" should be capitalized "PCA". The second "pca" is fine lowercase because it refers to a MATLAB function, though sometimes authors add parentheses for clarity, e.g. "pca()".

Author Rebuttal, first revision:

Reviewer #1:

Remarks to the Author:

The authors have provided a detailed response to my initial comments including new data analyses which I appreciate would have come at a considerable time and effort cost. These clarifications have brought to light the fact that I had not properly understood how their deconvolution method was being applied or the strong and, I believe invalid, assumptions that the authors were making about how evidence accumulation signals should align to stimulus and response. Having now examined the Unfold method in more detail and with the authors' clarification of their rationale, it is now my firm judgment that the authors are making a fundamental error in their interpretation of these results.

The authors response makes it clear that their approach rests on the strong assumption that an evidence accumulation signal should be exclusively or predominantly time-locked to response execution. In fact, in the case of the tasks that the authors have analyzed here, one would expect a real evidence accumulation signal to have a strong stimulus-aligned component given that its onset latency would be determined by stimulus onset.

The reviewer is correct that one of our basic assumptions was that the evidence accumulation signal should be tied to the response, an assumption that is grounded in relevant past papers in the literature. For instance, in the seminal paper by O'Connell et al in Nature Neuroscience, the authors test the prediction that (a) the signature of evidence accumulation should peak at the time of the response, with a common peak amplitude across conditions, and (b) that the ramp to this threshold should vary in slope, essentially taking longer, the weaker the signal. While there have been significant extensions to this basic model, and the original authors have communicated in personal communications that the expected patterns may be task-dependent, it is common in the literature that people use these exact criteria to test for evidence accumulation signals, and/or to draw conclusions about the underlying process based on the patterns in this signature (cf. PISAURO et al, Boldt et al). For that reason, we thought it necessary to address this common set of predictions.

The reviewer is correct, however, that other sequential sampling models may generate different predictions in terms of timing or indicators of signatures of evidence accumulation.

We had previously simulated the model used in Hunt et al (2012), to account for another class of evidence accumulation models than the DDM that makes distinct predictions for temporal effects of value in value-based choice. Now, just like the DDM is not the only evidence accumulation model, there are many variants of the DDM. The neurally informed DDM employed now by Kelly, O'Connell and colleagues can generate predictions for the precise scenario described by the reviewer. The original authors have been so kind to share their code, and we have translated this into Julia to simulate neural activity for a set of conditions not addressed so far. In short, we show that unfold can recover signatures of evidence accumulation in scenarios as described by the reviewer, and that the patterns predicted by the simulations do not match our empirical data. While we cannot possibly test every possible sequential sampling model since the space of such models is infinite, we believe that this particular exercise was a valuable stress-test for our approach and it substantially strengthens our conclusions.

In close agreement with observations in area LIP, evidence-dependent CPP build-up has been observed to commence circa 200ms after evidence onset across a variety of tasks. An evidence accumulation signal would also of course have a response-aligned component as its peak would be expected to covary with RT. Immediately then we should see the problem with applying a deconvolution algorithm that is designed to assign activity to either the stimulus- or response-aligned component or the residuals. No evidence accumulation signal would be expected to be fully retained in the response-aligned deconvolution component except in the extreme circumstance where jitter in signal onset vastly exceeds jitter in the delay between choice commitment (peak of evidence accumulation) and response. Given the strong exogenous cues marking stimulus onset in each task (stimulus onset in the visual search and Boldt tasks, auditory cue in Steinemann et al 2018) it is highly likely that there is at least as much jitter in the interval between commitment and button press. Indeed, there is strong empirical evidence of substantial motor time variability in perceptual choice tasks, including the work of Mathieu Servant and colleagues which shows that pre-movement EMG burst durations are subject to systematic modulation by time pressure and that button presses can be countermanded even after an EMG burst has been initiated. If the pre- and post-accumulation latency jitters are equal then CPP variability would be equally shared between the stimulus and response locked components and I think Unfold would likely apportion this activity to its residuals i.e. the activity is excluded. If post-commitment latency jitter is greater than onset jitter then Unfold would assign most of the activity to its stimulus-locked component. In other words, the authors' results are entirely consistent with what you would expect to see of a true evidence accumulation signal under realistic conditions.

The reviewer raises a reasonable concern about potential limitations of our analysis approach, which we were fortunately able to test directly by simulating the alternate possibilities suggested by the reviewer. Specifically, we simulated evidence accumulation signals that emerge under varying amounts of stimulus and response variability. As anticipated by the reviewer, the time-locking of signatures of evidence accumulation would indeed vary as a function of the ratio of these two sources of variability, moving towards the stimulus as the ratio motor variability to stimulus variability increases.

Critically, our simulations predict that any signatures of evidence accumulation should still be visible after deconvolution across these varying conditions, albeit visible at different times and time-locked to different events. While these simulations slightly changed the event-locking of evidence accumulation signals, none of them matched the human participant data and they are further unable to account for the moving peaks in our primary datasets.

This misunderstanding is evident in the authors' response to my queries. For example, they state that a key advantage of the Unfold method over RIDE is that it allows for substantial variability in the onset of stimulus-locked components. The authors argue that 'the subjective onset of the evidence is likely delayed and significantly more variable for low compared to high contrast.' This statement neatly highlights the key problem here because this is also exactly what you would expect of an evidence accumulation signal (later/slower build-up for weaker evidence). The authors further state "The P3 component has not generally been considered to be stimulus-evoked, but it can still be more closely aligned with

the stimulus or more closely aligned with the response. In the datasets at hand, we find that the component is more closely aligned with the stimulus.” Indeed, as I have tried to outline above, evidence accumulation signals can also be more closely aligned with the stimulus or more closely aligned with the response depending on the circumstances. In sum, showing that the CPP is more aligned to stimulus does not in any way contradict claims that it traces an evidence accumulation signal.

If the authors disagree with the above assessment then I think a simple solution that would not take much time would be for them to generate a simulated evidence accumulation signal and to run it through Unfold deconvolution. These simulation results would need to be presented in the paper and their accordance/disagreement with the authors claims discussed.

Using simulation code from Kelly & O’Connell, we were able to simulate an evidence accumulation signal with physiologically plausible delays, jitter between stimulus onset & evidence accumulation onset, and jitter between evidence accumulation offset & button press. This allowed us to directly address the reviewer’s concern that due to the mismatch in ratio, evidence accumulation signal could be “lost in the residuals” - which we show is not the case.

We now include simulations in the supplement and refer to them on page 21 in the main text:

“So far, we assumed that non-decision time, comprising stimulus and motor processes at both ends of the evidence accumulation process, is constant, and that the time required for motor processes is short. However, some work suggests that both stimulus and motor processes can vary from trial to trial, leading to an evidence accumulation process that is not closely tied to either stimulus or response (Kelly, Corbett, & O’Connell, 2021). To test whether unfold’s deconvolution approach would be robust to detecting evidence accumulation signals when accounting for these different forms of variability, we simulated the neurally informed DDM (simulation code adapted from Kelly & O’Connell) for strong and weak signals with varying ratios of stimulus and response variability. Indeed, as response variability increases, unfold will increasingly assign activity to the stimulus event, rather than the response event (Supplementary Figure 5). Importantly, these simulations show that any signatures of evidence accumulation are recoverable, either at the stimulus, the response, or both. “

Supplementary Figure 5. Comparison of Mass-univariate and Deconvolution analyses of simulated evidence accumulation signals. Simulated evidence accumulation signals for regimes with stimulus and motor variability ratios 2:1, 1:1 and 1:2 (left to right) were analyzed with mass

univariate analyses (top) and unfold (bottom), locked to the stimulus (left in each) and to the response (right in each). Evidence accumulation peaks slightly after the motor response (due to post-decision accumulation), and a difference in drift-rate (easy vs. hard) can be seen in ramp-up speed and amplitude differences. As expected, if the simulated evidence accumulation signal is more tightly locked to the response, then that evidence accumulation signal is most likely to emerge in the response-locked deconvolution (first 2x2 panel). If the evidence accumulation signal is more tightly time-locked to the stimulus, it will be most likely to emerge in the stimulus-locked deconvolution (last 2x2 panel). Note that, across these different conditions, no signal is lost as a function of deconvolution: if one would combine the deconvolved estimates with their respective event-timing, one could always retrieve the upper-row, overlapping estimates.

We also describe them briefly in the Method at the end:

“Simulations. We simulated the neurally informed DDM based on code shared by Simon Kelly and Redmond O’Connell. In this model, stimulus processing as well as motor processes vary, so that the timing of the evidence accumulation process in between both is jittered. Both jitters were modelled using the simplifying assumption of uniform distributions with duration “stim_jitter” and “motor_jitter”, respectively. Further, we report the ratio $\text{stim_jitter}/\text{motor_jitter}$, indicating whether the evidence accumulation process is more closely associated with the stimulus or the motor. We simulated 3 scenarios, 1) 2/1 ratio, whereby stim_jitter was set to 0.2s and motor_jitter to 0.1s, 2) 1/1 ratio, where both were set to 0.2s, and 3) 1/2 ratio, where stim_jitter was set to 0.1s and motor_jitter to 0.2s.”

The authors’ response has also brought to light some other issues in the description of methods that were missing in the original submission. “Note that to give the CPP the best possible shot of appearing, with these two new reanalyses we took the additional conservative step of not allowing stimulus-locked events to vary with RT”. There was no mention in the original submission of allowing stimulus-locked events to vary with RT. Any deconvolution approach that allows stimulus-locked events to vary with RT would appear tailor-made to eliminate evidence accumulation signals. I am glad that this step was not included in the re-analysis of the Steinemann data, it should be removed for the other analyses too if that has not already been done already. I am concerned that this important element of the analysis was not reported in the original submission as far as I could tell. In fact, the regression equations used for the deconvolution method are not reported for any of the tasks analysed and these should now be added as again they may involve the application of certain assumptions that should be made clear to the reader and justified. I note too that the methods for analysis of the Boldt and Steinemann data appear to be entirely missing from the revised submission. The authors should report which trials/conditions were included in their analyses, how the data were correct for artifacts, filters applied etc so that their analyses can be easily replicated. I am grateful to the authors for now adding stimulus-aligned deconvolution waveforms, however it would be important to show separate waveforms for fast/slow RT and high/low difficulty so that the presence of evidence accumulation dynamics can be assessed. Finally, if the authors do opt to publish their findings as they stand then I think there are some previous findings pertaining to the CPP that they should consider as they seem hard to reconcile with signal overlap. First, is Figure 3 of the O’Connell et al (2012, Nat Neurosci)

paper where a brief pulse in the evidence time-course led to a corresponding sustained perturbation of the CPP. The data shown in that plot extend to 1300ms and show only data points before response execution. Second, several studies have shown that evidence accumulation signals simulated from diffusion model fits to behaviour reproduce the CPP in detail including its stimulus and response-aligned waveforms, the relationship between its pre-response amplitude and choice RT+accuracy and modulations of its pre-response amplitude by experimental manipulations of time pressure and prior knowledge (e.g. Twomey et al 2015, Eur J Neurosci; Pisauro et al 2017, Nat Comms; Kelly, Corbett & O’Connell, 2021, Nat Hum Beh). The authors should discuss how their interpretation of these deconvolution results tally with this previous work validating the CPP as an evidence accumulation signal.

Thank you for pointing out these gaps in our previous description. We have now expanded on the method section to clarify which specific analyses we performed and how each event was modelled.

1) Regressors, p. 35

“Regressors for (Frömer, Maier, & Abdel Rahman, 2018) were visual field and stimulus quality, and both stimulus and response-locked activity were modeled with both regressors. Regressors for the three other studies were median split RT for response-locked activity, whereas we initially only modelled intercepts for stimulus locked activity. In a set of control analyses, we reanalyzed the data of the latter three studies to test whether stimulus and response-locked activity varied in ways predicted by unfold analyses of simulated data under different stimulus and response variability regimes.”

We now show a summary figure of our reanalyses in the Supplement and have added corresponding text in the main text.

Supplementary Figure 6. Test for stimulus and response-locked signatures of evidence accumulation in Mass-univariate analyses compared to unfold deconvolution. Across the 4 datasets, we do not see the signatures of evidence accumulation predicted by the simulations in Supplementary Figure 5, except for the visual search task (B). Note that in this task the same pattern is predicted by a non-integration account, where the P3 component associated with detection of the deviant object is delayed and more variable when the signal is degraded in the blurred condition. **A.** Data from Pisauro, Fouragnan, Retzler, & Philiastides (2017). **B.** Data from Frömer, Maier, & Abdel Rahman (2018). **C.** Data from Steinemann, O'Connell, & Kelly (2018). **D.** Data from Boldt, Schiffer, Waszak, & Yeung (2019).

Main text p. 22:

“We had indeed observed a pattern in principle consistent with such stimulus-locked evidence accumulation signatures, but also with non-integration accounts, in the visual search dataset (Frömer, Maier, & Abdel Rahman, 2018), see Supplementary Figure 4B. To test whether signatures of evidence accumulation may have been moved towards the stimulus in the remaining datasets, and thus been missed in our previous analyses, we reanalyzed these data again, this time allowing stimulus-locked data to vary with response time in addition to allowing response-locked data to vary. The resulting patterns were not consistent with any of the simulated patterns (Supplementary Figure 6).”

2) Method

We had previously reported analyses for all studies, however, we now specify in more detail which trials were used. We are also making all scripts available so that replicability should not be a concern.

p. 35:

“Please see the original publications for detailed information on EEG data acquisition in Study 3 (Pisauro et al., 2017), Study 4 (Frömer et al., 2018), Study 5 (Steinemann et al., 2018), and Study 6 (Boldt et al., 2019). We obtained preprocessed data for Studies 3 and 4. Raw data for Study 5 were concatenated across blocks, re-referenced and low-pass filtered at 40Hz. A subset of blocks (18 total from 298 blocks across all participants) could not be matched with the behavioral data and were excluded from further analyses. We obtained data for Study 6 with ocular artifact reduction and downsampled to 250Hz and removed horizontal and vertical EEG channels as well as M2. Data were then matched with behavioral data and low-pass filtered at 40Hz.”

3) Discussion

We have now added additional discussion of the current results relative to previous findings.

“... Our work highlights the importance of testing multiple predictions.

Doing so, several studies have shown that evidence accumulation signals simulated from diffusion model fits to behavior reproduce the CPP in detail, including its stimulus and response-aligned waveforms, the relationship between its pre-response amplitude and choice RT and accuracy, and modulations of its pre-response amplitude by experimental manipulations of time pressure and prior knowledge (e.g., Kelly, Corbett, & O'Connell, 2021; O'Connell, Dockree, & Kelly, 2012; Twomey et al., 2015). Here, we highlight several cases in which such seemingly strong evidence could emerge artifactually from component overlap by virtue of evidence accumulation and component overlap sharing a common link to response time. Thus, in addition to testing multiple predictions of evidence accumulation accounts, future research can avoid misinterpretations of neural activity by inspecting ERP image plots for temporal patterns in single-trial data and deconvolving stimulus and response-locked signals (Ehinger & Dimigen, 2019) to show that the putative evidence accumulation pattern cannot be produced by component overlap alone.

While we have provided evidence for the pervasive risk that these artifacts pose for inferring evidence accumulation from the CPP, we can not claim to have definitively ruled out that the CPP can in certain cases carry the signature of an evidence accumulation signal. For instance, one finding that is not easily accounted for by component overlap is O'Connell, Dockree, and Kelly's (2012) observation that perturbations in the stimulus lead to expected patterns in the CPP. In a continuous detection task, where participants needed to indicate when a stimulus was disappearing, O'Connell et al. showed that briefly increasing stimulus contrast led to an attenuation of the average stimulus-locked CPP that ramped up again as stimulus contrast continued to reduce. Such a pattern is difficult to explain with component overlap alone, but may still be accounted for by other non-integration accounts. Novel approaches are therefore needed to test the evidence accumulation hypothesis against alternative non-integration models (Latimer, Yates, Meister, Huk, & Pillow, 2015; Stine, Zylberberg, Ditterich, & Shadlen, 2020).”

Reviewer #2:

Remarks to the Author:

The authors addressed all of my previously raised concerns and remarks thoroughly.

We thank the reviewer for their positive evaluation.

Reviewer #3:

Remarks to the Author:

The authors have well-addressed the reviewers' comments. The new analysis of data from two additional perceptual decision making tasks to address reviewer concerns is particularly noteworthy and convincing. I have only 2 minor corrections and one suggested addition, to improve manuscript clarity and not necessary for final acceptance – in my opinion.

Minor

Bottom of page 16 and top of page 17: Should the deconvolved topographies using the algorithm of Ehinger & Dimigen (2019) be placed in the supplementals so that readers can compare to the topographies of Figure 3?

Thank you for this pointer, we have now included a novel Supplementary Figure 2 showing the topographies for both approaches side-by-side. We agree this is helpful.

Supplementary Figure 2. Estimate-topographies for MASS-univariate and Unfold analyses.

Depicted are average regression-coefficients in each 100 ms time window following stimulus onset (left column) and preceding the response (right column) for Appraisal and Choice PCs, respectively. Light shaded rows show Mass-Univariate results, and dark shaded rows show unfold results.

Electrodes and time-points that belong to significant clusters are highlighted as black dots.

We now refer to this Figure alongside Supplementary Figure 1 on page 17:

“Like our previous analyses, this approach again identified a positive stimulus-locked appraisal cluster with a centro-parietal distribution (peak around 810 ms at electrode Pz, $p = .004$) and response-locked choice clusters over frontal (positive, peak around -722 ms at electrode AFz, $p = .010$) and parietal (negative, peak around -616 ms at electrode P8, $p = .010$) sites, respectively (Supplementary Figure 1, 2).”

The references to Fig. 2A and Fig. 2B on page 11 should be Fig. 3A and Fig. 3B respectively.

Thank you for catching these, we have fixed them.

Methods on page 34: In the new paragraph the first “pca” should be capitalized “PCA”. The second “pca” is fine lowercase because it refers to a MATLAB function, though sometimes authors add parentheses for clarity, e.g. “pca()”.

Thank you for catching this one as well. We have fixed it.

Decision Letter, second revision:

11th December 2023

Dear Dr Froemer,

Thank you once again for your manuscript, entitled "Common neural choice signals emerge artifactually amidst multiple distinct value signals," and for your patience during the peer review process.

Your manuscript has now been evaluated by Reviewer #1, who continues to express concerns about the technical aspects of the manuscript. We feel that it will be necessary to consult an independent expert reviewer to resolve these concerns. We would therefore ask you to provide a full response to Reviewer #1, including a response to the reviewer and a revised manuscript. We will then ask an independent reviewer to provide a final opinion on the disputed issues.

In sum, we invite you to revise your manuscript taking into account all reviewer and editor comments. We are committed to providing a fair and constructive peer-review process. Do not hesitate to contact us if there are specific requests from the reviewers that you believe are technically impossible or unlikely to yield a meaningful outcome.

We hope to receive your revised manuscript within two months. I would be grateful if you could contact us as soon as possible if you foresee difficulties with meeting this target resubmission date.

- Include a “Response to the editors and reviewers” document detailing, point-by-point, how you addressed each editor and referee comment. If no action was taken to address a point, you must provide a compelling argument. When formatting this document, please respond to each reviewer comment individually, including the full text of the reviewer comment verbatim followed by your

response to the individual point. This response will be used by the editors to evaluate your revision and sent back to the reviewers along with the revised manuscript.

- Highlight all changes made to your manuscript or provide us with a version that tracks changes.

[REDACTED]

We look forward to seeing the revised manuscript and thank you for the opportunity to review your work. Please do not hesitate to contact me if you have any questions or would like to discuss these revisions further.

Sincerely,

[REDACTED]

REVIEWER COMMENTS:

Reviewer #1:

Remarks to the Author:

Once again I am grateful to the authors for providing additional clarifications and simulations. I am very mindful of the fact that I am now the only reviewer delaying publication and so I have returned my comments within 24 hours in order to limit any further delays. As the authors know, the strong and eye-catching claims of this paper have the potential to lead many to disregard past and future findings related to the CPP signal without necessarily having the time or expertise to critically evaluate the evidence in favour of those claims. Given this, it is vital that the paper meets a minimum standard of clarity in describing all methodological steps, the rationale for them, and explaining exactly how conclusions follow from their outcomes. I'm afraid that this standard has still not been met. As I outline in more detail below, I believe that the simulations confirm my original concern that it is not possible to make clear predictions regarding the alignment of evidence accumulation (EA) signals to stimulus or response without estimating sensory and motor time variability. Consequently, while the authors' simulations highlight an issue that COULD arise under very specific circumstances, the empirical data offer no evidence that it DOES arise in the perceptual tasks under consideration. As mentioned previously, I believe the results of the value-based choice task are compelling and should be the focus of the paper, my issue is with the strong claims regarding the perceptual choice data. I therefore again recommend that the authors consider refocusing the paper on the value-based task and temper their claims about the perceptual choice results.

Major Comments:

The simulations show exactly the pattern that I predicted in my original comments – evidence accumulation signals will vary in their alignment to the stimulus or response depending on the ratio of sensory to motor variability. It is heartening that, in the examples shown, Unfold can recover the activity, but this was only tangential to my central point (and I do apologise if my original comments were unclear on this). The key issue is that the simulations show that we cannot make clear predictions regarding the relative alignment of EA signals to stimulus versus response in a particular dataset without first estimating or making assumptions about those sensory: motor variability ratios and therefore the fact that Unfold reduces response-aligned build-up rate effects tells us very little about the nature of the CPP. With extreme ratios in either direction, the simulations show that Unfold will fully assign the EA build-up rate effects to the stimulus or to the response locked component while at intermediate ratios it will presumably divide the signal across both components. Strikingly though, when sensory and motor variability are equivalent in the simulation (probably the most realistic scenario), Unfold eliminates most of the build-up rate effects from the response-locked

component - the EA signal is equally aligned to stimulus and response and yet Unfold assigns its evidence-dependent activity to the stimulus component. This last observation alone surely raises major questions about the validity of the authors' approach and should give them pause but it is not mentioned in the revised manuscript or in their response to the reviewer comments. The authors' interpretation of their results thus seems to hinge entirely on their assumption that variability in pre-decisional sensory delays should far exceed post-commitment motor time variability despite the fact that in the tasks under consideration the evidence onsets are entirely predictable and/or exogenously cued.

This brings me to my other major concern which is that the authors state at various points that the simulations do not match the empirical data but nowhere in their response to my comments or in the manuscript itself do they offer any explanation of how they don't match or why they should be expected to. While the simulations are useful for demonstrating that EA signal alignment can vary depending on sensory and motor variability, I do not see how they can be used to directly verify the patterns observed in the empirical data. The authors state they used the model in Kelly et al (2021) for their simulation and, while details on the parameter values are not provided, I would presume they used the same or similar values to those reported in that paper. What they have not done is fit the model to any of their own behavioural datasets and then run simulations from those best fitting parameter values which is what would be needed to directly verify whether their Unfold results are consistent or inconsistent with what would be predicted of an EA signal. I am certainly not suggesting that the authors would need to run such analyses for the present paper, I am just pointing out the limits of what can be gleaned from their own simulations. To underline the point, the empirical signals show pre-response build-up over a far longer period (circa 400-800ms depending on the task) than those in the simulations (circa 200ms) indicating that they are not particularly representative of the data under scrutiny. In any case, the authors do not specify what features of the simulated signals they are comparing to the empirical data when they assert that they do not match. As mentioned above, it seems to me that there is a high degree of similarity between the pattern in the second simulation (sensory==motor variability) and the pattern in the Steinemann data where Unfold removes evidence-dependent build-up from the response-locked trace. It behooves the authors to properly engage with these important details, to fully clarify their thinking to the reader and/or to temper their claims. I note again here that my comments pertain to the perceptual choice data and not the value task, where as mentioned previously, I believe there is a convincing case that the CPP is not tracing the decision process.

In their response to my first comment, the authors suggest that a key purpose of the paper was to address the common prediction that the CPP should peak prior to response and have an evidence-dependent build-up. Here again, the simulations do actually support these standard predictions - the EA signals in the simulation peak just prior to response and their build-up rate effects of difficulty are evident in the response-aligned waveforms for all three simulations. The only issue arises when an effort is made to separate the signal out into stimulus and response locked components with Unfold which previous studies have not attempted to do. It is in fact Unfold itself that fundamentally changes how an EA signal will manifest in stimulus and response-locked components.

In their response to my first comment the authors imply that the DDM makes one set of predictions regarding EA signal alignment while 'other sequential sampling models may generate different predictions in terms of timing or indicators of signatures of evidence accumulation'. But it is crucial to keep in mind that most sequential sampling model variants including the DDM are entirely silent on this issue, combining sensory and motor time variability effects on behaviour into a single parameter and therefore making no specific predictions about EA signal alignment. So this is not a matter particular to the DDM or one of conflicting predictions across different models, the issue applies equally to any model of perceptual choice that does not furnish separate estimates of sensory and motor variability (and to my knowledge that is all of them bar Kelly et al 2021).

Minor comments:

In the revised manuscript the authors now state "However, some work suggests that both stimulus and motor processes can vary from trial to trial, leading to an evidence accumulation process that is not closely tied to either stimulus or response (Kelly, Corbett, & O'Connell, 2021)." This phrasing seems to present the case as if the notion of trial-to-trial variability in sensory or motor processing

times is a recent discovery when it has been well-established for many decades of research in psychophysics and neuroscience.

Changes to the manuscript are not clearly marked in bold and in some cases are not indicated in the reviewer comments either.

Author Rebuttal, second revision:

Reviewer #1:

Remarks to the Author:

Once again I am grateful to the authors for providing additional clarifications and simulations. I am very mindful of the fact that I am now the only reviewer delaying publication and so I have returned my comments within 24 hours in order to limit any further delays. As the authors know, the strong and eye-catching claims of this paper have the potential to lead many to disregard past and future findings related to the CPP signal without necessarily having the time or expertise to critically evaluate the evidence in favour of those claims. Given this, it is vital that the paper meets a minimum standard of clarity in describing all methodological steps, the rationale for them, and explaining exactly how conclusions follow from their outcomes. I'm afraid that this standard has still not been met. As I outline in more detail below, I believe that the simulations confirm my original concern that it is not possible to make clear predictions regarding the alignment of evidence accumulation (EA) signals to stimulus or response without estimating sensory and motor time variability.

Consequently, while the authors' simulations highlight an issue that COULD arise under very specific circumstances, the empirical data offer no evidence that it DOES arise in the perceptual tasks under consideration. As mentioned previously, I believe the results of the value-based choice task are compelling and should be the focus of the paper, my issue is with the strong claims regarding the perceptual choice data. I therefore again recommend that the authors consider refocusing the paper on the value-based task and temper their claims about the perceptual choice results.

We thank the reviewer for their concern and the fast turnaround. We share the reviewer's desire for rigor and appreciate them pointing out where we have been lacking. Before delving into the individual comments, we believe there is an important clarification to be made.

To quickly summarize the evidence this argument is based on, we show:

- 1) using simulations that component overlap between stimulus- and response-locked activity can produce patterns that would traditionally be interpreted as evidence accumulation signals,
- 2) using unfold - a method to account for stimulus and response-locked effects simultaneously, allowing us to evaluate signatures of evidence accumulation while

accounting for overlap - that putative evidence accumulation signals vanish or are at least substantially reduced below statistical detectability when controlling for component overlap,

3) using simulation recovery experiments, analyzing evidence accumulation traces with unfold, that we can recover non-standard signatures of evidence accumulation (with varying time-locking to stimulus, response, or in between), and

4) that the patterns emerging from these simulations, also do not match the vast majority of results from a reanalysis of 4 empirical datasets showing signatures of evidence accumulation using standard approaches (3 perceptual).

A key disagreement from the perspective of the reviewer seems to be the predicted temporal alignment of the evidence accumulation signal, and whether such alignment can be taken as evidence **for** component overlap mimicking signatures of evidence accumulation. The reviewer says that by removing – or at least substantially reducing – typical evidence accumulation patterns, we fail to provide evidence that the component overlap issue **does** arise in the relevant perceptual datasets. According to them, since we cannot unequivocally show this, the conclusion then is that we must assume that these signals reflect evidence accumulation by default. By this argument, evidence accumulation essentially serves as the null hypothesis, considered true until proven otherwise.

The problem we see with this argument is two-fold:

1) The temporal alignment - to the response - has been taken as evidence **in support of** evidence accumulation. To quote the reviewer's response:

“In their response to my first comment, the authors suggest that a key purpose of the paper was to address the common prediction that the CPP should peak prior to response and have an evidence-dependent build-up. Here again, the simulations do actually support these standard predictions – the EA signals in the simulation peak just prior to response and their build-up rate effects of difficulty are evident in the response-aligned waveforms for all three simulations.”

A large array of research has built on the influential Nature Neuroscience paper by O'Connell et al (2012) to identify the pattern that work had so clearly laid out. A key prediction in all of this work is that the CPP should peak at the time of the response. The problem is, merely time-locking it to the response does not tell us that. In our simulations, varying parameters around the evidence accumulation process, we know by definition that the signal is more closely related to the stimulus when stimulus variability is lower than motor variability and that it is more closely related to the response when motor variability is lower than stimulus variability. Yet, by mere eye, the temporal alignment in these simulations is at best ambiguous until Unfold is applied. In contrast to the reviewer's conclusion that “It is in fact Unfold itself that fundamentally changes how an EA signal will manifest in stimulus and response-locked components.”, Unfold reveals this ground truth in the simulated data, and we recover that the response-locked peak prediction only pertains to a specific case, namely in which the motor variability is smaller than the stimulus variability.

The reviewer makes a great point that these quantities need to be assessed separately to generate proper, falsifiable predictions about signatures of evidence accumulation for each study. This is a very important point that we now make sure to emphasize in the manuscript alongside other recommendations for future work on signatures of evidence accumulation.

2) More problematic to this argument, we show that the same original predictions – that activity should temporally align to the response and that the build-up rate should be evidence-dependent – **are not unique to evidence accumulation**. The same patterns can arise from an entirely different mechanism under plausible and common conditions (median response times < 1s). Since component overlap is a ubiquitous issue in EEG research (see Smith and Kutas, 2015b), not just in the particular case of evidence accumulation signals, where the issue has been considered with regards to sensory signals from the very beginning (O’Connell et al 2012, Kelly et al 2013, Steinemann et al, 2018), we can all agree that it needs to be accounted for - especially when it is not merely a nuisance, but **could** generate the theoretically expected empirical pattern artifactually. **This is one of the key take-homes of our paper: there is a confound that needs to be accounted for when interpreting putative signatures of evidence accumulation**. We do not conclude that there is no signature of evidence accumulation in the CPP, merely that more evidence is required to assert that there is after accounting for overlap.

Having said that, we acknowledge that we have not been clear enough about either of these points and we have therefore revised our manuscript accordingly as we unpack below.

Major Comments:

The simulations show exactly the pattern that I predicted in my original comments – evidence accumulation signals will vary in their alignment to the stimulus or response depending on the ratio of sensory to motor variability. It is heartening that, in the examples shown, Unfold can recover the activity, but this was only tangential to my central point (and I do apologise if my original comments were unclear on this). The key issue is that the simulations show that we cannot make clear predictions regarding the relative alignment of EA signals to stimulus versus response in a particular dataset without first estimating or making assumptions about those sensory: motor variability ratios and therefore the fact that Unfold reduces response-aligned build-up rate effects tells us very little about the nature of the CPP. With extreme ratios in either direction, the simulations show that Unfold will fully assign the EA build-up rate effects to the stimulus or to the response locked component while at intermediate ratios it will presumably divide the signal across both components. Strikingly though, when sensory and motor variability are equivalent in the simulation (probably the most realistic scenario), Unfold eliminates most of the build-up rate effects from the response-locked component - the EA signal is equally aligned to stimulus and response and yet Unfold assigns its evidence-dependent activity to the stimulus component.

This last observation alone surely raises major questions about the validity of the authors' approach and should give them pause but it is not mentioned in the revised manuscript or in their response to the reviewer comments. The authors' interpretation of their results thus seems to hinge entirely on their assumption that variability in pre-decisional sensory delays should far exceed post-commitment motor time variability despite the fact that in the tasks under consideration the evidence onsets are entirely predictable and/or exogenously cued.

This brings me to my other major concern which is that the authors state at various points that the simulations do not match the empirical data but nowhere in their response to my comments or in the manuscript itself do they offer any explanation of how they don't match or why they should be expected to. While the simulations are useful for demonstrating that EA signal alignment can vary depending on sensory and motor variability, I do not see how they can be used to directly verify the patterns observed in the empirical data. The authors state they used the model in Kelly et al (2021) for their simulation and, while details on the parameter values are not provided, I would presume they used the same or similar values to those reported in that paper. What they have not done is fit the model to any of their own behavioural datasets and then run simulations from those best fitting parameter values which is what would be needed to directly verify whether their Unfold results are consistent or inconsistent with what would be predicted of an EA signal. I am certainly not suggesting that the authors would need to run such analyses for the present paper, I am just pointing out the limits of what can be gleaned from their own simulations. To underline the point, the empirical signals show pre-response build-up over a far longer period (circa 400-800ms depending on the task) than those in the simulations (circa 200ms) indicating that they are not particularly representative of the data under scrutiny. In any case, the authors do not specify what features of the simulated signals they are comparing to the empirical data when they assert that they do not match.

We apologize that we weren't clear in our simulation approach. One of our main motivations was indeed to provide a positive control that Unfold does not remove evidence accumulation signatures if they do actually exist. Beyond that, we agree with the reviewer that the simulations make it clear that the evidence accumulation pattern need not be response-locked, and that the mere reduction in the response-locked pattern is not sufficient evidence that there is no signature of evidence accumulation. This is indeed an important novel consideration that has been revealed through our findings and the interactions with the reviewer (and Redmond O'Connell and Simon Kelly in personal communications outside of this review process). What the simulations allow us to do is to generate predictions for alternate cases where these signatures **shouldn't** be response-locked and to examine the qualitative patterns we would expect to see under each of these conditions. To the best of our knowledge, this is the first time this has been explicitly done.

The reviewer worries about the validity of our approach based on our finding that under equal stimulus and response variability, unfold assigns the evidence accumulation signal to

the stimulus. This is only true for the complex model of evidence accumulation (which we got from O’Connel & Kelly, using the same parameter settings as they used). Indeed as visible in the following simulation, there is no general bias for any one event-type using unfold.

In this (over-)simplified simulation of an evidence integration process, we simulate a Stimulus / C-Component / Response sequence. For simplicity, the Stimulus and Response again do not show any activity, only the C-component is simulated with a hanning window centered around the event-onset. The uniform jitter-distribution between Stimulus and C-Component, and between

Response and C-Component is equal, the conditions fast / slow influence the width of the uniform jitter distribution (a reproducible notebook with all parameter values can be found here

https://github.com/froemero/Common_Neural_Choice_Signals_emerge_artificially/blob/main/analyses/nb_2024-02-09_jittersimulation.jl). What we can see is that the deconvolved version does not favour either the stimulus nor the response, it also does not abolish the evidence integration signal. It is reduced in amplitude, which is expected as in the mass-univariate (the “classical”) case, one effectively accounts for the C-component twice, independently. This simulation is not included in the revised manuscript, but can be added to the supplement if the reviewers think this would be useful.

So why exactly did we see a bias towards stimulus in the complex model (O’connel & Kelly)? We speculate that any one of the many additional parameters could affect the effective time-locking of the signal peak, including an urgency signal, and post-decision evidence accumulation. The former is, for instance, why this model predicts peak amplitude differences rather than only build-up differences to a common peak amplitude as in the earlier papers. These features further loosen the relationship between the signal’s peak and the response, and could explain why the simulations show a stimulus-locked signal in the case of equal variance in stimulus and motor processes. Yet, as mentioned earlier, this is the only model accounting for these sources of variability, and thus our benchmark.

Importantly, what we see following deconvolution is different from the standard response-locked evidence accumulation signatures. These standard signatures show a common peak at the time of the response and an evidence-dependent build-up, visible in greater

activation for hard compared to easy trials leading up to the response. Instead, under the Kelly et al (2021) model, there should be increased activation for easy compared to hard trials **either** response-locked (motor variability < stimulus variability) **or** stimulus-locked. There should further be a concomitant reduction in the average waveform for deconvolved compared to average (e.g., mass-univariate) waveforms that is greater stimulus locked when motor variability < stimulus variability, and greater response-locked when stimulus variability > motor variability. We can assume that either stimulus-locked or response-locked versions of these qualitative patterns should be evident in empirical data analyzed in the same way if the data were generated by the same generative process under one of these parameterizations. Thus, we would expect the typical effect if motor variability is lower than stimulus variability - modulo some additional differences in the shape induced by other components of the model, such as the peak amplitude difference induced by urgency signals - and we would expect a stimulus-locked signal concomitant with a significant reduction in the response-locked signal otherwise.

To test whether the alternative pattern was evident in the data, we reanalyzed the empirical data again, this time in the same way as the simulations, allowing both stimulus- and response-locked activity to vary as a function of median RT, to see whether we find evidence of stimulus-locked signatures of evidence accumulation. We do indeed find such a pattern in one out of the four studies, the visual search study. (Though, note that in this study, instead of modeling RT, we modeled the stimulus conditions.) However, in the remaining 3 datasets, we do not find the corresponding set of patterns.

We have now revised the manuscript text accordingly:

“So far, we assumed that **signatures of evidence accumulation should be closely tied to the response. This is the typical prediction in the literature and plausible if** non-decision time (comprising stimulus and motor processes at both ends of the evidence accumulation process) is constant, and the time required for motor processes is short. However, both stimulus and motor processes can vary from trial to trial, leading to an evidence accumulation process that is not closely tied to either stimulus or response. **The only model that explicitly and independently accounts for these variance components is the neurally informed DDM** (Kelly, Corbett, & O'Connell, 2021). To test whether unfold's deconvolution approach would be robust to detecting evidence accumulation signals – **either response- or stimulus-locked** – when accounting for these different forms of variability, we simulated the neurally informed DDM (simulation code adapted from Kelly & O'Connell) for strong and weak signals with varying ratios of stimulus and response variability. **The simulations confirm that signatures of evidence accumulation need not be response-locked.**

The simulations also show that, as response variability increases, unfold will increasingly assign activity to the stimulus event, rather than the response event

(Supplementary Figure 5). Importantly, these simulations show that *any* signatures of evidence accumulation are recoverable, either at the stimulus, the response, or both. **The simulations also provide us with qualitative alternative predictions for signatures of evidence accumulation under different ratios of stimulus- and motor variability. Specifically, under the Kelly et al (2021) model, we would expect that as motor variability increases, evidence accumulation signals would be evident in a greater *stimulus*-locked amplitude for easy/fast compared to hard/slow trials.**

Of the four datasets we analyzed, the only one in which we observed a pattern in principle consistent with such stimulus-locked evidence accumulation signatures, but also with non-integration accounts, was the visual search dataset (Frömer, Maier, & Abdel Rahman, 2018; see Supplementary Figure 4B). To test whether signatures of evidence accumulation may have been moved towards the stimulus in the remaining datasets, and thus missed in our previous analyses, we reanalyzed these data again, this time allowing stimulus-locked data to vary with response time in addition to allowing response-locked data to vary. **The resulting patterns in the three remaining datasets showed neither significant response-locked signatures of evidence accumulation, nor the predicted stimulus-locked amplitude modulations, and were thus inconsistent with any of the simulated patterns** (Supplementary Figure 6).

As mentioned above, it seems to me that there is a high degree of similarity between the pattern in the second simulation (sensory==motor variability) and the pattern in the Steinemann data where Unfold removes evidence-dependent build-up from the response-locked trace. It behooves the authors to properly engage with these important details, to fully clarify their thinking to the reader and/or to temper their claims. I note again here that my comments pertain to the perceptual choice data and not the value task, where as mentioned previously, I believe there is a convincing case that the CPP is not tracing the decision process.

To compare our second simulation and the Steinemann data, we have juxtaposed the relevant figures below. As the reviewer suggests, the simulated pattern (panels on the left below) does look similar to what we find in our mass univariate analyses (panels on the right). However, note that we can also get the observed pattern from component overlap. After we perform deconvolution (bottom panels), the simulation and the (deconvolved) empirical data are no longer similar. While there is indeed a stimulus-locked P3 and a response-locked component, the stimulus-locked component in the simulations varies by condition, with a larger amplitude for easy compared to difficult trials. We do not see that in the empirical data. The marginal response-locked differences in the empirical data are neither significant nor does this pattern match what is in the simulations. Any appearance of similarity between simulations and observed data for the Steinemann study are therefore illusory.

Figure: Comparison of signatures of evidence accumulation in MASS-univariate analyses (top) and unfold analyses (bottom) of simulated data from an evidence accumulation model (A) and empirical data from Steinemann et al. (B). Within each panel stimulus-locked data are shown left and response-locked data right, dark lines show difficult/slow trials and light lines show easy/fast trials. Patterns for MASS-univariate analyses match between simulated and empirical data, but following deconvolution they do not match.

Taking this a step further, we conducted an additional control analysis, to test whether the observations in this dataset can be explained by component overlap alone. To this end, we first deconvolved stimulus and response events *without* assuming any modulation. We then analyzed the residuals from this analysis using MASS-univariate analyses with the median-split RT regressor as we did before. The first analysis would only remove stimulus- and response-related activity, and their overlap, agnostic of any variations in the signal as a function of RT or any other indicator of evidence accumulation. It would also leave any central components and variability in it untouched. Thus, if there is an evidence accumulation signal accounting for the component overlap, it should still be visible in the MASS-Univariate analysis as it always is. We do not see any such signature of evidence accumulation in the overlap-corrected data (see figure below).

Supplementary Figure 7. No evidence for evidence accumulation after correcting for component overlap. Signatures of evidence accumulation observed with MASS-univariate analyses of uncorrected data vanish when in the same way analyzing data corrected for component overlap by deconvolving stimulus- and response-related activity agnostic of RT.

We include this result in the Supplement and reference it in the text as follows (p. 22):

“These findings are complemented by results showing that signatures of evidence accumulation can no longer be found using MASS-univariate analysis following RT-agnostic overlap correction of the data (Supplementary Figure 7).”

In their response to my first comment, the authors suggest that a key purpose of the paper was to address the common prediction that the CPP should peak prior to response and have an evidence-dependent build-up. Here again, the simulations do actually support these standard predictions – the EA signals in the simulation peak just prior to response and their build-up rate effects of difficulty are evident in the response-aligned waveforms for all three simulations. The only issue arises when an effort is made to separate the signal out into stimulus and response locked components with Unfold which previous studies have not attempted to do. It is in fact Unfold itself that fundamentally changes how an EA signal will

manifest in stimulus and response-locked components.

In their response to my first comment the authors imply that the DDM makes one set of predictions regarding EA signal alignment while ‘other sequential sampling models may generate different predictions in terms of timing or indicators of signatures of evidence accumulation’. But it is crucial to keep in mind that most sequential sampling model variants including the DDM are entirely silent on this issue, combining sensory and motor time variability effects on behaviour into a single parameter and therefore making no specific predictions about EA signal alignment. So this is not a matter particular to the DDM or one of conflicting predictions across different models, the issue applies equally to any model of perceptual choice that does not furnish separate estimates of sensory and motor variability (and to my knowledge that is all of them bar Kelly et al 2021).

The reviewer is making an excellent and important point here that we have so far failed to appropriately emphasize in our manuscript. While we were referring to non-linear accumulators, such as the Wang model, used in Hunt et al (2012), which predict magnitude effects in addition to the difference value effects predicted by the DDM and its variants, the reviewer is exactly right that virtually all other models except Kelly et al (2021) have been mute on the issue of sensory and motor variability and how these substantially shape predictions. A primary reason that models typically do not include variable time before accumulation begins **and** after it ends is that such models would be overly flexible - they would not make specific predictions about what one should expect to see in signals of evidence accumulation - making their neural hypotheses difficult to test. They would further include parameter redundancy with respect to behavior - meaning that it would not be possible to infer such timing variability from behavior itself. To estimate such models additional independent estimates of those variability components are necessary. We have now emphasized this issue more in the results as well as the discussion and recommend that future research quantifies sensory and motor variability to generate falsifiable predictions specific to their study.

In the results when motivating the simulation approach, we write:

“However, both stimulus and motor processes can vary from trial to trial, leading to an evidence accumulation process that is not closely tied to either stimulus or response. **The only model explicitly and independently modeling these variance components is the neurally informed DDM** (Kelly, Corbett, & O'Connell, 2021).”

In the discussion section we now write a new paragraph:

“Perhaps more importantly, our results reveal that the very nature of the evidence accumulation signal expected in the study at hand may vary drastically as a function of decision parameters, such as stimulus or motor variability, the shape of the decision bound/urgency parameters etc. Rather than testing out of the box predictions, we therefore recommend that future research into signatures of evidence accumulation apply the analysis approaches outlined above to the empirical data as well as simulations from a best-fitting model informed by independent measures of stimulus and motor variability, and bound/urgency parameters (Kelly, Corbett, & O'Connell, 2021).”

Minor comments:

In the revised manuscript the authors now state “However, some work suggests that both stimulus and motor processes can vary from trial to trial, leading to an evidence accumulation process that is not closely tied to either stimulus or response (Kelly, Corbett, & O'Connell, 2021).” This phrasing seems to present the case as if the notion of trial-to-trial variability in sensory or motor processing times is a recent discovery when it has been well-established for many decades of research in psychophysics and neuroscience.

We apologize that we weren't clear. We did not intend to convey that the variability aspect is a recent discovery, but rather that incorporating it explicitly in sequential sampling models is. As noted above already, we have now emphasized the uniqueness of the modeling approach in Kelly et al (2021), which hopefully clarifies this ambiguity. For your convenience we copy the relevant text again here:

“However, both stimulus and motor processes can vary from trial to trial, leading to an evidence accumulation process that is not closely tied to either stimulus or response. **The only model that explicitly and independently accounts for these variance components is the neurally informed DDM** (Kelly, Corbett, & O'Connell, 2021).”

Changes to the manuscript are not clearly marked in bold and in some cases are not indicated in the reviewer comments either.

We apologize if we missed anything important. We definitely tried to put all the changes

we made into the response letter to minimize effort on the side of the reviewers.

Decision Letter, third revision:

17th June 2024

Dear Dr. Froemer,

Thank you for your patience as we've prepared the guidelines for final submission of your Nature Human Behaviour manuscript, "Common neural choice signals emerge artifactually amidst multiple distinct value signals" (NATHUMBEHAV-23030973C). Please carefully follow the step-by-step instructions provided in the attached file, and add a response in each row of the table to indicate the changes that you have made. Please also address the additional marked-up edits we have proposed within the reporting summary. Ensuring that each point is addressed will help to ensure that your revised manuscript can be swiftly handed over to our production team.

We would hope to receive your revised paper, with all of the requested files and forms within two-three weeks. Please get in contact with us if you anticipate delays.

Nature Human Behaviour offers a Transparent Peer Review option for new original research manuscripts submitted after December 1st, 2019. As part of this initiative, we encourage our authors to support increased transparency into the peer review process by agreeing to have the reviewer comments, author rebuttal letters, and editorial decision letters published as a Supplementary item. When you submit your final files please clearly state in your cover letter whether or not you would like to participate in this initiative. Please note that failure to state your preference will result in delays in accepting your manuscript for publication.

In recognition of the time and expertise our reviewers provide to Nature Human Behaviour's editorial process, we would like to formally acknowledge their contribution to the external peer review of your manuscript entitled "Common neural choice signals emerge artifactually amidst multiple distinct value signals". For those reviewers who give their assent, we will be publishing their names alongside the published article.

Cover suggestions

We welcome submissions of artwork for consideration for our cover. For more information, please see our guide for cover artwork.

ORCID

Non-corresponding authors do not have to link their ORCIDs but are encouraged to do so. Please note that it will not be possible to add/modify ORCIDs at proof. Thus, please let your co-authors

know that if they wish to have their ORCID added to the paper they must follow the procedure described in the following link prior to acceptance:

Nature Human Behaviour has now transitioned to a unified Rights Collection system which will allow our Author Services team to quickly and easily collect the rights and permissions required to publish your work. Approximately 10 days after your paper is formally accepted, you will receive an email in providing you with a link to complete the grant of rights. If your paper is eligible for Open Access, our Author Services team will also be in touch regarding any additional information that may be required to arrange payment for your article.

Please note that *Nature Human Behaviour* is a Transformative Journal (TJ). Authors may publish their research with us through the traditional subscription access route or make their paper immediately open access through payment of an article-processing charge (APC). Authors will not be required to make a final decision about access to their article until it has been accepted. Find out more about Transformative Journals

[REDACTED]

Best regards,

[REDACTED]

On behalf of

[REDACTED]

Reviewer #1:

Remarks to the Author:

I thank the authors once again for their detailed response to my latest round of comments. I am glad that they now acknowledge the crucial point that without knowing the true ratio of pre- to post-decision time variability, the Unfold results cannot be taken as proof that the CPP is not an evidence accumulation signal. Instead, I think we are now in agreement that the paper serves to highlight that A) the CPP does not appear to trace value-based decisions and B) an evidence-dependent build-up in grand average pre-response waveforms cannot be relied upon as a unique characteristic of evidence accumulation signals. However, while the authors acknowledge these points clearly in the Response to Reviewer document, only the bare minimum of changes have been made to the manuscript. The main discussion points at the centre of the review are given a very brief, vague and peripheral treatment in the revised text while there are still numerous parts of the manuscript (most prominently the Title, Abstract and Discussion, see below for specific examples) which are written in such a way as to give the casual reader the strong impression that the paper is establishing that the

CPP is not an evidence accumulation signal at all and that ALL previously reported EA characteristics 'arise artifactually' (the Title being the most obvious case in point). This may well be unintentional on the part of the authors but the reality is that they are having it both ways - leading with an eye catching claim and only presenting some qualifications that are buried in the final paragraph of the Discussion. The authors are still not taking sufficient care to properly articulate what their paper does and does not show. Nevertheless, I think these issues could be quite easily addressed with more comprehensive changes to the text and without necessitating any further analyses.

Specific Comments:

First, regarding the new analysis of Steinemann et al's data examining stimulus- versus response-locked components as a function of RT. Here something appears to be amiss because the fast/slow Unfold Deconvolution waveforms in panel B exhibit substantial baseline differences between -200 and 0. Applying a baseline correction would result in the stimulus-aligned peak being larger for fast than slow trials, in line with the predictions for an evidence accumulation signal.

Second, the authors state in their Response "According to them (the reviewer), since we cannot unequivocally show this (that the CPP is not an evidence accumulation signal), the conclusion then is that we must assume that these signals reflect evidence accumulation by default." I am disappointed that after several detailed exchanges, that I have apparently been so unclear in my statements as to allow for such an interpretation of my position. The authors have now acknowledged that without knowing the ratio of pre- to post-decision time variability, we cannot make predictions regarding how Unfold will deal with an evidence accumulation signal. They also appear to acknowledge now that the observation that Unfold eliminates grand-average pre-response build-up effects tells us nothing specific about the CPP – it neither proves nor disproves that it is an evidence accumulation signal. Consequently, just as it is unjustified to conclude from these results that the CPP is an EA signal, it is equally unjustified to conclude that it is not one (again my focus is on the perceptual choice tasks). What the paper does highlight is that the assumption that only an evidence accumulation signal can exhibit pre-response average build-up rate effects is invalid. These are completely different things and the distinction is currently still not adequately made in the manuscript until maybe the final paragraph of the Discussion.

If any default position was implied by my prior comments I think it was because, separate from the issue of clarifying what the authors have and have not shown with their results, I have also been repeatedly trying to get the authors to engage with the substantial body of research that has been dedicated to characterising the functional characteristics of the CPP without solely relying on assumptions about pre-response build-up rate. Having established that the authors' results show that pre-response build-up rate effects are not unique to EA signals, it behoves them then to cover in the Discussion the fact that quite a few studies have used other means of testing for EA signatures in the CPP. The authors work calls into question one line of evidence in favour of the CPP=EA theory but in ignoring the other lines of evidence they risk leading many readers to the wrong conclusion and do a disservice to their colleagues in the field. Toward the end of the Discussion (last paragraph of page 30) the authors can only point to one finding that they find difficult to reconcile with the idea that the CPP is purely the product of signal overlap (O'Connell et al 2012) which was suggested to them by me. But I had flagged that there were a range of other findings and these have still not made it in to the paper. Aside from the work from the Kelly and O'Connell groups already flagged in previous comments (including Devine et al 2018, eLife which analysed subtle stimulus perturbation effects at the single trial level in a random dot motion task and Kelly et al 2021, see comment below), this includes numerous papers using reverse correlation analyses by Wyart/Summerfield and using linear discriminant analyses by the Philiastides group, all of which highlight centro-parietal signals consistent with evidence accumulation without recourse to response-aligned build-up rate effects. Again, keeping in mind that we know from the simulations that Unfold can in some realistic circumstances eliminate grand-average build-up rate effects of even a true evidence accumulation signal, the authors owe it to the field to provide a more rounded discussion of their results that properly engages with the previous literature. Yes, they have shown that some assumptions about defining characteristics of evidence accumulation signals are problematic but it is also their responsibility to ensure that the reader not come away with the impression that these assumptions have been the only method used to test for EA signatures in the CPP.

So again, as alluded to above, my issue is with a sense of the authors trying to have it both ways. I don't see that the impact or readership of the paper would be in any way diminished by clearly stating that the work highlights that pre-response build-up effects are not a unique identifier of

evidence accumulation signals without then going the further unwarranted step of stating that 'common neural choice signals arise artifactually'. The authors could easily amend the text so that it is clear that this particular characteristic of those signals might or could arise artifactually on perceptual choice tasks but that the current data do not tell us whether they actually do or not. And then it costs the authors nothing to ponder other results pertaining to the CPP which do not hinge on this particular assumption and thus provide a fair treatment of the body of work contributed by several research groups.

Third, below are a list of suggested amendments to the text which I think would be needed to avoid misleading the reader:

1. The Title and the last sentence of the Abstract are at best highly ambiguous and very likely to mislead the reader. Are the authors claiming that the CPP in the perceptual choices tasks also reflects value signals? If so, this is not explained anywhere in the manuscript. If the authors are in fact referring to the value choice task specifically then amendments to the text are required because I think most readers will assume this is a general statement that applies to all tasks.
2. I think some confusion arises from the fact that the authors might be using the term 'signal' not to refer to the CPP (i.e. a centro-parietal positivity that emerges during decision formation, a label whose use does not necessitate embracing its proposed role in evidence accumulation) but specifically to pre-response grand average evidence-dependent build-up? If so, I would suggest reserving the word 'signal' for the CPP and referring to 'characteristics' (e.g. 'common choice signal characteristics') or a term like that when referring to build-up rate effects. This would help create separation between the demonstration that this particular signal characteristic is not unique to evidence accumulation signals and the claim that the CPP is not an evidence accumulation signal.
3. In the Discussion the authors state 'Across six datasets, we found that evidence accumulation signatures in the response-locked CPP MAY artificially arise from response time-dependent overlap with stimulus-related processing.' The title needs to be amended to also include the word 'may' or 'could'.
4. Abstract "Here, we use the temporal resolution of EEG to test whether choice-independent value signals could be dissociated from well-known temporal signatures of the choice process." There is a crucial piece of this sentence missing because the study is only testing whether such signals can be dissociated "based on the extent to which their grand average, response-aligned waveforms exhibit a build-up rate that scales with evidence strength." This is important again in light of other research that has sought to dissociate such signals using other means (mentioned above). This collection of studies should also be covered in the Discussion. Otherwise, at present, the authors unintentionally give the strong impression that trial-average response-aligned build-up rate effects are the ONLY method thus far used to validate the CPP as an evidence accumulation signal and this would be a gross misrepresentation of the literature.
5. Second last sentence of the Abstract 'we show that this signal disappears'. This statement too is highly misleading and this again probably relates to my first point about the distinction between 'signal' and 'characteristic'. According to the authors, Unfold does not make the CPP disappear but rather it eliminates the effect of evidence strength on its pre-response build-up rate. The signal itself (a centro-parietal positivity whether it is tracing evidence accumulation or not) is presumably still in the data but either shifted to the stimulus-locked component or spread across both. The authors have acknowledged this in their latest round of comments but this is not reflected in their persisting with such an eye catching statement in the Abstract which will give readers a wrong impression of what the study shows.
6. Introduction, I think the authors should flag at the end of their introduction where they summarize the results that their findings point to problems with one particular prediction regarding evidence accumulation signals whose validity has been assumed in some previous work. As written, there is a clear sense that the authors are indicating that they are showing that the CPP is not an evidence accumulation signal. Some clarification of the limits of the conclusions that can be drawn are necessary to prevent the less informed reader from getting the wrong impression.
7. Discussion: "Doing so, several studies have shown that evidence accumulation signals simulated from diffusion model fits to behavior reproduce the CPP in detail, including its stimulus and response-aligned waveforms, the relationship between its pre-response amplitude and choice RT and accuracy and modulations of its pre-response amplitude by experimental manipulations of time pressure and prior knowledge (e.g., Kelly, Corbett, & O'Connell, 2021; O'Connell, Dockree, & Kelly, 2012; Twomey et al., 2015). Here, we highlight several cases in which such seemingly strong

evidence could emerge artifactually from component overlap by virtue of evidence accumulation and component overlap sharing a common link to response time." I find it hard to fathom how this statement is retained by the authors after these discussions and the acknowledgments they now make in the Response. This statement clearly implies that the correspondences between model simulated EA waveforms and the CPP arose spuriously, including the amplitude modulations by time pressure and prior knowledge (and presumably also the inverted u-shaped relationship between CPP/simulated DV and RT that is shown in the paper too). The authors fail to mention for example that Kelly et al's model sought to account for pre- and post-decision delays (deriving empirical estimates of post-decision delays from ERP motor potentials), nor do they provide any discussion of how such a close correspondence in both stimulus and response aligned traces could arise from signal overlap. In their Response the authors provide a nice discussion of why Unfold might have allocated the EA activity to the stimulus-aligned component when pre-/post-decision time variability was equal but none of that makes it into the manuscript. Surely such considerations are crucial to the reader properly understanding the authors' results? And yet, in the very next paragraph the authors mention Kelly et al's model-based approach as offering a useful method of circumventing these issues with signal alignment so there is a clear contradiction here. Again, it just feels like the authors are keen to make a big claim and are avoiding engaging with previous work that undermines that claim. I think the authors should acknowledge that quite a few previous studies have provided evidence in favour of the CPP being an EA signal without resting on the assumption regarding pre-response build-up rate effects that is demonstrated here to be problematic.

8. Discussion first paragraph 'Remarkably, though, we found that choice-related activity was inconsistent with evidence accumulation, and that instead, putative signatures of evidence accumulation can emerge artifactually in standard response-locked analyses from overlapping stimulus-related activity.' Again, the authors have now acknowledged that the results are not necessarily inconsistent with evidence accumulation so why is this statement left in?

Reviewer #3:

Remarks to the Author:

The authors have admirably endeavored to address the concerns of Review 1. I am mostly convinced by the authors' arguments in the newest version of the manuscript and responses to Review 1. In my opinion, I do not think that the concerns of Reviewer 1 will ever be completely addressed with the current manuscript, and Reviewer 1's concerns would be more useful as a Commentary or response paper. I do not completely disagree with Review 1, as some of the authors' later arguments rely on the assumptions of Unfold. But I believe that future studies should help clarify this debate.

If the manuscript is further held up in review by adding additional analyses to debate with Review 1, there is a large risk of filling the paper with tangential analyses that obfuscate the authors' original important scientific arguments. In fact, my first Minor comment was likely caused by attempting to satisfy Review 1.

Minor

More details are necessary to understand the analysis for Supplementary Figure 7.

In general, I still feel as though the manuscript could have additional supplementary methods to explain Unfold and now the new simulations instead of relying on citations to other sources. However, the authors have provided analysis code that will aid technically-minded readers. I also blame the paper format of Nature (Human Behavior) that de-emphasizes Methods in favor of Results.

Decision Letter, fourth revision:

Reviewer #1:

Remarks to the Author:

I thank the authors once again for their detailed response to my latest round of comments. I am glad that they now acknowledge the crucial point that without knowing the true ratio of pre- to post-decision time variability, the Unfold results cannot be taken as proof that the CPP is not an evidence accumulation signal. Instead, I think we are now in agreement that the paper serves to highlight that A) the CPP does not appear to trace value-based decisions and B) an evidence-dependent build-up in grand average pre-response waveforms cannot be relied upon as a unique characteristic of evidence accumulation signals. However, while the authors acknowledge these points clearly in the Response to Reviewer document, only the bare minimum of changes have been made to the manuscript. The main discussion points at the centre of the review are given a very brief, vague and peripheral treatment in the revised text while there are still numerous parts of the manuscript (most prominently the Title, Abstract and Discussion, see below for specific examples) which are written in such a way as to give the casual reader the strong impression that the paper is establishing that the CPP is not an evidence accumulation signal at all and that ALL previously reported EA characteristics 'arise artifactually' (the Title being the most obvious case in point). This may well be unintentional on the part of the authors but the reality is that they are having it both ways - leading with an eye catching claim and only presenting some qualifications that are buried in the final paragraph of the Discussion. The authors are still not taking sufficient care to properly articulate what their paper does and does not show. Nevertheless, I think these issues could be quite easily addressed with more comprehensive changes to the text and without necessitating any further analyses.

Specific Comments:

First, regarding the new analysis of Steinemann et al's data examining stimulus- versus response-locked components as a function of RT. Here something appears to be amiss because the fast/slow Unfold Deconvolution waveforms in panel B exhibit substantial baseline differences between -200 and 0. Applying a baseline correction would result in the stimulus-aligned peak being larger for fast than slow trials, in line with the predictions for an evidence accumulation signal.

We agree with the reviewer that in this specific analysis it looks like a baseline correction would lead to a larger peak for fast compared to slow trials. We are hesitant to place too much weight on this possible result alone, because the approach we have taken here, to analyze median-split RT differences, will inevitably lump together a range of different trial types that ultimately have a similar issue as the MASS-univariate analysis would – even if it is to a lesser extent. If we still found a difference of that sort in the complementary residual analysis (Supplementary Figure 7), this would be convincing evidence that there is a reliable signature of evidence accumulation in this data. However, we do not see this. To do justice to the ambiguity, we now mention this in the text for the reader to be fully informed.

We write:

“For Study 5, the results are somewhat ambiguous due to a significant baseline difference, which if taken into account may produce a stimulus-locked peak difference in the expected direction. However, these findings are complemented by results showing that signatures of evidence accumulation can no longer be 54

found using MASS-univariate analysis following RT-agnostic overlap correction of the data (Supplementary Figure 7).”

Second, the authors state in their Response “According to them (the reviewer), since we cannot unequivocally show this (that the CPP is not an evidence accumulation signal), the conclusion then is that we must assume that these signals reflect evidence accumulation by default.” I am disappointed that after several detailed exchanges, that I have apparently been so unclear in my statements as to allow for such an interpretation of my position. The authors have now acknowledged that without knowing the ratio of pre- to post-decision time variability, we cannot make predictions regarding how Unfold will deal with an evidence accumulation signal. They also appear to acknowledge now that the observation that Unfold eliminates grand-average pre-response build-up effects tells us nothing specific about the CPP – it neither proves nor disproves that it is an evidence accumulation signal. Consequently, just as it is unjustified to conclude from these results that the CPP is an EA signal, it is equally unjustified to conclude that it is not one (again my focus is on the perceptual choice tasks). What the paper does highlight is that the assumption that only an evidence accumulation signal can exhibit pre-response average build-up rate effects is invalid. These are completely different things and the distinction is currently still not adequately made in the manuscript until maybe the final paragraph of the Discussion.

If any default position was implied by my prior comments I think it was because, separate from the issue of clarifying what the authors have and have not shown with their results, I have also been repeatedly trying to get the authors to engage with the substantial body of research that has been dedicated to characterising the functional characteristics of the CPP without solely relying on assumptions about pre-response build-up rate. Having established that the authors’ results show that pre-response build-up rate effects are not unique to EA signals, it behoves them then to cover in the Discussion the fact that quite a few studies have used other means of testing for EA signatures in the CPP. The authors work calls into question one line of evidence in favour of the CPP=EA theory but in ignoring the other lines of evidence they risk leading many readers to the wrong conclusion and do a disservice to their colleagues in the field. Toward the end of the Discussion (last paragraph of page 30) the authors can only point to one finding that they find difficult to reconcile with the idea that the CPP is purely the product of signal overlap (O’Connell et al 2012) which was suggested to them by me. But I had flagged that there were a range of other findings and these have still not made it in to the paper. Aside from the work from the Kelly and O’Connell groups already flagged in previous comments (including Devine et al 2018, eLife which analysed subtle stimulus perturbation effects at the single trial level in a random dot motion task and Kelly et al 2021, see comment below), this includes numerous papers using reverse correlation analyses by Wyart/Summerfield and using linear discriminant analyses by the Philiastides group, all of which highlight centro-parietal signals consistent with evidence accumulation without recourse to response-aligned build-up rate effects.

Again, keeping in mind that we know from the simulations that Unfold can in some realistic circumstances eliminate grand-average build-up rate effects of even a true evidence accumulation signal, the authors owe it to the field to provide a more rounded discussion of their results that properly engages with the previous literature. Yes, they have shown that some assumptions about defining characteristics of evidence accumulation signals are problematic but it is also their responsibility to ensure that the reader not come away with the impression that these assumptions have been the only method used to test for EA signatures in the CPP.

So again, as alluded to above, my issue is with a sense of the authors trying to have it both ways. I don't see that the impact or readership of the paper would be in any way diminished by clearly stating that the work highlights that pre-response build-up effects are not a unique identifier of evidence accumulation signals without then going the further unwarranted step of stating that 'common neural choice signals arise artifactually'. The authors could easily amend the text so that it is clear that this particular characteristic of those signals might or could arise artifactually on perceptual choice tasks but that the current data do not tell us whether they actually do or not. And then it costs the authors nothing to ponder other results pertaining to the CPP which do not hinge on this particular assumption and thus provide a fair treatment of the body of work contributed by several research groups.

We thank the reviewer for their candour. We believe there is a disconnect between what we are trying to say and what the reviewer is hearing. This might be because we have been unclear about the differentiation between response-locked evidence accumulation signals as commonly studied in decision-making, and the CPP specifically. We had expected to identify both choice-related and choice-unrelated neural value signals in value-based decision-making. We happened to not find the choice related value signals we expected based on the literature and we asked why. We identified component overlap as an issue in common approaches to studying evidence accumulation. At no point do we try to conclude that the CPP does or does not reflect evidence accumulation. We have been very careful about that because the studies here are not designed to probe this. We agree with you that not every reader will carefully read the manuscript, and so we have tried to clear up the more ambiguous passages as described below.

We now also include additional citations for testing for signatures of evidence accumulation in the CPP specifically and we have added another sentence to the paragraph on perturbations:

Along those lines, some studies have employed repeated stimulus presentations to better characterize how signatures of evidence accumulation evolve with each piece of evidence^{75,76}.

Third, below are a list of suggested amendments to the text which I think would be needed to avoid misleading the reader:

1. The Title and the last sentence of the Abstract are at best highly ambiguous and very

likely to mislead the reader. Are the authors claiming that the CPP in the perceptual choices tasks also reflects value signals? If so, this is not explained anywhere in the manuscript. If the authors are in fact referring to the value choice task specifically then amendments to the text are required because I think most readers will assume this is a general statement that applies to all tasks.

This paper is primarily about value signals in value-based choice. Our title aims to indicate that we show multiple value signals in value based choice (Appraisal and Choice PC clusters, as we stated in the abstract), and that the response-locked EA characteristic can emerge artifactually. Only that latter finding is related to the CPP. Our main findings on value-based choice are unrelated to the CPP except that we expected one of those to be the CPP and it was not. To address the question, we do not think the CPP in perceptual choice is a value signal. We hope that the revised abstract is clearer.

The final abstract reads as follows:

Previous work has identified characteristic neural signatures of value-based decision-making, including neural dynamics that closely resemble the ramping evidence accumulation process believed to underpin choice. Here, we test whether these signatures of the choice process can be temporally dissociated from additional, choice-*independent* value signals. Indeed, EEG activity during value-based choice revealed distinct spatiotemporal clusters, with a stimulus-locked cluster reflecting affective reactions to choice sets and a response-locked cluster reflecting choice difficulty. Surprisingly, *neither* of these clusters met the criteria for an evidence accumulation signal. Instead, we found that stimulus-locked activity can *mimic* an evidence accumulation process when aligned to the response. Re-analyzing four previous studies – including three perceptual decision-making studies – we show that response-locked signatures of evidence accumulation disappear when stimulus-locked and response-locked activity are modelled jointly. Collectively, our findings show that neural signatures of value can reflect choice-independent processes and look deceptively like evidence accumulation.

2. I think some confusion arises from the fact that the authors might be using the term ‘signal’ not to refer to the CPP (i.e. a centro-parietal positivity that emerges during decision formation, a label whose use does not necessitate embracing its proposed role in evidence accumulation) but specifically to pre-response grand average evidence-dependent build-up? If so, I would suggest reserving the word ‘signal’ for the CPP and referring to ‘characteristics’ (e.g. ‘common choice signal characteristics’) or a term like that when referring to build-up rate effects. This would help create separation between the demonstration that this particular signal characteristic is not unique to evidence accumulation signals and the claim that the CPP is not an evidence accumulation signal.

We agree that we need to better differentiate between individual characteristics and the CPP as such, and we believe we have now done this through additional edits. However, we use signal consistently throughout the manuscript in the sense of an EEG correlate (including value signal in the title). We are hesitant to deviate from this by using signal to exclusively refer to the CPP.

3. In the Discussion the authors state ‘Across six datasets, we found that evidence accumulation signatures in the response-locked CPP MAY artificially arise from response time-dependent overlap with stimulus-related processing.’ The title needs to be amended to also include the word ‘may’ or ‘could’.

We have now amended the title according to this suggestion and include ‘can’.

4. Abstract “Here, we use the temporal resolution of EEG to test whether choice-independent value signals could be dissociated from well-known temporal signatures of the choice process.” There is a crucial piece of this sentence missing because the study is only testing whether such signals can be dissociated “based on the extent to which their grand average, response-aligned waveforms exhibit a build-up rate that scales with evidence strength.” This is important again in light of other research that has sought to dissociate such signals using other means (mentioned above). This collection of studies should also be covered in the Discussion. Otherwise, at present, the authors unintentionally give the strong impression that trial-average response-aligned build-up rate effects are the ONLY method thus far used to validate the CPP as an evidence accumulation signal and this would be a gross misrepresentation of the literature.

The specific sentence quoted here needs to be considered in context, which is value-based decision-making and other processes that may transpire in parallel. This sentence was about our original hypothesis that we would identify appraisal signals that are distinct from choice signals, such as the CPP. Unless the reviewers are claiming that these appraisal signals are evidence accumulation signals of some sort, there should not be an issue with this sentence. This is however a moot point, since we had to cut the abstract in half to meet the word limit and in that process have removed all references to the CPP from the abstract. We are now only referring to response-locked signatures of evidence accumulation and are hopefully distinguishing more clearly between our original study goals and our incidental results.

5. Second last sentence of the Abstract ‘we show that this signal disappears’. This statement too is highly misleading and this again probably relates to my first point about the distinction between ‘signal’ and ‘characteristic’. According to the authors, Unfold does not make the CPP disappear but rather it eliminates the effect of evidence strength on its pre-response build-up rate. The signal itself (a centro-parietal positivity whether it is tracing evidence accumulation or not) is presumably still in the data but either shifted to the stimulus-locked component or spread across both. The authors have acknowledged this in their latest round of comments but this is not reflected in

their persisting with such an eye catching statement in the Abstract which will give readers a wrong impression of what the study shows.

As shown above, we have edited the abstract significantly so that this comment is addressed.

6. Introduction, I think the authors should flag at the end of their introduction where they summarize the results that their findings point to problems with one particular prediction regarding evidence accumulation signals whose validity has been assumed in some previous work. As written, there is a clear sense that the authors are indicating that they are showing that the CPP is not an evidence accumulation signal. Some clarification of the limits of the conclusions that can be drawn are necessary to prevent the less informed reader from getting the wrong impression.

This was not our intention. We have 1) removed reference to the CPP in this summary and 2) specified that this is about response-locked signatures of evidence accumulation. The edited sentence reads like this:

Confirming this, when we apply a novel analysis approach that deconvolves stimulus and response-related activity to four previous decision-making studies, we eliminate response-locked signatures of evidence accumulation previously observed in those data.

7. Discussion: "Doing so, several studies have shown that evidence accumulation signals simulated from diffusion model fits to behavior reproduce the CPP in detail, including its stimulus and response aligned waveforms, the relationship between its pre-response amplitude and choice RT and accuracy and modulations of its pre-response amplitude by experimental manipulations of time pressure and prior knowledge (e.g., Kelly, Corbett, & O'Connell, 2021; O'Connell, Dockree, & Kelly, 2012; Twomey et al., 2015). Here, we highlight several cases in which such seemingly strong evidence could emerge artifactually from component overlap by virtue of evidence accumulation and component overlap sharing a common link to response time." I find it hard to fathom how this statement is retained by the authors after these discussions and the acknowledgments they now make in the Response. This statement clearly implies that the correspondences between model simulated EA waveforms and the CPP arose spuriously, including the amplitude modulations by time pressure and prior knowledge (and presumably also the inverted u-shaped relationship between CPP/simulated DV and RT that is shown in the paper too). The authors fail to mention for example that Kelly et al's model sought to account for pre- and post-decision delays (deriving empirical estimates of post-decision delays from ERP motor potentials), nor do they provide any discussion of how such a close correspondence in both stimulus and response aligned traces could arise from signal overlap. In their Response the authors provide a nice discussion of why Unfold might have allocated the EA activity to the stimulus-aligned component when pre-/post-decision time variability was equal but none of that makes it into the manuscript. Surely such considerations are crucial to the reader properly understanding the authors' results? And yet, in the very next paragraph the authors mention Kelly et al's model-based approach as offering a useful method

circumventing these issues with signal alignment so there is a clear contradiction here. Again, it just feels like the authors are keen to make a big claim and are avoiding engaging with previous work that undermines that claim. I think the authors should acknowledge that quite a few previous studies have provided evidence in favour of the CPP being an EA signal without resting on the assumption regarding pre-response build-up rate effects that is demonstrated here to be problematic.

We apologize, this was unintentional and came about inadvertently when editing a previous version of the paper to address an earlier reviewer concern. The original point was to provide advice for how to avoid misinterpretation of – primarily response-locked – signatures of evidence accumulation. One recommendation was to test not only for response-locked but also other signatures of evidence accumulation, for instance, the stimulus-locked pattern in the original CPP paper. Certainly, more than simply looking at the response-locked pattern would need to be done. However, even for moving stimulus-locked peaks and response-locked correlations with evidence accumulation patterns there could be concerns about component overlap that would need to be addressed. Based on a reviewer suggestion we later also appended the other criteria. In order to avoid the potential implication that we are calling into question these specific papers we are citing to provide resources for testing multiple criteria, we have now modified the sentence as follows:

Here, we highlight several cases in which some of the relevant characteristics could emerge artifactually from component overlap by virtue of evidence accumulation and component overlap sharing a common link to response time.

8. Discussion first paragraph ‘Remarkably, though, we found that choice-related activity was inconsistent with evidence accumulation, and that instead, putative signatures of evidence accumulation can emerge artifactually in standard response-locked analyses from overlapping stimulus-related activity.’ Again, the authors have now acknowledged that the results are not necessarily inconsistent with evidence accumulation so why is this statement left in?

In context, this sentence is referring specifically to the value-based data, where the reviewer agreed that the findings are not consistent with evidence accumulation. We now make this clearer:

Here, we tested whether we could use EEG to temporally dissociate such choice-independent value signals from choice-related value signals. We anticipated that choice-independent value signals would follow shortly after stimulus onset, whereas choice-related activity should be coupled to and lead up to the response. We found this expected temporal dissociation. Remarkably, though, we found that the identified choice-related activity was inconsistent with evidence accumulation, and that instead, putative signatures of evidence accumulation can emerge artifactually in standard response-locked analyses from overlapping stimulus-related activity.

Reviewer #3:

Remarks to the Author:

The authors have admirably endeavored to address the concerns of Review 1. I am mostly convinced by the authors' arguments in the newest version of the manuscript and responses to Review 1. In my opinion, I do not think that the concerns of Reviewer 1 will ever be completely addressed with the current manuscript, and Reviewer 1's concerns would be more useful as a Commentary or response paper. I do not completely disagree with Review 1, as some of the authors' later arguments rely on the assumptions of Unfold. But I believe that future studies should help clarify this debate.

If the manuscript is further held up in review by adding additional analyses to debate with Review 1, there is a large risk of filling the paper with tangential analyses that obfuscate the authors' original important scientific arguments. In fact, my first Minor comment was likely caused by attempting to satisfy Review 1.

We thank the reviewer for their acknowledgement of our efforts.

Minor

More details are necessary to understand the analysis for Supplementary Figure 7. We have now added the following to the caption to Figure 7 to provide more detail:

Data were analyzed with unfold modeling only the occurrence of stimulus and response events. Activity associated with these events – that is constant across all trials – was then subtracted from the raw EEG. These residualized data were then analyzed with standard MASS-univariate analyses.

In general, I still feel as though the manuscript could have additional supplementary methods to explain Unfold and now the new simulations instead of relying on citations to other sources. However, the authors have provided analysis code that will aid technically-minded readers. I also blame the paper format of Nature (Human Behavior) that de-emphasizes Methods in favor of Results.

Final Decision Letter: